# Climate-forced Hg-remobilization associated with fern mutagenesis in the aftermath of the end-Triassic extinction

Remco Bos [1] ✉, Wang Zheng [2] ✉, Sofie Lindström [3], Hamed Sanei [4], Irene Waajen [1], Isabel M. Fendley [5,6], Tamsin A. Mather [5], Yang Wang[2], Jan Rohovec[7], Tomáš Navrátil [7], Appy Sluijs [1] & Bas van de Schootbrugge [1]

The long-term effects of the Central Atlantic Magmatic Province, a large igneous province connected to the end-Triassic mass-extinction (201.5 Ma), remain largely elusive. Here, we document the persistence of volcanic-induced mercury (Hg) pollution and its effects on the biosphere for ~1.3 million years after the extinction event. In sediments recovered in Germany (Schandelah-1 core), we record not only high abundances of malformed fern spores at the Triassic-Jurassic boundary, but also during the lower Jurassic Hettangian, indicating repeated vegetation disturbance and stress that was eccentricity-forced. Crucially, these abundances correspond to increases in sedimentary Hg-concentrations. Hg-isotope ratios ($\delta^{202}$Hg, $\Delta^{199}$Hg) suggest a volcanic source of Hg-enrichment at the Triassic-Jurassic boundary but a terrestrial source for the early Jurassic peaks. We conclude that volcanically injected Hg across the extinction was repeatedly remobilized from coastal wetlands and hinterland areas during eccentricity-forced phases of severe hydrological upheaval and erosion, focusing Hg-pollution in the Central European Basin.

The emplacement of the Central Atlantic Magmatic Province (CAMP) is often causally linked to the end-Triassic mass extinction (ETME, ~201.3 Ma)[1–4]. Several lines of evidence indicate phased and prolonged volcanism that emitted vast amounts of greenhouse gasses and pollutants to the atmosphere across the Triassic–Jurassic boundary (TJB)[5–7]. Global warming of 3–6 °C[8] due to a strong rise in atmospheric $p$CO$_2$[9] is commonly seen as the main driver of the ETME[10]. Two negative excursions in the stable carbon isotopic composition ($\delta^{13}$C) of organic carbon due to a release of isotopically light carbon into the ocean–atmosphere system coincide with phases of increased extinction rates in the marine and terrestrial realms[11,12]. In addition to global warming, a combination of acid rain, wildfires, increased seasonality,

and weathering rates are thought to have contributed to terrestrial extinction[13–15]. Ocean anoxia and severe shallow-shelf euxinia were likely a direct consequence of warming (decreased oxygen solubility) and increased nutrient/weathering flux (harmful algal blooms), contributing further to marine extinction rates[16–20].

A distinct feature that has been recorded for both the end-Permian and end-Triassic extinctions is the high abundance of malformed pollen and spores[2,21]. This has been interpreted to reflect volcanic halogen emissions causing ozone depletion and heat stress in vegetation[22–24]. More recently, sedimentary mercury (Hg) anomalies in TJB sections in Sweden and Denmark were linked to plant extinction and the profusion of malformed fern spores[2]. As one of the most toxic

[1]Department of Earth Sciences, Faculty of Geosciences, Utrecht University, Princetonlaan 8, 3584 CB Utrecht, The Netherlands. [2]Institute of Surface-Earth System Science, School of Earth System Science, Tianjin University, 300072 Tianjin, China. [3]Department of Geosciences and Natural Resource Management, Copenhagen University, Øster Voldgade 10, DK-1350 Copenhagen K, Denmark. [4]Lithospheric Organic Carbon (LOC) Group, Department of Geoscience, Aarhus University, Høegh-Guldbergs gade 2, 8000C Aarhus, Denmark. [5]Department of Earth Sciences, University of Oxford, Parks Road, Oxford OX1 3PR, UK. [6]Department of Geosciences, Pennsylvania State University, University Park, PA 16802, USA. [7]Institute of Geology of the Czech Academy of Sciences, Rozvojová 269, Prague 6 165 00, Czech Republic. ✉e-mail: r.bos@uu.nl; zhengw3@tju.edu.cn

metals on Earth, Hg can induce DNA damage through oxidative stress and cause stomatal closure and visible injuries such as lesions[25,26]. Furthermore, Hg-induced mutagenic changes in plant DNA will be transferred to the reproductive cells (e.g., spores) when DNA repair mechanisms become dysfunctional[2,27]. Coeval plant extinction and mutagenesis could, therefore, be an expression of environmental Hg-toxicity. Sedimentary Hg-enrichments have been used to trace volcanic activity in deep time and, in some cases, support a direct cause-and-effect relationship between volcanism and extinction[1,2,28-31]. TJB sections across northwestern (NW) Europe (Fig. 1) have been extensively studied for Hg concentrations and additional records in China[32], Nevada[30], Greenland, and Argentina[28] indicate the wide reach of CAMP emissions.

Volcanism is a source of gaseous Hg to the atmosphere, and large igneous province (LIP) volcanic eruptions are hypothesized to release substantial amounts of Hg to the global environment[33]. Furthermore, Hg is an ideal recorder for relatively short (~1 Myr duration) geological events, such as LIP eruptions that can be traced globally[28,34-36]. The low vapor pressure of elemental $Hg^0$ makes it susceptible to volatilization, and its long atmospheric residence time (0.5–2 years) provides the potential to be distributed globally in its gaseous phase[37]. Once oxidized, it becomes reactive gaseous mercury ($Hg^{2+}$) and can become bound to fine particles ($Hg_p$). Both $Hg^{2+}$ and $Hg_p$ are deposited via rainfall and/or particle fallout[38]. The speciation of Hg in terrestrial and marine reservoirs strongly affects its mobility and toxicity, and, therefore, its effects on the biosphere. Stable solid Hg forms include S-bound complexes (HgS-minerals and Hg–S complexes in sediments) and the highly toxic and bioaccumulating methylmercury (MeHg). MeHg is produced from $Hg^{2+}$ mainly through sulfate- and/or iron-reducing bacteria and then enters marine and terrestrial foodwebs[39]. In aqueous environments, Hg exists as dissolved $Hg^0$, dissolved MeHg and ligand-bound species (central-bound molecule), which can be reduced and mobilized to gaseous $Hg^0$ by microbial and photochemical processes[40]. Stomatal gaseous uptake of $Hg^0$ is the main pathway into vegetation[41] with this entry route also plausibly promoting mutagenesis in the plant's seeds/spores due to its proximity to the formation of the plant's reproductive organs and its circumvention of the root-protection systems[26]. Therefore, tracing Hg-cycling across the ETME, particularly Hg-degassing, can shed light on LIP-induced Hg-pollution and its potential long-term effects on the terrestrial vegetation. Previous studies have reported changes in Hg-cycling during CAMP phases[31,32], however, the long-term consequences of volcanically-degassed Hg in the environment following the cessation of LIP eruptions remain elusive.

Shifts in Hg sourcing, pathways, and mobilization can be traced using Hg-stable isotope signatures[32]. Hg-isotopic assessment through mass-dependent (MDF, $\delta^{202}Hg$) and mass-independent (MIF, $\Delta^{199}Hg$) fractionation shows a range of environmental interactions[42] (see the "Methods" section for a detailed description). MDF is found in nearly all kinetically controlled reactions, with resulting products having depleted $\delta^{202}Hg$ values while the remaining reactant pool retains heavier isotopes. In the terrestrial environment, stomatal uptake and storage of atmospheric $Hg^0$ in foliage results in strong negative MDF, causing significantly negative $\delta^{202}Hg$ in the terrestrial reservoir relative to the atmospheric reservoir[41]. Therefore, MDF can reflect changes in Hg-absorption of the terrestrial biosphere in coastal and shallow marine sediments. Photochemical reactions (e.g., photo-reduction and photo-demethylation) of bound-Hg produce positive MDF shifts in the retained fraction[42]. In contrast, MIF is often insignificant in biotic and most abiotic reactions. Instead, photochemical reactions play a key role in MIF variability of Hg in the environment[42]. Consequently, MIF is less prone to post-depositional processes in sediments and is considered a good tracer for Hg sources/contamination pathways[42]. MIF has previously been used to distinguish between pathways involved in Hg-enrichment of sediment across several LIP events[30-32]. Volcanic MIF is assumed to be near-zero ($\Delta^{199}Hg = -0.1‰$ to $+0.1‰$) but could develop positive values through atmospheric redox transformations during the dispersal of volcanic Hg[33]. Terrestrial Hg (i.e., in soils and vegetation) typically exhibits highly negative MIF, which is attributed to a photochemical reduction (loss of Hg) in foliage[41,43], whereby subsequent litterfall carries very low MIF values ($\Delta^{199}Hg = -0.6‰$ to $-0.2‰$)[42]. Depending on the type of Hg-bound and environment, photochemical reduction can produce both negative and positive shifts in MIF. However, degradation and photochemical reduction of Hg stored in terrestrial soil organics and bedrock causes mainly positive shifts in MIF in the retaining fraction and indicates Hg-degassing[42], potentially harmful to vegetation.

Here, we present combined bulk Hg-concentrations, Hg-stable isotope records, and fern spore abundance and malformations obtained from shallow marine sediments from the Lower Saxony Basin (Schandelah-1 core) in Germany (Fig. 1). The Schandelah-1 core provides a unique coastal margin archive of the late Rhaetian extinction (~201.5 Ma) to the early Sinemurian recovery (~196 Ma)[13]. Malformations in spores and pollen have been previously used to demonstrate stress and mutagenesis associated with major mass-extinction events[2,21]. Here, we quantify various types of trilete fern spores exhibiting patterns of abnormal development (i.e., teratology). Previously, the classification of spore malformations attempted to distinguish mutagenic severity based on morphological characteristics (Fig. 2 and supplementary Fig. 1)[2]. We adopt this approach to pinpoint intervals that show signs of elevated rates of mutagenesis. In order to assess the potential link between Hg-enrichments and/or volcanism to fern mutagenesis, we reconstruct Hg-cycling across the ETME and Early Jurassic to uncover the mobility and toxicity of environmental Hg. We hypothesize that increased amounts of gaseous forms of Hg in the environment would affect fern communities and produce malformed spores. Assessing changes in Hg-isotope compositions through the combined use of MDF and MIF records allows us to determine the main processes and/or sources that governed Triassic and Jurassic Hg-dynamics[44].

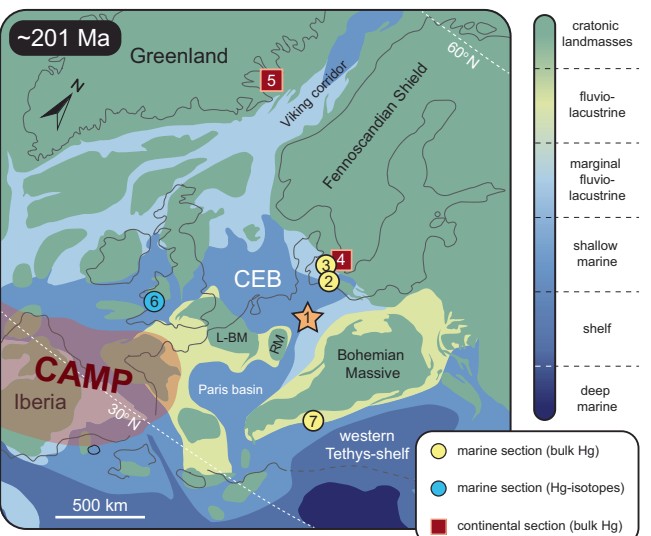

**Fig. 1 | Paleogeographic reconstruction of the European Epicontinental Seaway during the end-Triassic (~201 Ma). (1)** The orange star indicates the position of the Schandelah-1 drill-core (this study) in the Central European Basin (CEB). Circles and squares represent published marine and continental sections that contain Hg concentration and Hg-isotope data across the Triassic–Jurassic transition. These include (**2**) Rødby-1[2], (**3**) Stenlille-1/4[2], (**4**) Norra Albert/Albert-1[2], (**5**) Astartekløft[28], (**6**) St. Audrie's Bay[28,31] and (**7**) Kuhjoch[28]. RM Rhenish Massif, L-BM London-Brabant Massif.

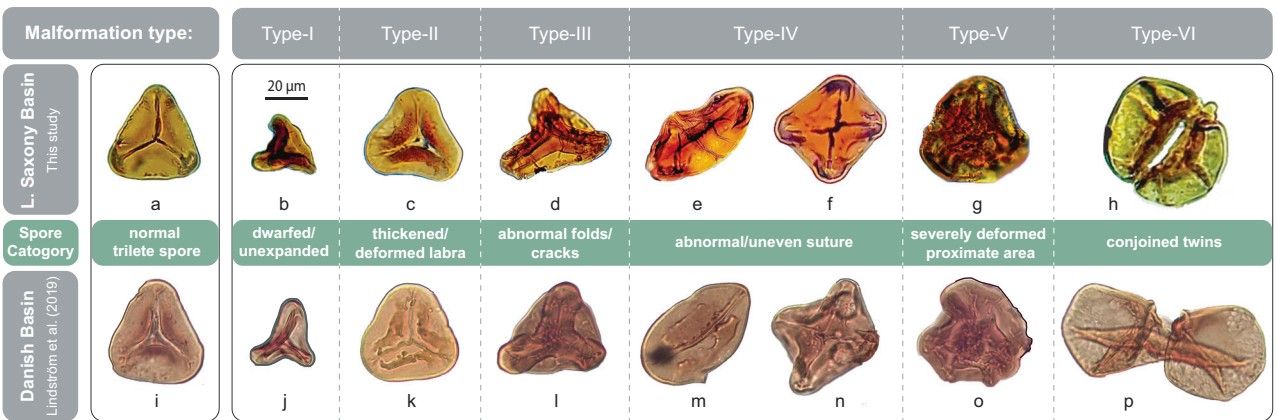

**Fig. 2 | Teratology of Late Triassic/Early Jurassic spores. a–h** Fossilized spores of pteridopsid ferns from the Lower Saxony Basin (Schandelah-1 core; this study) and **i–p** Danish Basin (Stenlille-1/4, Rødby-1 and N. Albert sections[2]). **a** and **i** Normal triangular spores derived from *Deltoidospora* spp. **b** and **j** Type-I malformations: dwarfed and unexpanded spores. **c** and **k** Type-II malformations: spores with thickened and/or deformed labra. **d** and **l** Type-III malformations: cracks and/or folds in the exine wall. **e** and **m** Type-IV malformations: elongated, oval spores with abnormal/uneven sutures showing single (monolete) laesura and **f** and **n** square spores with abnormal/uneven sutures showing multiple (quadrilete) laesurae. **g** and **o** Type-V malformations: severely deformed proximate area with no clear discernible trilete marks. **h** and **p** Type-VI malformations: attached spores conjoined by additional wall material.

## Results

### Stratigraphic framework and environmental setting

The stratigraphic framework for the Schandelah-1 core in this paper is based on ammonite[45] and palynomorph[13] biostratigraphy and cross-correlated with radiometric dates using stable carbon isotope stratigraphy[46]. Two distinctive negative organic C-isotope excursions (CIE; Fig. 3a and Supplementary Fig. 4) represent the Marshi and widely recorded Spelae CIEs (correlated with the Precursor and Initial excursions at the St. Audrie's Bay section (UK)), interpreted to reflect the volcanic injection of [13]C-depleted carbon in relation to the exogenic pool[5,46]. Temporal overlap of CAMP activity with our records from Schandelah-1 is based on the position of negative CIEs (i.e., Marshi and Spelae) and palynofloral diversity disturbances[13,45]. This is further linked to palynological records that have been correlated to U/Pb-dated ash beds and from several global locations, indicating a range of 450–150 kyr between the Marshi and Spelae CIEs[46–48]. In NW Europe, the onset of the ETME coincides with the last occurrence of the ammonite species *Choristoceras marshi*[49], which is closely associated with the Marshi CIE and a marine transgression[46]. This interval is known as the Contorta Beds in the Lower Saxony Basin. The Contorta Beds are succeeded by the Triletes Beds, which coincides with the maximum phase of marine extinction (Fig. 3)[50,51]. The Triletes Beds stand-out for their high abundance of fern spores (fern-spike interval) and a regressive event associated with relatively high organic δ[13]C values[2,13]. The onset of the Spelae CIE is also linked with a transgression[46] and is marked by increased sedimentary total organic carbon (TOC) content (Fig. 3b) and increased abundances in aquatic palynomorphs in the Schandelah-1 core[13,45]. The remainder of the Spelae CIE shows a slight increase in δ[13]C$_{TOC}$ values and widely coincides with a depauperate benthic marine interval[30]. The Lower Jurassic Hettangian (Angulatenton Fm) succession of Schandelah-1 roughly encompasses ~1.3 million years based on orbitally-paced fluctuations in δ[13]C$_{TOC}$ (Fig. 3a)[13]. This interval is dominated by shale-sandstone heterolithic facies. Repeated positive CIEs (He1–He4) in intervals of low and stable TOC values (<0.5 wt%) correlate with higher abundances of trilete spore morphotypes indicating intervals of increased palynofloral diversity disturbances[13]. Based on time series analyses, it has been shown that the long-eccentricity cycle (405 kyr) strongly influenced the climate, carbon cycle, and vegetation during the Hettangian (Fig. 3)[13]. The Hettangian–Sinemurian transition corresponds to a shift towards organic-rich claystone with overall higher TOC levels. This marked shift, combined with higher relative abundances of conifer-derived wind-driven bisaccate pollen, suggests a more distal position to the shore and a deepening of the basin. No significant biodiversity disturbances have been recorded in the Sinemurian palynoflora, indicating further stabilization of terrestrial vegetation and the regional climate, similar to pre-crisis conditions[13].

### Hg trends and anomalies

To correct for variable Hg-sequestration potential in the rock record, bulk Hg-concentrations are normalized for TOC. This is a well-known method of examining sedimentary Hg-enrichments in various marine depositional settings[52]. Although sulfur-rich and clay minerals can potentially host Hg, this is relatively uncommon, and, therefore, TOC corrections are most frequently employed[52]. The correlation between Hg and TOC for our samples is considered in Supplementary Fig. 2, and the details are discussed below. A total of five anomalous intervals of Hg/TOC are recognized within the studied section of the Schandelah-1 core (Fig. 3). Here, we identify increases of more than 1 standard deviation ($1\sigma = 64.3$ parts per billion per weight percent [ppb/wt%]) above median value (48.6 ppb/wt%) as anomalous. The Hg/TOC levels for the Marshi CIE, which showed the highest concentrations of bulk Hg (125–250 ppb), are substantially lower and fall around the median value, while the lowest measured Hg values of the Triletes Beds remain low in both records (bulk Hg = 0–10 ppb; Hg/TOC = 10–60 ppb/wt%). In contrast, the Hg-anomaly at the Spelae CIE is still prominent in the Hg/TOC record, with values ranging up to ~310 ppb/wt% (hereafter referred to as the Spelae Hg-anomaly). A strong positive linear relation exists between Hg and TOC for the Rhaetian and lowermost Hettangian strata ($R^2 = 0.61$; Supplementary Fig. 2), with the Spelae Hg-anomaly showing a notable offset. This regression is somewhat forced by two data points from the Marshi CIE, leaving a gap of data between 1.5% and 3.0% TOC. In contrast, in the Hettangian interval ($R^2 = 0.01$, Supplementary Fig. 2), there is no correlation. This is expressed by more pronounced Hg/TOC anomalies, particularly in the upper part of the Hettangian, where two increases up to ~330 ppb/wt%, similar in magnitude to the Spelae Hg-anomaly, are present. Overall, the Hettangian Hg/TOC levels plot along a median value of 48.6 ppb/wt%, while Hettangian Hg-anomalies indicate sharp increases followed by sudden decreases, similar to bulk Hg-concentrations. Another strong positive correlation between Hg and TOC is found for the Sinemurian section ($R^2 = 0.49$; Supplementary Fig. 2). The Sinemurian Hg/TOC record shows stable and low levels below median value

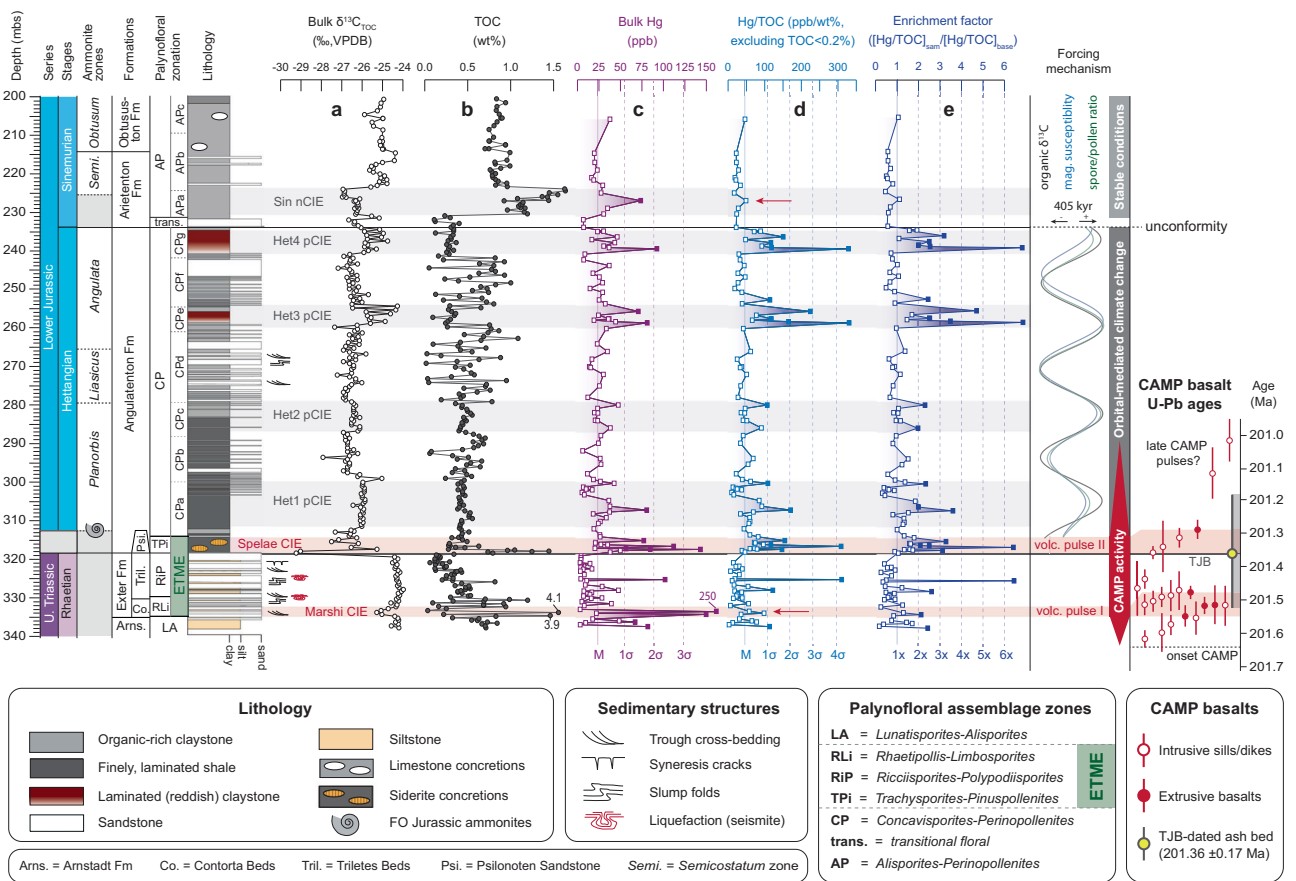

**Fig. 3 | Mercury and organic carbon geochemistry of Late Triassic/Early Jurassic in northern Germany (Schandelah-1).** Stratigraphic plots show **a** organic carbon isotopes ($\delta^{13}C_{TOC}$)[13,45], **b** total organic carbon (TOC), **c** bulk mercury (Hg) concentrations, **d** mercury over total organic carbon correction (Hg/TOC), and **e** Hg-enrichment factor for the Schandelah-1 core spanning the upper Rhaetian to lower Sinemurian. The lithostratigraphic column is present alongside key biostratigraphical information that includes ammonite[45] and palynofloral assemblage zones[13]. **c** and **d** The solid line represents the median value (M), and the dashed lines represent multiples of standard deviation (σ) of the bulk Hg and Hg/TOC normalization. Red arrows indicate the position of Hg anomalies that are not reflected in the Hg/TOC record. **e** The Hg-enrichment factors (dashed lines) are calculated relative to the Hg/TOC median value (solid line). The solid square data points represent anomalies above 1 standard deviation (**c** and **d**) or two times the

enrichment factor (**e**). Horizontal gray bars represent the position of Hettangian positive carbon isotope excursions (Het1–4 pCIEs) and sedimentary Hg-enrichments. A negative carbon isotope excursion is recognized in the Sinemurian strata (Sin nCIE). The horizontal red bars indicate the position of the Marshi and Spelae carbon isotope excursions (CIEs), closely associated with the End-Triassic Mass-Extinction. Secular variation is derived from a recent study[13] and shows spectral filters from bulk $\delta^{13}C_{TOC}$ (black), magnetic susceptibility (blue), and spore/pollen ratio (green) records. This spectral range was interpreted to represent the 405 kyr long-eccentricity cycle. Basalt U/Pb ages indicate the range of Central Atlantic Magmatic Province (CAMP) intrusive (open circles) and extrusive (solid circles) volcanism in the time domain and are correlated to the Schandelah-1 core based on the position of the Marshi and Spelae CIEs (red bars)[2,11,46].

(<48.6 ppb/wt%) and a spike (74.2 ppb) observed in the bulk Hg record. In addition, we calculated an enrichment factor for the Hg/TOC normalized record relative to a baseline value (Hg/TOC median value = 48.6 ppb/wt%) to identify anomalous intervals (Fig. 3e). We consider an enrichment factor of 2 or greater to be anomalous, which is consistent with the standard deviation assessment of the Hg/TOC record.

## Spore malformations
Malformations in trilete fern spores are observed throughout the studied section except for the Sinemurian interval (Fig. 4). These trilete, smooth triangular spores derive from several fern and tree-fern families, including the Dipteridaceae, Dicksoniaceae, and Matoniaceae[2] (Fig. 2). Abundances in spore malformations initially increase directly following the Marshi CIE to ~12% of the total spore assemblage and directly diminish towards the lowest observed fraction (<3%) in the Triletes Beds (Fig. 4b). The highest fraction of spore malformations is found across the Triassic-Jurassic transition and is contemporaneous with the Spelae Hg-anomaly, where malformed specimens comprise up to 30% of the counted spores.

Trends in malformed spore abundances follow a similar sharp increase and decrease as seen for the Spelae CIE. The most common malformation is Type-I (dwarfed/unexpanded; see Fig. 2) and is most likely related to abortion and premature shedding of non-viable spores[2]. Similar to records from the Danish Basin[2], the severity of the malformations increases and culminates during the Spelae Hg-anomaly with more frequent occurrences of Types III, IV, and V. Following the extinction interval, Hettangian malformed spore abundances show four distinct peaks coinciding with increased Hg-concentrations and positive CIEs. These reoccurring increases in spore malformations typically show average abundances of 10.2% over a background signal that averages at 3.5%, with the highest fraction observed in the upper Hettangian (16% to 18%). Type-I and Type-II (thickened/deformed labra) are most abundant in the Hettangian strata with more frequent occurrences of Type-III, IV and V malformation within intervals containing a high abundance of spore malformations (Supplementary Data 1). Spore malformation abundances decrease in the lowermost Sinemurian to 0% and have not been detected in the upper interval of the studied section. Spore malformations typically show no features indicating reworking (i.e.,

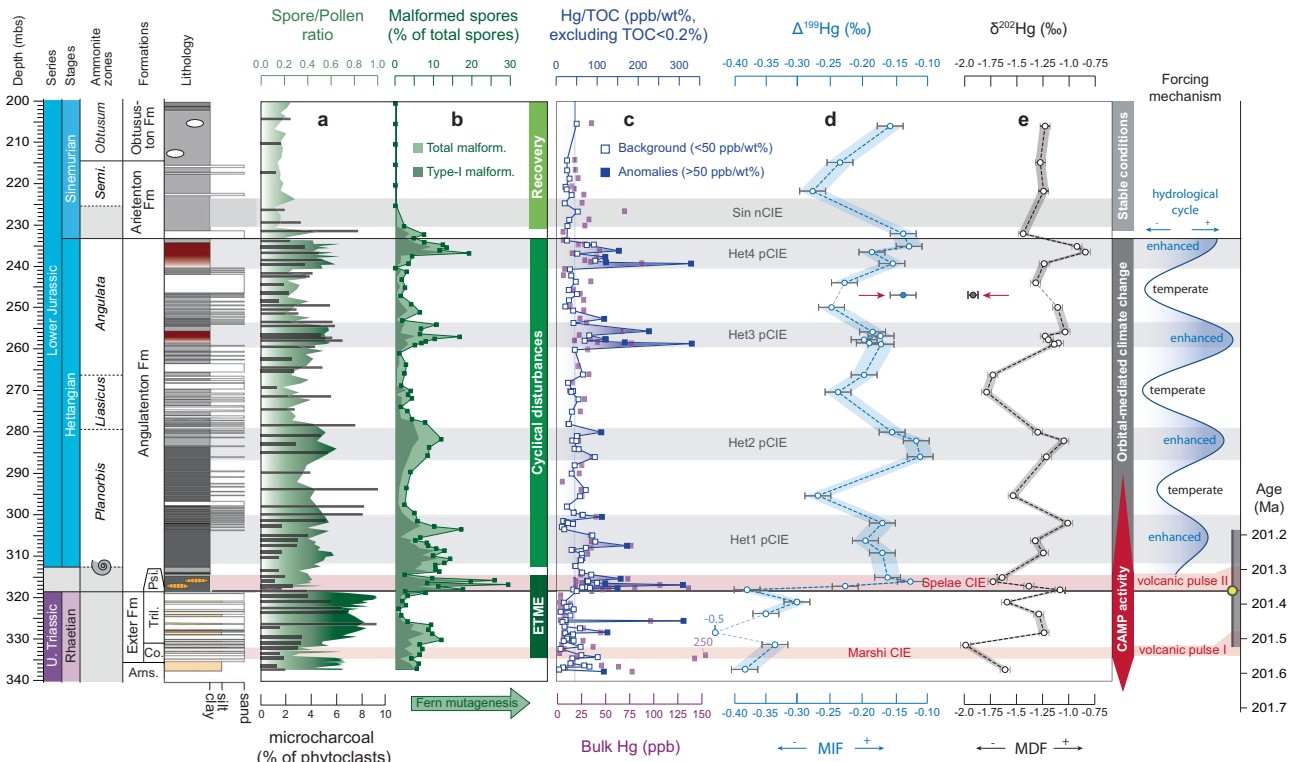

**Fig. 4 | Stratigraphic profile of mercury isotope and terrestrial palynology records in northern Germany (Schandelah-1).** Stratigraphic plots show **a** spore/pollen (S/P) ratio with microcharcoal abundances, **b** malformed spore abundances, **c** mercury over total organic carbon correction (Hg/TOC), **d** $\Delta^{199}$Hg (MIF = mass-dependent fractionation), **e** $\delta^{202}$Hg (MDF = mass-dependent fractionation). **a** spore/pollen ratio is overlain with the relative abundance of microcharcoal. Both these records are derived from palynological slide counts[13]. **b** The relative abundance of malformed (abnormally formed) spores is calculated as a fraction of total spores in the assemblage. Dark green area indicates the relative contribution of spore type-I (Fig. 2; dwarfed/unexpanded malformations). **c** Open squares represent Hg/TOC values below 1 standard deviation,

while solid squares depict anomalous samples above 1 standard deviation (Fig. 3). The shaded blue bar represents background values below 1 standard deviation. **d** and **e** Hg-isotope measurements are depicted with individual error bars representing standard deviation ($2\sigma$; $\Delta^{199}$Hg = 0.04‰ and $\delta^{202}$Hg = 0.08‰). Solid circles represent a single measurement (246.5 mbs) inconsistent with general pattern and is classified a potential outlier. Secular variation in the hydrological cycle is derived from variation in the magnetic susceptibility record of Schandelah-1 (see Fig. 3)[13]. Age estimates for the Marshi and Spelae carbon isotope excursions (CIEs) are based on correlations of astronomical-tuned C-isotope records to basalt U/Pb dated sections (see Fig. 3)[2,11,46].

darkened/broken wall material), and therefore, we are confident this record represents an in situ signal (see the "Methods" section for palynological counting strategy).

## Hg-isotopes

Hg-isotopic records from Schandelah-1 are presented using the $\Delta^{199}$Hg and $\delta^{202}$Hg notation, representing mass-independent (MIF) and mass-dependent fractionation (MDF), respectively (Fig. 4d and e). $\Delta^{199}$Hg shows highly negative values for the Rhaetian interval (−0.50‰ to −0.31‰) with an average of −0.38‰. Following a sharp positive excursion (−0.39‰ to −0.13‰) at the Spelae CIE, average $\Delta^{199}$Hg values shift to −0.18‰ for the Hettangian section. Furthermore, $\Delta^{199}$Hg values in the Hettangian section show repetitive positive shifts from an average baseline of −0.22‰ up to values of −0.17‰, coinciding with anomalies in Hg concentrations. The main feature in the Sinemurian seems to be a broad negative shift in $\Delta^{199}$Hg (average value = −0.23‰).

Rhaetian $\delta^{202}$Hg values average at −1.55‰ with relatively low values in the pre-extinction interval (−1.99‰ to −1.61‰). Higher values (−1.60‰ to −1.01‰) characterize the Triletes Beds, which directly underly a negative excursion from −1.01‰ to −1.73‰ at the Spelae CIE/anomaly. Hettangian $\delta^{202}$Hg varies (−1.79‰ to −0.84‰) in tandem with $\Delta^{199}$Hg, with four positive shifts coinciding with anomalies in Hg concentrations, except for one outlier (246.5 mbs), which displays opposite excursions for MDF and MIF.

## Discussion

In the Schandelah-1 record, the abundance of malformations of fern spores across the TJB varies in concert with the Spelae CIE and sedimentary Hg-enrichments in the Schandelah-1 record (Fig. 4). The spore malformation types and abundances of the Schandelah-1 record are strikingly similar to the Danish Basin[2], suggesting a synchronous widespread mutagenic event at the TJB with a single underlying cause. Multiple mechanisms have been proposed to explain malformations in pollen and spores associated with the emplacement of large igneous provinces and mass-extinctions. Ozone layer depletion due to volcanic halocarbon emissions may have led to increased UV-B radiation and has been previously linked to malformations in gymnosperm pollen and permanent lycopsid tetrads across the Permian-Triassic boundary[21–23]. Other explanations include rising global temperature due to increased greenhouse gas emissions causing heat-stress, which can induce polyploidy, abnormal meiosis, and cytokinesis in extant angiosperm pollen[24]. Although increased UV-B radiation and heat-stress could have played a contributing role, these scenarios fall short of explaining the prolonged and repeated mutagenesis in fern communities as seen during the Hettangian of the Schandelah-1 core after major CAMP activity had ceased (Fig. 4). U/Pb dates of CAMP basalts indicate two phases of pulsed flood volcanism that ended following the Spelae CIE[47,53], although with a few smaller outflows in the Hook Mountain Group from Morocco and New Jersey known to be later[1]. Only minor sill intrusions of limited size are coeval with the Planorbis

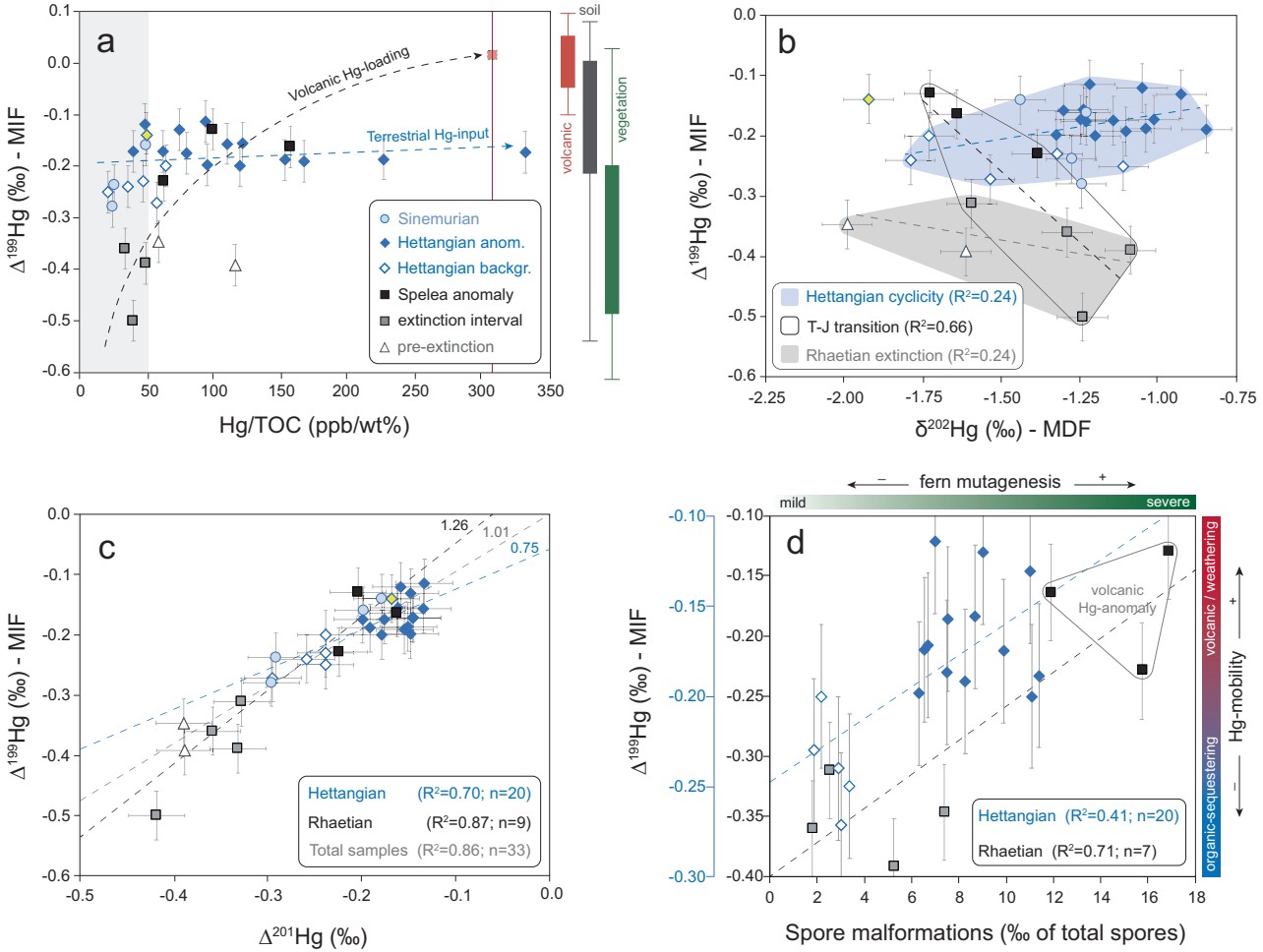

**Fig. 5 | Mercury isotopic signature variation across the Triassic–Jurassic transition.** Cross-plots depicting $\Delta^{199}Hg$ (MIF = mass-independent fractionation) plotted against **a** mercury over total organic carbon correction (Hg/TOC), **b** $\delta^{202}Hg$ (MDF = mass-dependent fractionation), **c** $\Delta^{201}Hg$ and **d** spore malformations for the Schandelah-1 core. **a** Pathways for Hg-enrichment in relation to $\Delta^{199}Hg$ (MIF). The shaded gray bars represent background Hg concentrations (<50 ppb/wt%). The red square indicates the position of the volcanic endmember, based on the highest measured Hg/TOC (vertical red line = 309.36 ppb/wt%) and zero MIF ($\Delta^{199}Hg$ = -0‰) of volcanogenic Hg. The black dashed arrow indicates the predicted path of volcanic Hg-enrichment based on volcanic-related intervals (squares) and the estimated volcanic endmember. The blue dashed arrow indicates the path of terrestrial Hg-enrichment. The position of the terrestrial-input endmember is assumed to be the highest value for Hg/TOC in the Hettangian interval and $\Delta^{199}Hg$ values similar to the background (-−0.1‰ to −0.3‰). The yellow diamond represents an outlier data point. The range of Hg-MIF reservoirs is depicted in boxplots. **b** Relative variation of $\Delta^{199}Hg$ (MIF) versus $\delta^{202}Hg$ (MDF) for three separate intervals (excluding the Sinemurian measurements) with the slopes of the respective linear relations depicted with the dashed lines. **c** Three $\Delta^{199}Hg/\Delta^{201}Hg$ ratios based on total, Rhaetian, and Hettangian intervals (dashed lines) showing variation in mass-independent fractionation (MIF). **d** The relation of $\Delta^{199}Hg$ and the corresponding relative abundances of spore malformations. Average spore malformations were calculated using three samples closest to the Hg-isotope measurements if sample rates did not match. Error bars represent standard deviation (2$\sigma$).

zone (early Hettangian)[1], although erosion might have reduced the volume. Therefore, the effect on ozone layer degradation was also likely limited, eliminating UV-B radiation as multi-million-year stressor. In contrast, Hg is continuously present in the Earth's crust and can be mobilized through gaseous volatilization, transported in particulates (rivers/runoff), and organic matter degradation that involves several surface processes, such as deforestation, weathering, and wildfires[54]. The strong correlation between Hg-concentrations and elevated malformed spore abundances throughout the studied section (Supplementary Fig. 3) suggests that the accumulation of mobilized mercury in terrestrial environments played a key role.

The enrichment of Hg in sediments is governed by several processes that drive burial and preservation on geological timescales. Most Hg is adsorbed to organic matter (OM) and, to a lesser degree, on S-rich minerals, a process that takes place mostly in aquatic systems[55]. The size of the carbon sink normally dictates the adsorption potential and concentration of Hg in sediments[30]. This is clearly observed for the Marshi CIE, where the highest Hg-concentrations are correlated with high TOC levels. In NW Europe this event is linked to widespread marine transgression[13] driving increased carbon burial. A decoupling of Hg and TOC is observed at the Spelae Hg-anomaly (Supplementary Fig. 2), suggesting carbon burial did not play a significant role in Hg-enrichment. The pathway of volcanic Hg-enrichment in marine sediments is largely facilitated through atmospheric deposition of oxidized $Hg^{2+}$ and subsequent scavenging by organic matter (OM). In addition, other minerals such as sulfides and clays can be hosts of Hg[36,56]. For instance, during photic zone euxinia (PZE) in the upper water column, the amount of free $H_2S$ can result in in situ pyrite framboid formation. These conditions favor Hg-mitigation through S-drawdown in the case of excess Hg, overriding the OM-drawdown[34]. For the Schandelah-1 record, a short-lived peak in TOC and the influx of aquatic palynomorphs[13] at the onset of the Spelae CIE is indicative of a transgression. Molybdenum isotope data from Schandelah-1 reveal localized euxinic conditions at this level[16]. Therefore, the initial rise in

bulk Hg-concentrations at the Spelae CIE (318.5–317.7 mbs) is likely coupled to increased sulfur-binding and burial, while the subsequent decrease in TOC and increase in Hg/TOC reflects excess burial of, likely volcanogenic, Hg.

Although CAMP-eruptions spanned at least 800 kyr, many global sites only record a few prominent Hg-anomalies (Supplementary Fig. 4), most notably at the Spelae CIE[2,28,30,31,57]. Some contemporaneous T–J boundary sections record a notable increase in bulk sedimentary Hg at the Marshi CIE (Supplementary Fig. 4), similar to the Schandelah-1 record. However, this interval is often not expressed by a Hg/TOC anomaly. There are a few exceptions, however, suggesting a potential volcanic loading of atmospheric Hg at the start of the ETME. Mediation of enriched environmental Hg through carbon burial and/or sulfur adsorption could be responsible for the variable expression from site to site. Furthermore, the generation and emission of volatiles through volcanic intrusions is dependent on the composition of intruded sedimentary rock[3]. It has been suggested that earlier CAMP phases associated with the Marshi CIE produced more halocarbons from carbonate and evaporite-rich deposits in the Amazon Basin[53]. While the CAMP phase associated with the Spelae CIE potentially intruded organic-rich sediments that had a higher potential to generate gaseous Hg[3,53]. Therefore, the magnitude of a sedimentary Hg-anomaly is dependent on a combination of burial, preservation, and generation of Hg in the environment.

Similar to the Spelae Hg-anomaly, the four Hettangian intervals with Hg-anomalies show a decoupling of Hg to TOC. The regular reoccurring nature of these Hg-enrichments and the assumed absence of eruptive CAMP activity (discussed above) suggest they are unlikely to have been caused by volcanism. Instead, terrestrial reservoirs (bedrock, soil, and vegetation) need to be considered for their ability to accumulate Hg, which can be intermittently delivered to shallow marine depositional environments. In addition, the effects of redox conditions and early diagenesis need to be taken into account[58]. We utilize the Hydrogen Index (HI) and Oxygen Index (OI) of organic matter[13] to assess the influence of (post-depositional) oxidation on our Hettangian Hg record. Hg/TOC shows no significant correlation with either HI or OI (Supplementary Fig. 2), although three samples with exceptionally high Hg/TOC values (>200 ppb/wt%) exhibit lower HI in the upper Hettangian red claystone intervals. This could indicate the presence of paleo-redox fronts in the sediment that are characterized by lower HI and higher OI in open marine systems, expressing themselves through sharp increases of Hg/TOC[58]. However, the organic matter from the Schandelah-1 section is thermally immature and dominated by terrestrial input[13], where intervals showing elevated Hg/TOC coincide with pulses of increased weathering, as evident from reworked palynomorphs[13]. Malformed spore specimens, however, show no features of being reworked. Overall, this indicates that particle-bound Hg of the Hettangian anomalies was delivered via increased transport of terrestrial sediments (clays) and/or terrestrial organic material (high O/C and low H/C).

Large-scale volcanic activity increases the global Hg budget, resulting in Hg accumulating in terrestrial environments[32,59]. Terrestrial reservoirs, such as plants and soils, can accumulate Hg and act as a source to shallow marine/lacustrine environments when Hg is remobilized[35,36,59]. The residence time of Hg determines the redeposition potential, which is closely tied to soil organics in terrestrial environments. About 50% of Hg deposited on the ocean's surface is re-emitted to the atmosphere, while only a small fraction (10%) of Hg deposited in soils is recycled[60]. This results in a residence time of Hg in soils of about 1000 years with a total residence time of 3000 years in the atmospheric–ocean–terrestrial system[60]. However, bedrock reservoirs, such as coal beds and mineral sources, have accumulated Hg over longer geological timescales that can be similarly mobilized by erosion and runoff[61]. Part of this terrestrial Hg finds its way to the marine realm via rivers, which may contribute up to 10% of the total oceanic Hg input[62]. Thus, the terrestrial Hg flux to shallow marine basins is significantly larger than the total oceanic input. However, deriving terrestrial Hg-concentrations from marine sediments, such as from the Schandelah-1 record, remains challenging. Periods of increased runoff and erosion would ultimately displace large quantities of terrestrially stored Hg from vast catchment areas and concentrate it in low-lying basins and deltaic/coastal fronts, such as the Central European Basin (CEB; Fig. 1), and ultimately cause elevated marine sedimentary Hg-concentrations in those areas.

In the Rhaetian interval of Schandelah-1, sources of Hg are consistent with plant material where MIF values show highly negative variability ($\Delta^{199}$Hg, −0.50‰ to −0.30‰). Mass-dependent fraction shows considerably more variation ($\delta^{202}$Hg, −2.00‰ to −1.00‰) during the Rhaetian (Fig. 5b). Positive shifts in MDF in the Triletes Beds preceding the Spelae CIE were likely driven by the reduction of $Hg^{2+}$ via microbial or abiotic processes[42]. In addition, the low Hg concentrations are indicative of the absence of Hg-input. The volcanic origin of the Spelae Hg-anomaly in the Schandelah-1 record is supported by the sharp positive shift in MIF co-occurring with a negative shift in $\delta^{13}C_{TOC}$ (Fig. 4). An increase in atmospheric dispersal of volcanic $Hg^0$ (assumed to be near-zero MIF) fits the observed pattern, causing the $\Delta^{199}$Hg values of in those beds enriched in Hg to shift towards 0‰ (Fig. 5a, black dashed arrow). Additionally, magmatic sill intrusions into coalbeds provide volcanism with another potential Hg-source. Hg-isotope studies of coal-fired plant emissions demonstrate slightly positive to slightly negative MIF and substantial negative MDF[63], although this varies slightly depending on the particular coal deposit. Volcanic Hg-enrichment at the Spelae Hg-anomaly displays a notable shift towards more negative MDF, potentially indicative of subsurface coal-burning and/or plant influence (Fig. 5b). A similar pattern was noted in the St. Audrie's Bay section (Fig. 1), showing a positive shift in MIF and a negative shift in MDF at the Spelae (initial) CIE[31].

In the Hettangian interval of Schandelah-1, higher MIF values (Fig. 4d) mark a shift in Hg-sourcing and/or a change in photochemical reduction. The ratio of $\Delta^{199}$Hg/$\Delta^{201}$Hg depicts a slope of -1.00 (Fig. 5c) that indicates MIF variability is consistent with the photochemical reduction of $Hg^{2+}$, deriving from rainfall, foliage, sediments, and coals[42]. Both $\Delta^{199}$Hg and $\delta^{202}$Hg values exhibit repeated positive shifts in tandem with Hg-anomalies, which suggest that increased input of Hg was either derived from a mixture of multiple terrestrial sources and/or isotopic alteration during transport[44]. For instance, modern soil reservoirs typically show higher MIF ($\Delta^{199}$Hg = −0.2‰ to +0.1‰) and likely represent a contribution of both leaf foliage (litter fall, highly negative MIF) and mineral/bedrock sources (near-zero MIF)[41,54]. Modern foliage also tends to show lower $\delta^{202}$Hg values than those of mineral soils[43,64]. Other studies suggest significant shifts in MIF can be attributed to a higher proportion of reworking from abundant Triassic coal beds[57,61]. Fossilized terrestrial organic matter derived from coal deposits shows low MIF (average $\Delta^{199}$Hg = ~−0.2‰)[54]. This suggests that the terrestrial Hg-isotope signature in the marine system is dependent on the composition of the transported material. While during the Rhaetian, the MIF composition of Schandelah-1 represents a vegetation endmember (low Hg/TOC and highly negative MIF), during the Hettangian MIF could represent a mixture of the plant (litterfall) and soil/bedrock (mineral) derived Hg.

During the Hettangian, loss of vegetation cover, as evidenced by decreased abundances of conifer-type vegetation[13], would increase the fraction of Hg-mineral sources in the shallow marine environment. Furthermore, elevated abundances of reworked Upper Triassic palynomorphs (up to 5% of total assemblage) suggest periodic increases in weathering[13], indicating the redeposition of older coal/bedrock-derived Hg. This pattern is also evident in sections in China that straddle the Triassic-Jurassic boundary, confirming a prominent role for terrestrial Hg during the early Hettangian through increased weathering[32]. Previous examinations of the Schandelah-1 core have

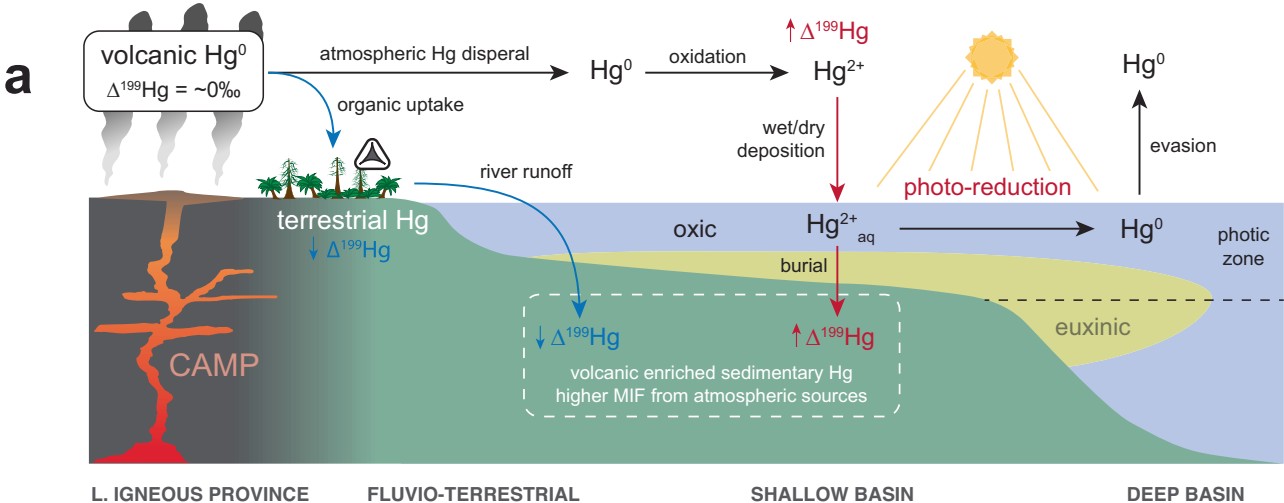

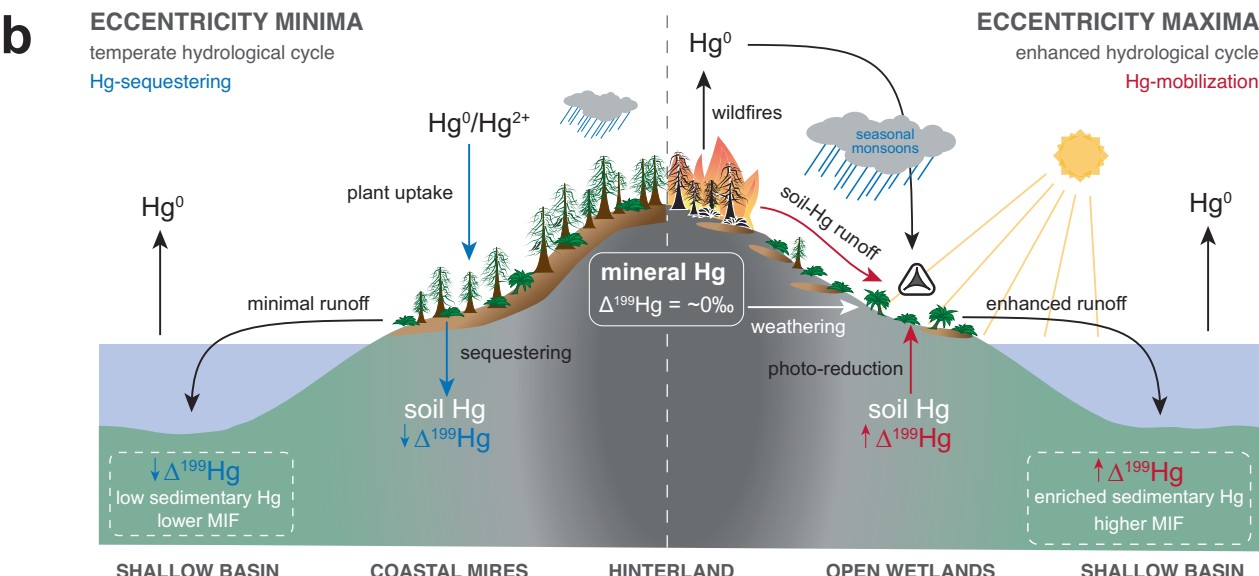

**Fig. 6 | Environmental model illustrating mechanisms of Hg-mobility and isotope fractionation in the Central European Basin.** This conceptual model illustrates the major terrestrial disturbance forces in the marginal fluvio-lacustrine environment and the resulting depositional changes in the shallow marine basin. Scenario **a** shows periods of explosive eruptions of the Central Atlantic Magmatic Province (CAMP) with high Hg-input, associated with the Spelae carbon isotope excursion (CIE) and Triassic–Jurassic transition. The intervals during the Hettangian

**b** are dominated by orbital variation. Here, the scenarios of eccentricity minima (left) and maxima (right) show the two extremes in orbital forcing and disturbance[13]. The white triangle (spore) indicates the conditions for the formation of spore malformations. Blue arrows indicate environmental Hg pathways/processes that result in negative shifts in mass-independent fractionation ($\Delta^{199}$Hg), while red arrows indicate processes resulting in positive shifts.

revealed simultaneous increases in the S/P ratio ($\Sigma$spores/($\Sigma$pollen + $\Sigma$spores)) and relative abundance of microcharcoal particles[13], which coincide with intervals of elevated spore malformations (Fig. 4). Changes in wildfire frequency and weathering intensity/runoff due to major swings in the hydrological regime were modulated by the long-eccentricity cycle (405 kyr) and were likely exacerbated by extreme greenhouse conditions[13]. During eccentricity maxima, shifts to open landscapes resulted in large inputs of terrestrial material and soil/bedrock-stored Hg, driving positive shifts in both MIF and MDF. In addition, emissions from wildfires could have played a contributing role in the mobilization and distribution of gaseous Hg. The effect of wildfire Hg-emissions on MIF variability has been demonstrated to be non-significant[65] and, therefore, did not influence the overall signature.

However, open and flooded landscapes would provide the ideal setting for terrestrial (soil, bedrock, and vegetation) Hg to be photochemically reduced, resulting in shifts to gaseous Hg⁰ species. Photochemical reduction of terrestrial Hg, which yields higher MIF (higher $\Delta^{199}$Hg values), shows a correlation with shading from canopy[66]. Sediments from salt marshes also show highly positive MIF, which has been attributed to in situ photochemical reduction[67]. Hence, shifts towards open landscapes through the loss of high canopy vegetation affect the mobility of Hg, not only by making soils and the underlying bedrock more susceptible to erosion but also by increasing exposure to sunlight, causing volatilization/degassing of terrestrial Hg.

The Spelae Hg-anomaly interval of Schandelah-1 shows evidence of increased extinction rates and increased stress/mutagenesis in fern

communities (i.e., spore malformations). Hg-enrichments driven by volcanic emissions coincide with the Spelae CIE reflecting gaseous Hg being distributed atmospherically as noted by a positive shift in MIF (Fig. 6a). This suggests Hg-toxicity is dependent not only on the concentration but also on the speciation (i.e., gaseous phase). Although other toxic metals are emitted from volcanic events and could potentially impose plant mutagenesis, the volatile nature of Hg results in uptake in plants via stomata[41] and increases its re-deposition potential[60]. The similarities in morphological traits of malformed spores found throughout the Schandelah-1 section suggest a single underlying cause. This is in line with gaseous Hg as the main mutagenic stressor in fern communities across the ETME and Early Jurassic Hettangian.

During the Hettangian stage, wildfire emissions and photochemical reduction of terrestrially sourced Hg led to periodical increases in volatilization of $Hg^0$. This could have induced mutagenesis in local fern communities comparable to volcanic atmospheric deposition at the TJB (Fig. 5d). Coastal mires and hinterland areas acted as catchments and storage for Hg during times of increased terrestrial biomass production (Fig. 6b). Temperate conditions during long-eccentricity (405 kyr) minima promoted high canopy mire vegetation and Hg-sequestering. Increasing seasonal (precessional) contrast during long-eccentricity maxima caused a collapse of vegetation cover and promoted the proliferation of pioneering fern taxa that were subjected to Hg-mobilization and degassing. Transport of terrestrial organic and eroded bedrock from vast catchment areas likely resulted in higher concentrations of Hg in low-lying coastal margins and shallow basins (Fig. 6b). Furthermore, a collapse of terrestrial and marine biomass due to the ETME likely impeded the re-absorption of excess Hg[68] and caused long-term consequences. The mutagenic potency of Hg on vegetation would be significantly magnified if shifted to its gaseous form ($Hg^0$), causing stomatal uptake and circumventing root-protective systems[25–27,41]. Palynofloral diversity disturbances appear to have ceased during the early Sinemurian when a conifer-dominated biome stabilized, and sea level rise led to enhanced carbon burial[13]. Despite the low Hg-record resolution of the Sinemurian in the Schandelah-1 record, enhanced carbon burial and widespread forest biomes may have led to the mediation of Hg-pollution and long-term sequestering in soils and marine sediments.

Based on quantification of spore teratology and Hg concentrations, as well as Hg-isotope records, we establish a link between fern mutagenesis and Hg-pollution due to CAMP volcanism and subsequent mobilization from soil/bedrock reservoirs during times without large-scale volcanic eruptions. Hg-mobilization continued during the Hettangian under extreme greenhouse climate conditions. Evidence suggests that Hg was mainly sourced through continental erosion during the Hettangian based on positive shifts in $\delta^{202}Hg$ and $\Delta^{199}Hg$ (MDF and MIF). In contrast, Hg at the Triassic-Jurassic boundary was volcanically sourced consistent with other studies. Climate-driven collapse of vegetation through eccentricity-paced increases in wildfire activity and weathering impeded the re-absorption of Hg, which continued to disturb and stress Hettangian coastal ecosystems. The strong correlation of spore malformations with high MIF/MDF values further indicates that the photo-reduction of terrestrial Hg was periodically enhanced during the Hettangian in open coastal/wetland areas due to the loss of canopy cover. Our results further indicate that gaseous Hg played a significant role in mobility and toxicity, which directly impacted fern communities. Hg pollution may have been especially severe in coastal regions depending on the ability of such areas to absorb and store Hg for longer periods. Although environmental Hg-dynamics over hundreds of thousands of years are still unclear, our results point to the long-term implications of large-scale volcanism on terrestrial vegetation following major extinction.

## Methods

### Palynology and spore teratology

A total of 91 samples were analyzed for spore teratology and were prepared and processed using the palynological methods at Utrecht University. Approximately 5–7 g of oven-dried material was crushed and processed once with 10% hydrochloric acid (HCl) and twice with 38% hydrofluoric acid (HF). Residual material was sieved using a 10 µm nylon-mesh, homogenized and mounted on glass slides using glycerin gel. A smaller number of slides were permanently mounted using a combination of 5% polyvinyl alcohol (PVA) solution and glass glue. To assess the relative abundance of spore malformations, all normal and malformed spores were counted until a total of 300 palynomorphs were recognized using a Leica DM 2500 transmitted light microscope (×40 magnification).

Specimens exhibiting clear malformed characteristics were counted as malformed. Folded, broken, or obscured specimens were omitted from malformed categories and instead were counted as normal. This could result in the obtained fraction of relative aberrancy being an underestimation in some samples. In addition, only a few malformed spore specimens were found that exhibited reworked features such as darkened and broken wall material and are similarly omitted form the malformation counts. We completely focused on a single sporomorph group known as LTT-spores (laevigate, triangular, trilete spores)[2], which is known to have been produced by ferns of the Dipteridaceae, Dicksoniaceae, or Matoniaceae[69] during the Late Triassic and Early Jurassic in NW Europe[2]. These fern families can be assigned to several different spore taxa, most notably to the genera *Concavisporites* and *Deltoidospora*. In addition, we pinpointed intervals of palynofloral disturbance based on the spore/pollen (S/P) ratio, which is calculated as the total number of spores over the sum of the total number of pollen and spores ($\Sigma$spores/($\Sigma$pollen + $\Sigma$spores)) for each counted sample. Reworked palynomorphs are counted separately and typically exhibit darkened and broken wall material. All palynological data is summarized in Supplementary Data 1.

Various forms of environmental stress may reflect different types of disturbance during spore formation. Similar to the methods described in a previous study[2], we focused on morphological traits related to function and viability rather than taxonomy on the species level. Classification in this study was organized based on the rarity of all observed malformations, with type-I being the most common and type-VI the rarest. *Type-I malformations*: Dwarfed and unexpanded spores are considered to be the result of premature shedding from sporangia. Unexpanded forms were likely not mature and, therefore, non-viable. *Type-II malformations*: Spores with thickened and/or deformed labra, sometimes exhibiting uneven trilete rays (Supplementary Fig. 1i), may represent immature spore tetrads that have not completely separated. *Type-III malformations*: Cracks and/or folds in the exine wall are features that often co-occur with type-II malformation within the same specimens (Fig. 2d and l; Supplementary Fig. 1l). These abnormalities are likely related to later stages of improper exine wall development. *Type-IV malformations*: Abnormal/uneven suture showing single (monolete) marks and multiple (quadrilete) marks could represent unbalanced meiosis and cytokinesis, indicating a malfunctioning of the mother cell. *Type-V malformations*: Spores with severely deformed proximate areas with no clear discernible trilete marks were likely non-viable and part of deformed spore tetrads due to genetic disturbance. *Type-VI malformations*: Spores conjoined by additional wall material were only encountered twice in the studied section and could be the result of improper development of the spore tetrad (unbalanced meiosis).

### Organic carbon analysis

Organic carbon records were originally reported in several previous studies[13,45] (Supplementary Data 2). Powdered samples (-0.3 g) were

analyzed for carbon content using a CNS analyzer (NA 1500) at Utrecht University. The detection limit for this analysis is at -0.2%. All samples that were below this value were omitted from the Hg/TOC correction. Prior to analysis, samples were treated twice using 10% HCl and rinsed with de-ionized water for the removal of carbonates. The total organic carbon (TOC) content was calculated by multiplying the measured carbon content with a ratio of the de-carbonated and original sample weights. Hydrogen and Oxygen indices are derived from a previous study of Schandelah-1[13] (Supplementary Data 2).

### Mercury analysis

Prior to analysis, a total of 129 individual freeze-dried sediment samples were crushed using an agate mortar and pestle and further ground to a fine homogenized powder. Bulk mercury (Hg) concentrations were determined on a Hydra IIC direct mercury analyzer using thermal decomposition, amalgamation, and atomic absorption spectrophotometry following EPA Method 747342 at Teledyne Leeman Labs (Hudson, NH, USA). A total of 55 samples were weighed (10–50 mg) into quartz boats and quantified by the external standard method. Calibration curves range from 1 to 1500 μg/g. A total of 2–3 certified reference materials (CRMs: MESS-3, DOLT-5, NIST2709a, and PACS-3 [National Research Council of Canada and NIST]) were introduced in every batch of 10 samples along with 3 blanks and 1–2 pairs of duplicates. The mean CRM recoveries were within the expected ranges (101.9 ± 9.0%). An additional 55 samples were analyzed with a Lumex 915+ device combined with a pyrolysis unit (PYRO-915) at the University of Oxford (UK). Sediment samples were pyrolyzed at 700 °C and calibrated using a paint-contaminated soil standard (NIST2587, 290 ppb [ng/g] Hg). A duplicate was introduced for every 10 samples. Long-term observations of the NIST2587 standard established a reproducibility of, on average, 6%[58]. Lastly, additional samples that were selected for Hg-isotope measurements were also analyzed for bulk Hg concentrations at the School of Earth System Science (Tianjin University), as described in the following section. All bulk Hg data is summarized in supplementary Data 3.

### Mercury isotope analysis

Mercury isotopes were analyzed using multi-collector inductively coupled plasma mass spectrometry (MC-ICP-MS, Neptune Plus, Thermo Scientific) at the School of Earth System Science, Tianjin University, based on published methods[70,71]. Prior to isotopic analysis, Hg in 33 samples was extracted by acid digestion and then purified using an ion-exchange chromatographic method[71]. Briefly, powdered samples were weighed into 30 ml Teflon beakers and digested with a mixture of trace metal grade concentrated acids containing $HNO_3$, HCl, and BrCl with a volume ratio of 12:6:1 at -100 °C for 48 h. Then the digested samples were centrifuged to remove solid residues. The centrifuged solutions were loaded onto columns containing anion exchange resin AG1-X4 (200–400 mesh, Bio-Rad). After rinsing with 2 M HCl to remove the matrix, Hg was eluted with 12 ml 0.5 M $HNO_3$ + 0.05% L-cysteine and then digested with 0.2 M BrCl prior to isotope analysis. The Hg concentrations in the eluted solutions were determined using a Lumex RA-915 M Hg Analyzer. The Hg yield of the digestion and anion-exchange chromatographic procedures was 101 ± 14% (2 SD, $n = 34$). Procedural blanks and one standard reference material (SRM), GBW07311 (Stream sediment), and the NIST SRM 3133 Hg isotope standard were processed alongside samples. Recoveries for all SRM were 103 ± 8% (2 SD, $n = 6$) (Supplementary Data 5). Mercury concentrations in procedural blanks were typically <1% of the Hg present in samples.

The matrix-separated samples were diluted to 1.0–1.5 ng/g of Hg using a 5% HCl solution. Then, Hg in the diluted solutions was reduced by $SnCl_2$ (3%, w/v) to gaseous $Hg^0$, which was then carried into the plasma of MC-ICP-MS by Hg-free Ar. Simultaneously, thallium (Tl) aerosol (NIST SRM 997) was generated by the Aridus II desolvator and was introduced together with $Hg^0$ vapor into the plasma. Five Hg isotopes ($^{198}Hg$, $^{199}Hg$, $^{200}Hg$, $^{201}Hg$, and $^{202}Hg$) and two Tl isotopes ($^{203}Tl$, $^{205}Tl$) were simultaneously measured via Faraday cups. Instrumental mass bias was corrected by using a combination of internal calibration with measured $^{205}Tl/^{203}Tl$ ratios and standard–sample–standard bracketing (relative to the NIST SRM 3133 Hg standard). The bracketing standard was matched to samples in terms of both matrix and Hg concentration (<10% difference). On-peak zero corrections were applied to all measured masses. Mercury isotope compositions are reported using $\delta$ notation defined by the following equation:

$$\delta^x Hg(‰) = \left[ \frac{(^x Hg/^{198}Hg)_{sample}}{(^x Hg/^{198}Hg)_{std}} - 1 \right] \times 1000 \qquad (1)$$

where $^x Hg$ is $^{199}Hg$, $^{200}Hg$, $^{201}Hg$, or $^{202}Hg$, and "std" represents the NIST SRM 3133 standard. The MDF is reported as $\delta^{202}Hg$, and MIF is reported as the capital delta notation ($\Delta$) according to the following equation:

$$\Delta^x Hg(‰) = \delta^x Hg - (\delta^{202}Hg \times \beta) \qquad (2)$$

where $x$ is the mass number of Hg isotopes 199, 200, and 201. $\beta$ is a scaling constant used to estimate the theoretical kinetic MDF, and it is 0.2520, 0.5024, and 0.7520 for $^{199}Hg$, $^{200}Hg$, and $^{201}Hg$, respectively[72].

To ensure data quality, each sample was measured at least twice, and a commonly used reference standard, NIST SRM 8610, was measured for every 6–7 samples to monitor instrument performance. The averages of all NIST 8610 are: $\delta^{202}Hg = -0.57 ± 0.07‰$, $\Delta^{199}Hg = -0.04 ± 0.06‰$, $\Delta^{200}Hg = 0.01 ± 0.02‰$ (2 SD, $n = 20$), consistent with the published values[72]. The GBW07311 yielded average $\delta^{202}Hg$, $\Delta^{199}Hg$ and $\Delta^{200}Hg$ values of $-0.54 ± 0.05‰$, $-0.27 ± 0.04‰$ and $-0.02 ± 0.07‰$ ($n = 2$, 2SE), respectively (Supplementary Data 4), which are also consistent with the published values[70]. All isotope data are reported in Supplementary Data 4, and analytical uncertainties are reported as either 2 standard error (2SE) of sample replicates or 2 SD of all measurements of the NIST 8610, whichever is higher.

## Data availability

All data generated in this study are provided in the Source Data files. Additional data needs to be requested from the corresponding authors.

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

## Acknowledgements

We are grateful to the staff, crew, and sponsors of the Schandelah Scientific Drilling Project for their efforts in material recovery and for providing us with the material and data studied in this report. We are especially grateful for assistance in interpreting the data and the environmental implications of mercury cycling by J.R and T.N. This research was funded through the Dutch Research Council open program awarded to B.v.d.S. (NWO ALWOP.623). In addition, the Hg-isotope analysis was conducted at Tianjin University (China) and was funded by the National Natural Science Foundation of China and awarded to W. Z. (grant No. 41973009). T.A.M. and I.F. are supported by ERC (Consolidator Grant V-ECHO: ERC-2018-COG-818717-V-ECHO).

## Author contributions

This project was based on a study by S.L. and expanded upon by R.B. and B.v.d.S. with input from S.L. and H.S. Palynological and teratology analysis was primarily conducted by R.B. with contributions from I.W. and input from B.v.d.S. and S.L. Bulk mercury analysis was conducted by R.B. with contributions from H.S., T.A.M. and I.M.F. Mercury isotope analysis was conducted by W.Z. with assistance from Y.W. Interpretation of mercury data was assisted by J.R., T.N. and A.S. This manuscript was developed by R.B. and W.Z., S.L., H.S., I.W., I.M.F., T.A.M., Y.W., J.R, T.N., A.S. B.v.d.S. contributed to editing the final draft of the manuscript.

## Competing interests

The authors declare no competing interests.
