## [Peer Review File · Nature Communications]

Climate-forced Hg-remobilization associated with fern mutagenesis in the aftermath of the end-Triassic extinctionREVIEWER COMMENTS

Reviewer #1 (Remarks to the Author):

Remco Bos and co-workers present in their paper entitled "Climate-forced Hg-remobilization driving mutagenesis in ferns in the aftermath of the end-Triassic extinction" compelling evidence for the effects of volcanic mercury release combined with climate changes, in particular precipitation during the Triassic – Jurassic transition. It is a very thoroughly conducted study, the palynological and cyclostratigraphic analyses were published by Remco et al in GPC, 2022. Here, the authors focus on sedimentary mercury, both concentration and its isotopic composition. The authors show for the first time that based on the mercury isotope composition the volcanic mercury release correlates with the CAMP activity in the TJ interval while younger mercury spikes are related to remobilization and reworking of mercury during periods of enhanced erosion during eccentricity forced wet periods. Interesting is the correlation of the mercury spikes with increased abundance of deformed spores. The latter is thought to be the result of genetical damage of ferns by the toxic mercury load (however I think that some of these, for example the folded spores are just the result of sediment compaction and lithification).

I can follow most of the authors lines of evidence and conclusions. However, I find it difficult to understand how mercury is remobilized by soil erosion during wetter periods while the palynological record, in particular the spore abundance is an "in situ" signal. Would it be not much more logical that a major part of the spores in the Hettangian higher up in the succession are also reworked spores from the TJ extinction interval? I do find the data presented showing convincing evidence that the younger mercury peaks are mainly due remobilization / reworking. BUT WHY is the palynological record is mainly "in situ"? Maybe I only have overlooked some important detail in the authors arguments, but I would like to ask the authors to explain the fundamental discrepancy between the interpretation of the mercury and the palynological signal.

I recommend accepting the paper for publication after minor modifications regarding my question above.

Reviewer #2 (Remarks to the Author):

Bos et al. present an interesting study on the nature of mercury cycling and its potential toxic role in damaging ecosystems during the Triassic-Jurassic extinction. I enjoyed reading large parts of this manuscript, and think it could be an important contribution to our understanding of this event. However, there are some places where I think the authors have oversimplified things, and where clarification and/or more discussion of alternative scenarios is needed. In addition, I would like to see some quantitative estimate of the Hg flux that would have been required to cause the Hg pollution and toxicity hypothesised by the authors, and whether this is realistic in the context of the post-CAMP Hettangian.

The evidence presented by the authors that mercury cycle was disrupted during and after the Triassic extinction in this region (and, as previous studies have demonstrated, the world) is compelling. But what is not clear is whether this would automatically result in spore mutagenesis. Presumably there is a threshold of Hg pollution required to cause this damage? Based on modern studies, can the authors give an idea about how much Hg runoff, remobilization and/or uptake would be needed to lead to mutations? On lines 55–56, the authors mention structural damage of DNA and growth impairment from administering of 2 ppm mercury solutions. Is it possible to model the mercury cycling changes required to produce this level of pollution? And does it produce an outcome that could realistically be occurring in the aftermath of the Triassic extinction? For me, this is really crucial if the authors want to make a compelling argument for Hg toxicity producing these mutations, as just because there is

increased mercury flux and/or uptake does not automatically mean there will be ecological destruction if the change is from 'very low' to 'low'. We really need to have some idea of the numbers involved.

On a related note, I think that the authors are a little too quick to dispense with alternative reasons for causing the mutagenesis, and this also ties in with some erroneous or at least oversimplistic statements regarding CAMP volcanism. On lines 247–248, it is stated that 'only a few sill intrusions are observed within the Planorbis Zone'. This is inconsistent with current age estimates for CAMP basalts and the of the beginning of the Planorbis Zone. The latter is thought to have started very soon after the Jurassic as defined by P. Spelae (probably only 100 kyr at most, so about 201.2 Ma). And the youngest known CAMP lavas from North America and Morocco are dated to around 201.0 Ma, or even later (Whiteside et al., 2010, PNAS; Blackburn et al., 2013, Science). And that is just the lavas that we know about. CAMP is so poorly preserved that it is almost certain that there were other extrusive eruptions happening later (although maybe not much later). I'm not saying that we should assume extrusive CAMP eruptions were occurring throughout the Hettangian (some old ⁴⁰Ar/³⁹Ar studies do suggest that, but no recent ones), but they certainly did continue for a few hundred thousand years after the TJ boundary and Spelae CIE. Of course, that can't explain the spore mutations later in the Hettangian, but it should be considered more for the episodes during the mass extinction interval and immediately afterwards. At the moment, the authors don't really seem to have a reason for ruling out volcanic UV-B destruction other than 'volcanism wasn't happening', but if eruptions were taking place, another argument is needed. One thing to consider is the fact that studies from intrusive sills in the Amazonas Basin have suggested major halogen output at the start of the extinction event. If this were the case, yet few spore mutations were occurring at that time, it might support the authors' argument better. Even then, are there any other ways that mutated spores could arise during times of major wildfire activity and runoff (like those that are recorded throughout the Hettangian part of the record)?

Finally, there are a few places where the authors need to be careful with their stratigraphic descriptions. On lines 277–278, it is stated that 'many global sites only record a single prominent Hg-anomaly at the Spelae CIE.' But in the case of Kuhjoch (Austria) and New York Canyon (Nevada, USA), two sites with some of the best-studied ammonite and $\delta^{13}\text{C}$ stratigraphy available, the first Hg peak is very clearly beginning below the appearance of P. Spelae, at the Marshi CIE. Which begs the question, why isn't a peak occurring correlative with the assumed Marshi CIE at Schandelah-1? There are other examples of stratigraphic unclarities too. For example, on lines 130–132 a maximum duration of 450 kyr is given for the time between the Marshi and Spelae CIEs, but that is quite a lot longer than any estimate I've seen recently, assuming that the extinction begins at 201.6–201.5 Ma and the TJ boundary is at 201.3 Ma. On another occasion, the 'Spelae Event' is referred to, but it isn't clear how long the authors are proposing this to be, or if it is synonymous with the 'Spealae CIE'. None of these necessarily cause major issues for the main results of this study, but some tidying up and clarification of terminology is warranted (more details below).

Other comments:

L. 30–33: 'focusing Hg-pollution in shallow marine basins.' Well, at least in this area, we don't know about elsewhere outside of the central European region.

L. 46: Jost et al. (2017, PNAS) and Somlyay et al. (2023, EPSL).

L. 56: 2 ppm is a very high concentration of mercury to occur in a natural setting, which would require very severe Hg pollution. How realistic is this for the Triassic–Jurassic?

L. 61: Percival et al. (2017, PNAS) did not study China. Shen et al. (2022b, Nature Communications) and Zhang et al. (2022, Frontiers in Ecology and Evolution) have done.

L. 67: I feel as though something is missing from this sentence. Why does the susceptibility to

volatilization make Hg more likely to have a long enough residence time to be globally distributed? There are other gases that do not. I'm also not convinced that the Lindberg et al. (2002, Environmental Science and Technology) paper on the Arctic is the best work to cite here, especially given the comparative lack of polar snow and ice during the Triassic and Early Jurassic. The 2009 volume by Pironne and Mason (Mercury Fate and Transport in the Global Atmosphere) could be a good starting point.

L. 78–80: Presumably this is only applicable to spore-producing plants, rather than all plants? If spore producers were more common in the Triassic–Jurassic, clarify this for the non-expert.

L. 92–93: On line 88 it is stated that 'Hg-MDF is found in nearly all kinetically-controlled reactions'. If this is the case, how can Hg-MDF reliably indicate changes in terrestrial Hg-absorption and the state of the terrestrial biosphere, when there are so many other potential influences?

L. 101–102: Atmospheric Hg is itself characterized by slightly negative MIF (at least today) due in part to the photochemical reduction of mercury, with the mercury featuring positive MIF removed through wet deposition and the residual gaseous Hg inventory left more negative. So plants take up atmospheric mercury that is already characterized by slightly negative MIF, and make it more so.

L. 127–128: 'the widely recorded Marshi (precursor) and Spelae (initial) CIEs'. Be careful with phrasing here. Most of these sites had CIEs recorded initially as the Initial and Main CIE (the Precursor CIE was identified at far fewer locations), which were then re-interpreted as the Marshi and Spelae CIEs.

Also, the Precursor and Initial CIEs were interpreted as being equivalent to the Marshi and Spelae CIEs at St Audries Bay, but at Kujoch the Initial CIE was relabelled Marshi and a separate CIE coeval with the TJ boundary was labelled the Spelae. So the conversion between the 2002 Hesselbo system and 2017 Lindström system is not uniform everywhere, and given that Kujoch is the GSSP, presumably the more sensible reference to make is with that site, not St Audries Bay.

L. 128–130: I know that the authors cite a previous manuscript here but I think more details are needed (at least in a sentence or two) about how an overlap with CAMP can be achieved at this site based on just CIEs and palynology.

L. 132: The TJ boundary has been well dated to no later than 201.3 Ma, based on dating of ash layers just above P. Spelae in Peru and Nevada. In the Fundy Basin of North America and Argana Basin of Morocco, CAMP magmas just above the extinction level have been dated to about 201.56 Ma. So assuming that the Marshi CIE marks the onset of this extinction and the CIE just below those CAMP lavas, the gap between the Marshi and Spelae CIEs should be no more than 250 kyr (and some estimates are shorter). So I'm not sure how the authors have reached a figure as high as 450 kyr. I'd suggest either to explain this, or (if it isn't important for the manuscript) to remove it.

L. 142: Is this duration of 1.3 Myr consistent with previous estimates of the Hettangian? I've seen different durations estimated.

L. 152–153: Technically the lack of disturbances in Sinemurian palynoflora shows stabilization of the vegetation, from which stabilization of the climate has been inferred, not the other way around as implied here.

L. 156: Why only normalizing against TOC, rather than against other possible host species such as sulfur or clays?

L. 174–175: This is true, although from both going over the data in the supplementary tables I found that the average Hg concentration and the variations in them are just as high for the Sinemurian as

for the Hettangian. The thing that has changed is that TOC levels are much higher in the Sinemurian. So has mercury cycling really changed, or is the signal just being overprinted by the excess TOC burial?

L. 184: What is a fern ally?

L. 190–191: The trends are similar, but not perfect. Charcoal is low but S/P high just above the Triassic-Jurassic boundary (310 m), whilst charcoal levels rise but S/P stays low between 300–290 m. So there could be more nuance here.

L. 194–199: The obvious devil's advocate question here is why the base of the extinction interval is below the biggest spike in plant turnover and malformation and the biggest d13C excursion (if we assume that the biggest negative excursion in d13C = the largest carbon cycle disruption)? Taking this line of reasoning further, could it be argued that the very large d13C negative at 318 m is the Marshi CIE and then the Spelae CIE is immediately above it between 316–311 m? In the absence of ammonite biostratigraphy I assume that CIEs have been assigned based on correlations with other sites (as laid out in Bos et al., 2023, Global and Planetary Change).

This is more of a curiosity question about the stratigraphy than a major issue with the manuscript, but I'd be keen to understand the rationale better.

L. 204: Why are these occurrences of Type III, Type IV, or Type V not shown in Figure 4, or in a supplementary Figure?

L. 245–249: 'two phases of extrusive volcanism that ended following the Spelae event'. What is meant by the Spelae event? Is this synonymous with the CIE at the TJ boundary? The youngest CAMP lavas are from about 201.0 Ma, which is well in to the Hettangian after the boundary at 201.3 Ma. Also, it would be better to cite the original U-Pb references on line 246.

Following on from this, the sentence 'This post-ETME intrusive phase was likely restricted to passive and diffusive emissions unable to significantly affect the stratosphere, where the ozone layer resides.' is highly speculative. We know that there were definitely Hettangian lavas, and given how poorly CAMP is preserved, if there are some surviving lavas then it is not unreasonable to assume that there could have been extensive extrusive volcanism. In any case, we know very little about the dynamics of LIP magmas and how they could have impacted the stratosphere. Indeed, it has been modelled that effusive eruptions could have ejected gas emissions high enough to potentially reach the lower stratosphere (see Schmidt et al., 2016, Nature Geosciences).

L. 267–269: So why is there no attempt to normalize against these other potential host phases? Especially given that on line 259, adsorption on to S-rich minerals is also mentioned. See earlier point.

L. 271–275: See also the model of Sanei et al. (2012, Geology), which invokes a similar model regarding the balance of TOC and S drawdown of Hg being disrupted at a time of increased volcanic influx (in the context of the Siberian Traps and Permian extinction). Might be worth thinking about.

Also are the several sites referred to as having euxinic conditions around the Spelae CIE all in this area of central Europe? Or distributed globally? Clarify.

L. 275–277: Again, be careful with phrasing here. Firstly, the Hg peaks at several sites (certainly from the Thibodeau et al., 2016; Percival et al., 2017; and Kovacs et al., 2020 papers) were correlated against the Initial and/or Main CIEs, and as noted above, conversion to the Marshi/Spelae CIEs is not always straightforward. But in any case, at both Kuhjoch and New York Canyon, the Hg peak begins at the Marshi CIE (see Lindström et al., 2021, Earth Science Reviews), not the Spelae. And there are some sites where there is only one clear d13C excursion, which makes it harder to work out which

level (and any correlative Hg spike) is equivalent to (e.g., Kovacs et al., 2020, *Global and Planetary Change*; De Graaff et al., 2022, *PPP*).

L. 282–283: 'assumed absence'.

L. 286–288: Looking at the supplementary data tables, many of the samples have fairly low HI values (often under 50), and relatively high OI (often over 100, sometimes well over 100). Whilst the fact that the organic matter is primarily terrestrial does make low HI and higher OI more likely, some of these OI values are very high and does suggest some oxidation of the organic matter throughout. Is there any petrographic evidence to support this? If so, even if there is no direct correlation between Hg and HI or OI, it could still hint that the record is potentially oxidized, which can complicate interpretation of Hg and Hg/TOC trends.

L. 306–318: This section would really benefit from some quantitative modelling to estimate the flux of Hg available for runoff and/or stomatal uptake. Fendley et al. (2019, *EPSL*) reconstructed the Hg cycle and volcanic output during the KPg boundary by utilizing the Hg proxy record and box model of Amos et al. (2015, *Environmental Science and Technology*). Could something similar be attempted, even roughly, here to at least test whether the degree of Hg pollution required to cause spore mutation is consistent with the recorded Hg and Hg/TOC peaks? I acknowledge that this could be quite a big ask, but given the potential impact of the conclusion and journal, having some quantification would really help to strengthen the arguments presented.

L. 321: 'assumed to be near-zero MIF'. Yes, assuming that the Hg emissions were magmatically sourced rather than from heating of sediments by intrusive magmas (see Shen et al., 2022a, *Nature Communications*).

L. 324: I wouldn't describe -0.5 to -0.3 as 'minimal variability'. It may be less variability than in the rest of the Schandelah-1 record, but it's still quite a lot and is equal to or in some cases more variability than in other TJ records (see compilation in Shen et al., 2022a), especially given the relative paucity of data points. It's also worth noting that there is only one data point from below the proposed Marshi CIE and extinction interval, so not really a pre-CAMP background.

L. 333: The word 'volcanic' isn't really needed before 'Hg-input' here. The low Hg concentrations presumably indicate a low input of mercury overall, regardless of actual source(s).

L. 345: Give the low MIF values for these coal deposits.

L. 349–351: Re-emphasize any other evidence that independently supports this to strengthen the argument here.

L. 429–430: It is worth reminding the reader here that a similar scenario has been suggested for the marine realm as well (see Thibodeau et al., 2016), with CAMP potentially delaying the recovery of calcareous fauna and allowing an initial dominance of siliceous organisms post-extinction (although not for as long).

Figures 3 and 4: I find the illustrated history of CAMP (showing an extrusive phase at the start with only intrusive pulses at the end) odd. The oldest known CAMP magmas are intrusive bodies from Guinea and Bolivia (Davies et al., 2017, *Nature Communications*), and as noted in earlier comments, there are later extrusives from well into the Hettangian.

General point on figures: Given that the authors have an age model for this site, it would be nice to have a figure plotting $\delta^{13}\text{C}$, Hg, palynology, CAMP magmatism vs time in Ma.

References:

- Blackburn, T.J., Olsen, P.E., Bowring, S.A., McLean, N.M., Kent, D.V., Puffer, J., McHone, G., Rasbury, E.T. and Et-Touhami, M., 2013. Zircon U-Pb geochronology links the end-Triassic extinction with the Central Atlantic Magmatic Province. *Science*, 340(6135), pp.941-945.
<https://doi.org/10.1126/science.1234204>.
- Davies, J.H.F.L., Marzoli, A., Bertrand, H., Youbi, N., Ernesto, M. and Schaltegger, U., 2017. End-Triassic mass extinction started by intrusive CAMP activity. *Nature communications*, 8(1), 15596.
<https://doi.org/10.1038/ncomms15596>.
- De Graaff, S.J., Percival, L.M., Kaskes, P., Déhais, T., De Winter, N.J., Jansen, M.N., Smit, J., Sinnesael, M., Vellekoop, J., Sato, H. and Ishikawa, A., 2022. Geochemical records of the end-Triassic Crisis preserved in a deep marine section of the Budva Basin, Dinarides, Montenegro. *Palaeogeography, Palaeoclimatology, Palaeoecology*, 606, 111250, <https://doi.org/10.1016/j.palaeo.2022.111250>.
- Fendley, I.M., Mittal, T., Sprain, C.J., Marvin-DiPasquale, M., Tobin, T.S. and Renne, P.R., 2019. Constraints on the volume and rate of Deccan Traps flood basalt eruptions using a combination of high-resolution terrestrial mercury records and geochemical box models. *Earth and Planetary Science Letters*, 524, 115721, <https://doi.org/10.1016/j.epsl.2019.115721>.
- Hesselbo, S.P., Robinson, S.A., Surlyk, F. and Piasecki, S., 2002. Terrestrial and marine extinction at the Triassic-Jurassic boundary synchronized with major carbon-cycle perturbation: A link to initiation of massive volcanism?. *Geology*, 30(3), pp.251-254, [https://doi.org/10.1130/0091-7613\(2002\)030<0251:TAMEAT>2.0.CO;2](https://doi.org/10.1130/0091-7613(2002)030<0251:TAMEAT>2.0.CO;2).
- Jost, A.B., Bachan, A., van De Schootbrugge, B., Lau, K.V., Weaver, K.L., Maher, K. and Payne, J.L., 2017. Uranium isotope evidence for an expansion of marine anoxia during the end-Triassic extinction. *Geochemistry, Geophysics, Geosystems*, 18(8), pp.3093-3108,
<https://doi.org/10.1002/2017GC006941>.
- Kovács, E.B., Ruhl, M., Demény, A., Fórizs, I., Hegyi, I., Horváth-Kostka, Z.R., Móricz, F., Vallner, Z. and Pálffy, J., 2020. Mercury anomalies and carbon isotope excursions in the western Tethyan Csóvár section support the link between CAMP volcanism and the end-Triassic extinction. *Global and Planetary Change*, 194, 103291, <https://doi.org/10.1016/j.gloplacha.2020.103291>.
- Lindberg, S.E., Brooks, S., Lin, C.J., Scott, K.J., Landis, M.S., Stevens, R.K., Goodsite, M. and Richter, A., 2002. Dynamic oxidation of gaseous mercury in the Arctic troposphere at polar sunrise. *Environmental science & technology*, 36(6), pp.1245-1256, <https://doi.org/10.1021/es0111941>.
- Lindström, S., van De Schootbrugge, B., Hansen, K.H., Pedersen, G.K., Alsen, P., Thibault, N., Dybkjær, K., Bjerrum, C.J. and Nielsen, L.H., 2017. A new correlation of Triassic–Jurassic boundary successions in NW Europe, Nevada and Peru, and the Central Atlantic Magmatic Province: a time-line for the end-Triassic mass extinction. *Palaeogeography, Palaeoclimatology, Palaeoecology*, 478, pp.80-102, <https://doi.org/10.1016/j.palaeo.2016.12.025>.
- Lindström, S., Callegaro, S., Davies, J., Tegner, C., van De Schootbrugge, B., Pedersen, G.K., Youbi, N., Sanei, H. and Marzoli, A., 2021. Tracing volcanic emissions from the Central Atlantic Magmatic Province in the sedimentary record. *Earth-Science Reviews*, 212, 103444,
<https://doi.org/10.1016/j.earscirev.2020.103444>.
- Percival, L.M., Ruhl, M., Hesselbo, S.P., Jenkyns, H.C., Mather, T.A. and Whiteside, J.H., 2017. Mercury evidence for pulsed volcanism during the end-Triassic mass extinction. *Proceedings of the National*

Academy of Sciences, 114(30), pp.7929-7934, <https://doi.org/10.1073/pnas.1705378114>.

Pironne, N. and Mason, R. (Eds.), *Mercury Fate and Transport in the Global Atmosphere: Emissions, Measurements and Models*. Springer, Dordrecht, The Netherlands, 2009.

Sanei, H., Grasby, S.E. and Beauchamp, B., 2012. Latest Permian mercury anomalies. *Geology*, 40(1), pp.63-66, <https://doi.org/10.1130/G32596.1>.

Shen, J., Yin, R., Algeo, T.J., Svensen, H.H. and Schoepfer, S.D., 2022. Mercury evidence for combustion of organic-rich sediments during the end-Triassic crisis. *Nature Communications*, 13(1), 1307, <https://doi.org/10.1038/s41467-022-28891-8>.

Shen, J., Yin, R., Zhang, S., Algeo, T.J., Bottjer, D.J., Yu, J., Xu, G., Penman, D., Wang, Y., Li, L. and Shi, X., 2022b. Intensified continental chemical weathering and carbon-cycle perturbations linked to volcanism during the Triassic–Jurassic transition. *Nature Communications*, 13(1), 299, <https://doi.org/10.1038/s41467-022-27965-x>.

Somlyay, A., Palcsu, L., Kiss, G.I., Clarkson, M.O., Kovács, E.B., Vallner, Z., Zajzon, N. and Pálffy, J., 2023. Uranium isotope evidence for extensive seafloor anoxia after the end-Triassic mass extinction. *Earth and Planetary Science Letters*, 614, 118190, <https://doi.org/10.1016/j.epsl.2023.118190>.

Thibodeau, A.M., Ritterbush, K., Yager, J.A., West, A.J., Ibarra, Y., Bottjer, D.J., Berelson, W.M., Bergquist, B.A. and Corsetti, F.A., 2016. Mercury anomalies and the timing of biotic recovery following the end-Triassic mass extinction. *Nature Communications*, 7(1), 11147, <https://doi.org/10.1038/ncomms11147>.

Whiteside, J.H., Olsen, P.E., Eglinton, T., Brookfield, M.E. and Sambrotto, R.N., 2010. Compound-specific carbon isotopes from Earth's largest flood basalt eruptions directly linked to the end-Triassic mass extinction. *Proceedings of the National Academy of Sciences*, 107(15), pp.6721-6725, <https://doi.org/10.1073/pnas.1001706107>.

Zhang, P., Lu, J., Yang, M., Bond, D.P., Greene, S.E., Liu, L., Zhang, Y., Wang, Y., Wang, Z., Li, S. and Shao, L., 2022. Volcanically-induced environmental and floral changes across the Triassic-Jurassic (TJ) transition. *Frontiers in Ecology and Evolution*, 10, <https://doi.org/10.3389/fevo.2022.853404>.

Reviewer #3 (Remarks to the Author):

Climate-forced Hg-remobilization driving mutagenesis in ferns in the aftermath of the end-Triassic extinction

Remco Bos, Wang Zheng, Sofie Lindström, Hamed Sanei, Irene Waajen, Isabel Fendley, Tamsin A. Mather, Yang Wang, Jan Rohovec, Tomáš Navrátil, Appy Sluijs, Bas van de Schootbrugge

Dear authors,

I read the manuscript with great interest. The work presents new insights into the environmental dynamics in the aftermath of large-scale volcanism. The study is the first to trace mercury (Hg) concentrations and isotopes through the Lower Jurassic Hettangian stage in a stratigraphic framework and resolution high enough to interpret shifts in relation to orbital cycles, and in combination with data on the mutagenesis in ferns. The Hg concentrations and Hg isotopes show that orbitally-paced modulation of the climate leads to the remobilization and emission of initially volcanically sourced Hg over a time period of (at least) 2 million years following volcanic emission. The study further shows that the recycling of Hg into the environment causes the repeated increase in the abundance of mutagenic fern spores. The findings are of interest to a wide range of readers from the fields of earth science, paleoclimate and paleoenvironments.

Overall, the study presents a robust data set, which is reported, presented and discussed in detail. In parts, the text is hard to follow for readers that are not familiar with Hg isotope data, and the discussion of the data should be extended to certain aspects, also in regards to published literature (for details see below).

I recommend the study for publication after minor revision.

For the revision, I suggest to address the following concerns/suggestions:

- Hg concentrations and isotope data obtained from the Triassic–Jurassic (T–J) boundary interval should be discussed in the context of recent studies providing similar data sets from the same time interval from various settings (e.g., [e.g., 1, 2, 3]). Sections on which Hg data is available for the T–J boundary are shown in Figure 1, but there is little discussion in how your data compare to the published data and data interpretation.
- The Hg enrichments and Hg isotope data of the T–J boundary interval should also be discussed in respect to thermogenic Hg generated through volatilization of sedimentary organic matter through CAMP-related igneous sills
- I recommend to discuss the impact of wildfires on the Hg isotope values. Since the frequency of wildfires is modulated by orbital parameters (e.g., [4]), this should be especially important for the Hettangian data.
- With the orbital pacing of the climate and subsequent environmental shifts being the main mechanism driving the Hg remobilization in the Hettangian, I recommend to put more emphasis on this aspect, also in the visual presentation of your data. You refer to the eccentricity pacing of other data as shown by [5], and present a correlation of the bandpass filter of the dominant 26.5 m dominant spectral peak (which are interpreted to represent long eccentricity cycles) to the Hg concentration data in Figure 3. You also present a model illustrating the mechanisms of Hg mobilization and isotope fractionation in response to eccentricity forcing (Figure 6). Correlation between eccentricity pacing of the Hg isotopes shifts and abundance of malformed spores is not presented but referred to in the text. I recommend to visualize this aspect more clearly in figure 4 and add some information in the text additional to referring to [5].
- The naming of Marshi CIE (referring to the precursor CIE) and Spelae CIE (referring to the initial CIE) is somewhat confusing when looking at the stratigraphic position of these shifts in other section with respect to the last occurrence of *C. marshi* and first occurrence *P. Spelae* (e.g. [6, 7]). In the Kuhjoch section, for example, the initial CIE corresponds to the LwO of *C. marshi*, while the first occurrence of *P. spelae* marks the main CIE [6]. I recommend to show a correlation in the supplements that correlates the isotope shifts recorded in Schandelah-1 to a section with better (bio) stratigraphic control in this stratigraphic interval.
- HI and OI data are not mentioned in the results section but discussed in the discussion part. It is not clear of the data are from previously published work or new data of this study.
- The data resolution is not sufficient to make statements about whether the eccentricity-paced Hg mobilization had ceased in the Sinemurian, considering the time covered by the Arietenton and Obtususton Formations.

Minor comments:

Abstract:

Line 27: change to lower Jurassic Hettangian stage (provides more context for the reader that is not familiar with the Jurassic stages since you refer to 'early Jurassic peaks' later in the text)

Line 28 – change 'spikes' to increase

Line 29 – volcanic source of the Hg enrichment

Line 33 – do you mean forcing?

Introduction:

Line 40–42: mention that the negative carbon-isotope excursion indicate the release of isotopically-light carbon into the ocean-atmosphere system, otherwise the context is not clear

Line 45–46: perhaps link the nutrient flux to the increased weathering rates for context

Line 55–56: mention that the experimental studies were performed on plants
Line 56: concentrations of $>10 \mu\text{M}$ (2 ppm) Hg
Line 65: add reference for the hypothesized release of Hg
Line 69–75: Hg isotopes are not my expertise and I find the introduction of the Hg isotope system hard to follow in parts. Does the deposition via rainfall and and/or particle fallout refer to both Hg^{2+} and Hgp? Otherwise, how does the Hg^{2+} enters the environment that contains sulfate- and/or iron-producing bacteria?
Line 75: explain briefly what ligand-bound means
Line 79: gaseous uptake of Hg^0 is
Line 82–83: Make clearer that Hg-speciation across the ETME has been presented in previous studies, and the new approach of this study is the Hg-speciation throughout the Hettangian and lower Sinemurian to assess the long-term effects.
Line 85: A shift Shifts in
Line 87: refer to methods section here for details on $\delta^{202}\text{Hg}$ and $\Delta^{199}\text{Hg}$
Line 88: low depleted
Line 92: often can
Line 92: what does 'state of terrestrial biosphere' refer to?
Line 92–104: This paragraph is very difficult to follow. First, the MDF isotope effect of stomatal Hg uptake in foliage is discussed, then MIF is discussed, and at the end the photochemical reaction of MDF is mentioned. For MIF, the photochemical reactions are mentioned as a key role in MIF variability. That photochemical reduction causes positive shifts is only at the very end of the paragraph. I think this can be re-structured in a way that is easier to follow.
Also, be consistent with either, e.g., Hg-MDF or just MDF.
Line 116–117: this could be more specific with what you mean with 'a shift to gaseous forms of Hg'. Do you mean re-mobilization through continental erosion and subsequent shift to gaseous Hg through photochemical reaction?

Results:

Line 125: cross-correlated cross-correlated
Line 126–128: check references. The Marshi (precursor) CIE is not discussed in [7] and not explicitly associated with carbon injection in [8]. Please cite a reference that clearly defines the stratigraphic position of these two CIEs, and defines the Marshi CIE as precursor, and the Spelae CIE as initial CIE. Ideally present a correlation in the supplements. As far as I am aware, the LwO of *P. spelae* marks the T-J boundary, but the initial CIE (which is here called Spelae CIE) correlates with the last occurrence of *C. marshi* [6]
Line 128–130: Also refer to [9], who are cited in [5] referring to the palynological assemblage
Line 132–133: Ref. [10] correlate the onset of the extinction interval globally. If the regression is limited to the NW European realm the sentence needs to be restructured.
Line 138: Give reference for the transgression and state how you identify the transgression in the Schandelah-1 core. Also refer to the [5, 9] regarding the $\delta^{13}\text{C}$ and TOC data in text, and give reference for TOC data in Fig. 3
Line 140: often widely
Line 142: Make clear that the duration of ~ 1.3 Ma is referring to the Hettangian succession preserved in the Schandelah-1 core, and not to the Hettangian stage in general. Due to the unconformities at the base and top of the Angulatenton Fm, the cyclostratigraphy performed on the Schandelah-1 data did not capture the entire stage (as mentioned in [5])
Line 145: be more specific about what you refer to with the word 'disturbance'
Line 148: How do you define 'pre-extinction' $\delta^{13}\text{CTOC}$ values? In the Schandelah-1 record, there are only 5 $\delta^{13}\text{CTOC}$ values from the strata preceding the Marshi CIE.
Line 152: What kind of disturbances are you referring to?

Hg trends and anomalies

Line 152: this would be easier to follow if the stratigraphic position of the Spelae CIE/anomaly was more precisely defined in the above text

Line 169: I am not sure if you can distinguish between a high and a low anomaly
Line 178–179: comma missing

Palynofloral disturbance and spore malformations:

Line 182–191: this paragraph is mainly summarizing the findings of [5], which should not be part of the results section

Line 201–204: For consistency, if you mention that malformation Type-1 is linked to abortion and premature shedding of non-viable spores, you should mention what the other malformation types are linked to (if known). This could be placed in the supplements

Line 204: Unless you clearly link the four distinct peaks in spore malformation abundance to the long-eccentricity cycles identified in TOC and the CIEs you should refer to them as reoccurring rather than periodic

Line 208–211: refer to data table in the supplements

Line 211: lower lowermost Sinemurian

Hg-isotopes:

Line 219: following the sharp negative shift (...), average $\Delta^{199}\text{Hg}$ values remain a -0.18‰ within the Hettangian

Line 221: periodic reoccurring/repetitive

Line 222: anomalies in Hg concentrations

Line 222–223: the main feature in the Sinemurian seems to be a broad negative shift in $\Delta^{199}\text{Hg}$

Line 223: place discussion on $\delta^{202}\text{Hg}$ in new paragraph

Line 223–225: there is a word missing in this sentence.

Line 223–226: You call the Triletes Beds the main extinction interval, but Fig. 3 and 4 mark the ETME for the interval of the Contorta beds to Ppsilonoten Sst, which makes this section hard to follow. Also, check the ranges given here. The higher values preceding the negative Spelae shift seem to range from -1.60‰ to -1.01‰ .

Line 227: anomalies in Hg concentrations

Discussion:

Line 232–233: label the Marshi and Spelae CIEs as nCIEs in Figure 3 and 4, or refer to the Marshi and Spelae negative CIEs in text

Line 234–235: Be more specific about the similarities. Is it the timing, or the severity of malformation?

Line 240–241: better refer to greenhouse gas emissions as it includes other carbon sources such as thermogenic methane

Line 242–245: I suggest to rephrase this sentence, suggesting that the prolonged and repeated mutagenesis in fern communities suggests additional mechanisms acting on longer time scales

Line 265: carbon burial did not play a significant role in Hg enrichment

Line 271: a short-lived peak in TOC is not sufficient evidence for a transgression event

Line 273–275: please be more specific what you refer to as initial rise in bulk Hg concentrations. Do you refer to the interval that correlates with increased TOC? It could also be useful to separate the two intervals in the TOC/Hg cross-plot (Figure S2B). How do you exclude that the decoupling between Hg and TOC (meaning more Hg than it would be expected if it was linked increased organic matter drawdown) is not caused by volcanic input? This would be consistent with the MIF data.

The increase of Hg-burial through increased S-binding is a local acting mechanisms and inter-basin correlation is not sufficient proof. Can you provide any sedimentological or geochemical evidence that suggest photic zone euxinia in the Schaldelah-1 core?

Line 280: add that these extrusive phases might be reflected in the Hg-enrichment intervals to bring this into context

Line 281: Spelae Hg anomaly (to indicate that the Hg enrichment was not driven by increased OM deposition and/or preservation)

Line 282: cyclic repetitive/reoccurring nature

Line 287–288: It is not clear whether the HI and OI data are part of this study (mentioned in methods section, but not discussed in the results section), or if this is published data from [5], in which case you need to add the reference and remove the respective part in the methods section

Line 287: how do you distinguish post-depositional from primary (e.g. water column) oxidation?

Do you have Tmax data as maturity indicator? What would be the effects of thermal alteration on Hg concentrations and Hg isotopes?

Line 290–291: 'halted paleo-redox front' needs a bit more explanation

Line 292–296: specify if this is true for all shifts in Hg/TOC or if you refer to the Hg-enrichments in the Hettangian interval

Line 311–318: shifted to its gaseous form would magnify the mutagenic potency of Hg on vegetation, but it would also decrease the amount of Hg that is transported and accumulated in shallow marine environments. Regarding gaseous Hg, I am wondering about the effect of fires on the release of Hg, also in relation to eccentricity pacing (see [4] for eccentricity forcing of wild fires)

Line 319: Spelae Hg anomaly

Line 324: $\Delta^{199}\text{Hg}$ values range from -0.50‰ to -0.31‰ according to results section, and were described as highly negative

Line 325–326: it would be useful to mention in the introduction of the Hg isotope system that negative MIF in vegetation is attributed to photochemical reduction (loss of Hg) in foliage

Line 328–329: Positive shift in MDF in the Hettangian Triplets beds preceding the Spelae CIE (makes it easier to follow)

Also, does the reference indicate that this is the interpretation of another study, or are you referring to the processes? Can you add more context on the microbial and abiotic processes that you are referring to?

Line 329–332: This is really hard to follow. Do you mean that due to low abundance of vegetation the uptake by soil increases, where it is degraded and produced positive MDF? Also, wouldn't a low abundance of vegetation cover lead to higher photochemical reduction in soils?

Line 319–335: the paragraph on MDF and MIF signals associated with the T–J transition and volcanism is missing discussion of published data from the same interval. There are a number of Hg-concentration records and Hg-isotope records from different sites [e.g., 1, 2, 3]. Do the data show the same trends and lead to similar conclusions? The paragraph is also missing discussion regarding Hg release from combustion of organic-rich sediments

Line 341: specify mineral sources

Line 345: this might not be the right reference there. But since coal is plant material, a negative MIF would be expected

It could also be mentioned that ref. [5] show that the positive CIEs in the Hettangian (that correspond to increased mutagenesis) also correspond to the occurrence of reworked palynomorphs, implying increased reworking in this interval

Line 377–378: the periodic nature of what?

Line 380–381: This needs a bit more detail. The fire activity is mentioned here for the first time.

Additional to the photochemical reduction, I am wondering if fires would not lead to a volatilization of Hg, and what impact the fire would have on the Hg-isotope signature (see, e.g., [11]).

The orbital pacing of wild fire events (e.g., [4]) seems to be an important aspect here that should be mentioned here, also in correlation with the charcoal abundance presented in [5].

Line 384: cyclic reoccurring

Line 389: specify what extinction you refer to (marine?)

Line 308–499: This plot shown in Fig. 5D appears as important to the overall story and may need a bit more detail. It would be useful to indicate which data corresponds to the Hettangian Hg-isotope anomalies

Line 399–404: Here you mention orbital pacing and refer to Fig. 6B, which shows the scenarios of eccentricity minima and maxima, and in the following text you briefly discuss the impact of orbital eccentricity forcing on the vegetation and Hg mobilization and degassing. The climate forcing in the fern mutagenesis appears as one of the main findings of the study (according to the title of the manuscript, and the abstract). For the importance this aspect of orbital forcing is giving in the

beginning of the manuscript, the discussion as relatively short. In Figure 3 you show the correlation of Hg-concentration anomalies and eccentricity maxima. It would be useful to also show the correlation of the spore malformation and Hg-isotope data to the eccentricity. The climatic effects of eccentricity minima and maxima need a reference, e.g., [12].

Line 411–413: specify 'disturbed conditions'

I think the resolution of the presented data is not sufficient for stating that the orbitally-paced Hg release has ceased in the Sinemurian. The time interval that is covered by the Sinemurian strata (bucklandi to mid-obtusum Zone) may cover some 4 Myr (see [13]). The amount of data points for both Hg concentrations and isotopes may not be sufficient to identify eccentricity cycles. Furthermore, considerable amounts of strata are missing in the Schandelah-1 core due to a hiatus (indicated by missing turneri Zone)[9].

Line 412: The time scale presented in this study refers to 1.3 Million years based on [5]. The duration of 2 million years for the Hettangian is mentioned here for the first time and needs a reference

Line 418: this is the first time that extreme greenhouse conditions are mentioned. Needs reference and some context.

Line 420–421: It needs to be pointed out that the volcanic sourcing of Hg during the T–J transition is not a new finding but was already observed in previous studies

Line 414–430: the concluding paragraph should mention the eccentricity pacing to draw a full circle back to the abstract and bring the climatic shifts in context

Materials and methods:

Line 475–488: is all organic carbon data is not discussed in the results section. Please specify if the data (or some of the data) are part of this study, or refer to reference

Figures:

Fig. 1: The sections containing Hg concentration data and Hg isotope data shown in the figure are mentioned in the introduction, but are not discussed in relation to your data

Fig. 4: Ammonite Zonation is missing in the stratigraphic column. The ammonite zonation is crucial in order to correlate the data to other sections.

I hope this review is fair and the comments constructive.

Kind regards,

Marisa Storm

1. Thibodeau, et al. (2016) - Mercury anomalies and the timing of biotic recovery following the end-Triassic mass extinction. Nature Communications
2. Yager, et al. (2021) - Mercury contents and isotope ratios from diverse depositional environments across the Triassic–Jurassic Boundary: Towards a more robust mercury proxy for large igneous province magmatism. Earth-Science Reviews
3. Shen, et al. (2022) - Mercury evidence for combustion of organic-rich sediments during the end-Triassic crisis. Nature Communications
4. Hollaar, et al. (2021) - Wildfire activity enhanced during phases of maximum orbital eccentricity and precessional forcing in the Early Jurassic. Communications Earth & Environment
5. Bos, et al. (2023) - Triassic–Jurassic vegetation response to carbon cycle perturbations and climate change. Global and Planetary Change
6. Ruhl, et al. (2009) - Triassic–Jurassic organic carbon isotope stratigraphy of key sections in the western Tethys realm (Austria). Earth and Planetary Science Letters
7. Hesselbo, et al. (2002) - Terrestrial and marine extinction at the Triassic–Jurassic boundary synchronized with major carbon-cycle perturbation: A link to initiation of massive volcanism? Geology
8. Lindström, et al. (2012) - No causal link between terrestrial ecosystem change and methane release during the end-Triassic mass extinction. Geology
9. van de Schootbrugge, et al. (2018) - The Schandelah Scientific Drilling Project: A 25-million year

record of Early Jurassic palaeo- environmental change from northern Germany. Newsletters on Stratigraphy

10. Lindström, et al. (2017) - A new correlation of Triassic–Jurassic boundary successions in NW Europe, Nevada and Peru, and the Central Atlantic Magmatic Province: A time-line for the end-Triassic mass extinction. *Palaeogeography, Palaeoclimatology, Palaeoecology*

11. Richter, et al. (2023) - Impact of forest fire on the mercury stable isotope composition in litter and soil in the Amazon. *Chemosphere*

12. Martinez, et al. (2015) - Orbital pacing of carbon fluxes by a ~9-My eccentricity cycle during the Mesozoic. *Proceedings of the National Academy of Sciences*

13. Storm, et al. (2020) - Orbital pacing and secular evolution of the Early Jurassic carbon cycle. *Proceedings of the National Academy of Sciences*

Reviewer #1:

Remco Bos and co-workers present in their paper entitled "Climate-forced Hg-remobilization driving mutagenesis in ferns in the aftermath of the end-Triassic extinction" compelling evidence for the effects of volcanic mercury release combined with climate changes, in particular precipitation during the Triassic – Jurassic transition. It is a very thoroughly conducted study, the palynological and cyclostratigraphic analyses were published by Remco et al in GPC, 2022. Here, the authors focus on sedimentary mercury, both concentration and its isotopic composition. The authors show for the first time that based on the mercury isotope composition the volcanic mercury release correlates with the CAMP activity in the TJ interval while younger mercury spikes are related to remobilization and reworking of mercury during periods of enhanced erosion during eccentricity forced wet periods. Interesting is the correlation of the mercury spikes with increased abundance of deformed spores. The latter is thought to be the result of genetical damage of ferns by the toxic mercury load (however I think that some of these, for example the folded spores are just the result of sediment compaction and lithification).

I can follow most of the authors lines of evidence and conclusions. However, I find it difficult to understand how mercury is remobilized by soil erosion during wetter periods while the palynological record, in particular the spore abundance is an “in situ” signal. Would it be not much more logical that a major part of the spores in the Hettangian higher up in the succession are also reworked spores from the TJ extinction interval? I do find the data presented showing convincing evidence that the younger mercury peaks are mainly due remobilization / reworking. BUT WHY is the palynological record is mainly “in situ”? Maybe I only have overlooked some important detail in the authors arguments, but I would like to ask the authors to explain the fundamental discrepancy between the interpretation of the mercury and the palynological signal.

I recommend accepting the paper for publication after minor modifications regarding my question above.

The Hg-signal of the Hettangian is indeed interpreted as a partly reworked signal. Positive excursions in Hg-isotope records strongly suggest Hg to be derived from soil and/or bedrock material. Soil erosion of the larger catchment area is transported to low-lying coastal margins (proximity to Schandelah-1 depositional site) during times of extreme precipitation. A potential bedrock Hg-signal is inferred from the presence of reworked upper Triassic palynomorphs described in Bos et al. (2023) and exhibit darkened and broken wall material. During examination and the counting of palynological slides we assessed these features to represent reworking and excluded any spore specimen (normal and malformed) that did not exhibit an *in situ* appearance, although these were far and few in between. In addition, the reworked palynomorph fractions during these intervals is low (1-5% of total assemblage, see Bos et al., 2023) and would not account for the increased malformed signal that we present for the Hettangian. We are confident that the presented malformed spore signal is an *in situ* one. Furthermore, some of the spore malformations, particular the folded specimens, have been suggested to be the result of preservation (sediment compaction and lithification). However, we note an increased frequency of these morphologies in samples that also exhibit higher frequencies in other malformations. Therefore, we included this morphology to represent a malformed signal (see Lindström et al., 2019). We have reiterated the counting strategy regarding the exclusion of reworked palynomorphs in the Material and Methods section and expressed our confidence on an *in situ* signal in the Results section. In addition, we have constructed a major revised environmental model (figure 6) that emphasis the pathways of

transported materials from hinterland catchment areas to low-lying coastal margins. Finally, we changed some descriptions of malformed features in figure 2 (and Material and Methods) to better reflect the observed morphologies.

Reviewer #2:

Bos et al. present an interesting study on the nature of mercury cycling and its potential toxic role in damaging ecosystems during the Triassic-Jurassic extinction. I enjoyed reading large parts of this manuscript, and think it could be an important contribution to our understanding of this event. However, there are some places where I think the authors have oversimplified things, and where clarification and/or more discussion of alternative scenarios is needed. In addition, I would like to see some quantitative estimate of the Hg flux that would have been required to cause the Hg pollution and toxicity hypothesised by the authors, and whether this is realistic in the context of the post-CAMP Hettangian.

The evidence presented by the authors that mercury cycle was disrupted during and after the Triassic extinction in this region (and, as previous studies have demonstrated, the world) is compelling. But what is not clear is whether this would automatically result in spore mutagenesis. Presumably there is a threshold of Hg pollution required to cause this damage? Based on modern studies, can the authors give an idea about how much Hg runoff, remobilization and/or uptake would be needed to lead to mutations? On lines 55–56, the authors mention structural damage of DNA and growth impairment from administering of 2 ppm mercury solutions. Is it possible to model the mercury cycling changes required to produce this level of pollution? And does it produce an outcome that could realistically be occurring in the aftermath of the Triassic extinction? For me, this is really crucial if the authors want to make a compelling argument for Hg toxicity producing these mutations, as just because there is increased mercury flux and/or uptake does not automatically mean there will be ecological destruction if the change is from ‘very low’ to ‘low’. We really need to have some idea of the numbers involved.

The concentrations of environmental Hg required to induce mutagenesis and structural DNA damage varies depending on the type of plant and setting. It is true that the most often concentrations mentioned in several studies is around 2 ppm Hg in solutions which clearly produce structural DNA damage which would lead to mutagenesis. However, the toxicity of any given substance or element is dependent on more than just the concentrations. In this manuscript we argue that the speciation of Hg is a leading factor for its potential mutagenic-inducing qualities. Both volcanic Hg-degassing and photochemical reduction of remobilized Hg produce gaseous Hg that is likely more easily incorporated in plant structure through stomatal uptake. In addition, the production of spore-producing sporangia requires the formation of newly formed plant material that more easily take up the environmental gaseous Hg. The volatile nature gives Hg an edge over any other heavy metal pollutant. This makes Hg more susceptible for remobilization and provides several ways for Hg to enter plant structures to cause interference. The actual concentrations of gaseous Hg emissions required to induce these interferences and damages is unknown, although the speciation (namely gaseous Hg) is at least an important predictor of potential mutagenesis in ferns. We have made changes in the text to emphasize this point more clearly. Furthermore, we are of the opinion that the geological record is insufficiently equipped to confidently demonstrate terrestrial Hg concentrations from mostly marine records such as the Schandelah-1 core. This would require a precise estimation to the size of volcanic Hg-emissions, the size of Hg weathering catchment areas and sedimentation rates to deduce terrestrial Hg concentrations. Experimental studies and modern pollution sites are much better suited for the problem of Hg concentrations, of which we are involved at the moment and show very promising preliminary results. We are confident that our results show a clear correlation between marine Hg concentrations and spore malformations. The close proximity of the Schandelah-1 core to the paleo-shoreline combined with evidence for terrestrially-derived Hg provides further evidence that terrestrial Hg was also elevated. This suggests increased concentrations of Hg were necessary to induce fern

mutagenesis. In addition, the speciation and origin of early Hettangian Hg-anomalies, as indicated by Hg-isotope records, clearly demonstrate Hg derived from terrestrial/weathering sources that has undergone photochemical reduction and a shift towards gaseous Hg. Similarly, the gaseous volcanic Hg-degassing has high potential to be taken up by plant stomata. Although we made changes in text to emphasize these points, we believe that precise estimation of concentrations of terrestrial Hg from marine records is beyond the scope of this manuscript.

On a related note, I think that the authors are a little too quick to dispense with alternative reasons for causing the mutagenesis, and this also ties in with some erroneous or at least oversimplistic statements regarding CAMP volcanism. On lines 247–248, it is stated that ‘only a few sill intrusions are observed within the Planorbis Zone’. This is inconsistent with current age estimates for CAMP basalts and the of the beginning of the Planorbis Zone. The latter is thought to have started very soon after the Jurassic as defined by P. Spelae (probably only 100 kyr at most, so about 201.2 Ma). And the youngest known CAMP lavas from North America and Morocco are dated to around 201.0 Ma, or even later (Whiteside et al., 2010, PNAS; Blackburn et al., 2013, Science). And that is just the lavas that we know about. CAMP is so poorly preserved that it is almost certain that there were other extrusive eruptions happening later (although maybe not much later). I’m not saying that we should assume extrusive CAMP eruptions were occurring throughout the Hettangian (some old $^{40}\text{Ar}/^{39}\text{Ar}$ studies do suggest that, but no recent ones), but they certainly did continue for a few hundred thousand years after the TJ boundary and Spelae CIE. Of course, that can’t explain the spore mutations later in the Hettangian, but it should be considered more for the episodes during the mass extinction interval and immediately afterwards. At the moment, the authors don’t really seem to have a reason for ruling out volcanic UV-B destruction other than ‘volcanism wasn’t happening’, but if eruptions were taking place, another argument is needed. One thing to consider is the fact that studies from intrusive sills in the Amazonas Basin have suggested major halogen output at the start of the extinction event. If this were the case, yet few spore mutations were occurring at that time, it might support the authors’ argument better. Even then, are there any other ways that mutated spores could arise during times of major wildfire activity and runoff (like those that are recorded throughout the Hettangian part of the record)?

We are aware that our interpretations of environmental stressors are somewhat dependent on the position and extent of CAMP volcanism in relation the Schandelah-1 record. In order to clarify this, we have added a timescale in figure 3 that shows U-Pb dated CAMP intrusions and how they correlate to the Schandelah-1 core. This correlation is based on Lindström et al., 2017 and Lindström, 2021 which uses terrestrial palynology, carbon isotope stratigraphy and Hg records infer the position of dated CAMP intrusions (Blackburn et al., 2013; Davies et al., 2017; Heimdal et al., 2018). In addition, we constructed a new supplementary figure (4) that show the correlation between the $\delta^{13}\text{C}_{\text{TOC}}$ and Hg/TOC records of the sites depicted in figure 1. This provides more detail on the progression of CAMP phases in relation to palynofloral disturbances and extinctions. Based on these correlations, we infer that the oldest CAMP intrusions occur prior to the Marshi CIE, while the first major pulse of occurs roughly synchronous with the initial phase of terrestrial and marine extinction which is in many sites closely associated with the Marshi CIE (supplementary figure 4). The second phase of terrestrial (palynofloral) disturbance/extinction occurs in tandem with the Spelae CIE which correlates to a second pulse of volcanism according to the correlation presented in Lindström et al., 2021 and partly overlaps with the TJ boundary. Age estimates of the Marshi CIE range from 201.52 to 201.48 Ma, while the Spelae CIE occurs close to the TJ boundary at 201.36 Ma. The uncertainties of these age estimates and error margins, together with variable sedimentation rates, make it difficult to confidently state an age progression of the Schandelah-1 core, particularly in the Hettangian section. However, the position of palynofloral

disturbances/extinctions combined with the position of the negative CIEs and Hg/TOC anomalies provides sufficient tie points to map the progression of CAMP volcanism within the Schandelah-1 record. The position of the latter post-ETME CAMP intrusions is more confidently dated using U-Pb ages to 201.1 to 200.9 Ma which roughly correlates the Planorbis ammonite zone. Various studies such as Lindström et al., 2021 suggest that the $^{40}\text{Ar}/^{39}\text{Ar}$ dates for these youngest intrusions are unreliable and give ages much younger than indicated by U-Pb ages. Additionally, many CAMP intrusions/extrusions are eroded and likely give an incomplete progression. Therefore, we aim to rely further on the geochemical records (detailed in Lindström et al., 2021) to determine the presence of magmatic activity. Lindström et al., 2021 concludes no major CAMP pulses following 201.2 Ma in Europe, Greenland and North America. Based on a holistic assessment of the palynological records (detailed in Bos et al., 2023) with geochemical record (presented in this study), we are confident in asserting the terrestrial (weathering) origin of the Hettangian Hg/TOC anomalies. See response to reviewer #1 for details on our palynological assessments.

Regarding potential mutagenic stressors, we fully acknowledge that increased UV-B radiation due to ozone depletion could have played a contributing role in the formation of malformed spores. However, we do suggest that this scenario falls short of explaining the prolonged presence of *in situ* spore malformation in the Hettangian of Schandelah-1, as detailed in the manuscript. In order to break down the ozone layer sufficiently to increase UV-B to levels that would induce metagenesis (at least in conifer pollen) requires major volcanic injection of halocarbons into the stratosphere according to experimental results of Benca et al., 2018. This scenario might apply to the earlier CAMP pulse that is associated with the Marshi CIE (Heimdal et al., 2018), which is thought to have produced vast amounts of halocarbon emission due to volcanic intrusions into carbonate and evaporite source rocks (Heimdal et al., 2018). The response in palynofloral records of Schandelah-1 depicts a higher abundance of pollen tetrads of *Ricciisporites tuberculatus*, although it is not clear if this a malformation signal. Spore malformations are not prevalent in this interval only showing a minor peak directly following the Marshi CIE, suggesting that ferns did not respond to increased levels of UV-B radiation (if these were actually happening) or were perhaps shielded by high canopy vegetation. Benca et al., 2018 suggested that increased UV-B radiation could have had a more severe impact on upper canopy conifer vegetation during the end-Permian extinction, while simultaneously shielding vegetation in the undergrowth and understory (where most ferns reside). The first true spore malformation spike is synonymous with the Spelae CIE that also exhibits a clear Hg/TOC anomaly linked to gaseous volcanic injection (Hg-isotopes). Subsequent spore malformation increases correlate with more Hg/TOC anomalies in the Hettangian section following a periodic pattern. Combined, these records hint that climate-driven processes are responsible for observed fern mutagenesis, rather than volcanically forced processes such as ozone depletion. Heat stress due to wildfire activity potentially also contributed to the formation of spore malformation, as stated in text, although there exist little empirical evidence for this. The effects of Hg pollution are better constrained and an obvious candidate to explain mutagenesis in ferns as recorded in spore malformation abundance intervals.

Finally, there are a few places where the authors need to be careful with their stratigraphic descriptions. On lines 277–278, it is stated that ‘many global sites only record a single prominent Hg-anomaly at the Spelae CIE.’ But in the case of Kuhjoch (Austria) and New York Canyon (Nevada, USA), two sites with some of the best-studied ammonite and $\delta^{13}\text{C}$ stratigraphy available, the first Hg peak is very clearly beginning below the appearance of P. Spelae, at the Marshi CIE. Which begs the question, why isn’t a peak occurring correlative with the assumed Marshi CIE at Schandelah-1? There are other examples of stratigraphic

uncertainties too. For example, on lines 130–132 a maximum duration of 450 kyr is given for the time between the Marshi and Spelae CIEs, but that is quite a lot longer than any estimate I've seen recently, assuming that the extinction begins at 201.6–201.5 Ma and the TJ boundary is at 201.3 Ma. On another occasion, the 'Spelae Event' is referred to, but it isn't clear how long the authors are proposing this to be, or if it is synonymous with the 'Spelae CIE'. None of these necessarily cause major issues for the main results of this study, but some tidying up and clarification of terminology is warranted (more details below).

In order to address this issue of stratigraphy and Hg/TOC anomalies, we have constructed a new supplementary figure (4) to show the proposed correlation (see previous responses). In this new supplementary figure we have replotted most Hg/TOC data that excludes low TOC values (<0.15) in order to get a clear image of Hg-enrichments that are not forced by a drop in TOC concentrations. Not all sites depict two clear Hg/TOC anomalies, although the Kuhjoch section indeed has an anomaly present synchronous with the Marshi CIE (See supp. Figure 4 for details on the carbon isotope stratigraphy), this is only reflected in most other sections as a bulk Hg anomaly. The reason for this is not clear and could be due to mediation by carbon and/or sulphur burial in these intervals. We have a few sentences in text to address this issue (L. 279-282). The maximum range of 450 kyrs refers to the total uncertainty in age estimations between the Marshi and Spelae CIEs, partly based on astronomically tuned sections (Lindström et al., 2017). The duration of this periods could be much shorter to about 150 kyrs as you mentioned. We have changes to depict the range between 450 to 150 kyrs (L. 135). We also changed "Spelae Event" to "Spelae CIE" and "Spelae anomaly" to "Spelae Hg-anomaly", in order to make this clearer.

Other comments:

L. 30–33: 'focusing Hg-pollution in shallow marine basins.' Well, at least in this area, we don't know about elsewhere outside of the central European region.

L. 30-33: We agree. We have changed the text to reflect this notion

L. 46: Jost et al. (2017, PNAS) and Somlyay et al. (2023, EPSL).

L. 45-46: We added Jost et al., 2017 as an extra reference for this statement

L. 56: 2 ppm is a very high concentration of mercury to occur in a natural setting, which would require very severe Hg pollution. How realistic is this for the Triassic–Jurassic?

L. 56-57: See previous response on the factors that determine toxicity (concentration and speciation). This statement covers what previous experiments have reported but does not suggest this concentrations in required to induce plant DNA damage at the TJ boundary. The paragraph goes on to say that speciation of Hg is an important understudied factor.

L. 61: Percival et al. (2017, PNAS) did not study China. Shen et al. (2022b, Nature Communications) and Zhang et al. (2022, Frontiers in Ecology and Evolution) have done.

L. 62-63: This is true. We have reformulated the sentence and cited the correct references to reflect this.

L. 67: I feel as though something is missing from this sentence. Why does the susceptibility to volatilization make Hg more likely to have a long enough residence time to be globally distributed? There are other gases that do not. I'm also not convinced that the Lindberg et al. (2002, Environmental Science and Technology) paper on the Arctic is the best work to cite here, especially given the comparative lack of polar snow and ice during the Triassic and Early Jurassic. The 2009 volume by Pironne and Mason (Mercury Fate and Transport in the Global Atmosphere) could be a good starting point.

L. 66-67: We understand the confusion. It is the susceptibility to volatilization in combination with the long atmospheric residence time that gives Hg^0 its potential to be distributed globally. We have reformulated this sentence to reflect this statement more accurately. In addition, we agree that the 2009 volume by Pironne and Mason is a more applicable reference here for a connection with the Triassic-Jurassic world.

L. 78–80: Presumably this is only applicable to spore-producing plants, rather than all plants? If spore producers were more common in the Triassic–Jurassic, clarify this for the non-expert.
L/ 78-80: Stomatal uptake as the main pathway of Hg in vegetation is true for nearly all plants. We are most interested in spore-producing vegetation (ferns and fern allies), but we changed it to “reproductive organs” to include a larger variety of vegetation. The discussion part further lays out why we think this is more apparent in fern-type vegetation.

L. 92–93: On line 88 it is stated that ‘Hg-MDF is found in nearly all kinetically-controlled reactions’. If this is the case, how can Hg-MDF reliably indicate changes in terrestrial Hg-absorption and the state of the terrestrial biosphere, when there are so many other potential influences?

L. 92-93: This true. That partly the reason why this study and many others focus more on MIF of Hg-isotopes. However, we can still utilize MDF in combination with MIF to interpret the most likely scenario of Hg-speciation and sourcing, but we need to be more careful and not assess MDF only.

L. 101–102: Atmospheric Hg is itself characterized by slightly negative MIF (at least today) due in part to the photochemical reduction of mercury, with the mercury featuring positive MIF removed through wet deposition and the residual gaseous Hg inventory left more negative. So plants take up atmospheric mercury that is already characterized by slightly negative MIF, and make it more so.

The range of MIF in atmospheric Hg can be slightly negative but also exhibit positive values. Terrestrial Hg (soils and vegetation) is significantly more negative than the atmosphere (-0.6 to -0.2). To make this point more clearly, we changed the wording to say that terrestrial Hg is highly negative compared to atmospheric sources.

L. 127–128: ‘the widely recorded Marshi (precursor) and Spelae (initial) CIEs’. Be careful with phrasing here. Most of these sites had CIEs recorded initially as the Initial and Main CIE (the Precursor CIE was identified at far fewer locations), which were then re-interpreted as the Marshi and Spelae CIEs.

L. 129-131: We agree that the wording here is not entirely accurate. We have reformulated to state the Marshi CIE is recorded near the start of the extinction (terrestrial and marine) and that the Spelae (initial) is more widely recorded and recognized.

Also, the Precursor and Initial CIEs were interpreted as being equivalent to the Marshi and Spelae CIEs at St Audries Bay, but at Kujoch the Initial CIE was relabelled Marshi and a separate CIE coeval with the TJ boundary was labelled the Spelae. So the conversion between the 2002 Hesselbo system and 2017 Lindström system is not uniform everywhere, and given that Kuhjoch is the GSSP, presumably the more sensible reference to make is with that site, not St Audries Bay.

L. 129-131: The correlation efforts by Lindström et al., 2017 incorporate palynological records that indicate several phases of extinction in relation to the position of the negative CIEs. This provides a more robust assessment of how the carbon isotope stratigraphy correlation to other sites, although there are still inconsistencies. The GSSP section at Kuhjoch displays several

difficulties for confident correlation that include gaps in the sedimentary record which were detailed in Bos et al., 2023 for initial correlation of the Schandelah-1 section to other sites. The St. Audries Bay section is more complete in that sense which is highlighted in the correlation scheme of Lindström et al., 2017.

L. 128–130: I know that the authors cite a previous manuscript here but I think more details are needed (at least in a sentence or two) about how an overlap with CAMP can be achieved at this site based on just CIEs and palynology.

L. 132-133: We feel strongly that previous results should not be overly expressed in this section, since two previous publications (van de Schootbrugge et al., 2019 and Bos et al., 2023) have confidently resolved the stratigraphy and position of terrestrial extinction. However, we have reformulated “palynological assemblages” to “palynofloral diversity disturbances” to clarify that the position of disturbed terrestrial biomes is used to infer the onset and progression of terrestrial extinction. As stated in text, this is combined with the negative CIEs on the Schandelah-1 record and based on previous correlative efforts, a temporal overlap of CAMP is inferred. These are not conclusions or results that we want to present here and therefore we are more comfortable to cite the studies that have concluded this.

L. 132: The TJ boundary has been well dated to no later than 201.3 Ma, based on dating of ash layers just above P. Spelae in Peru and Nevada. In the Fundy Basin of North America and Argana Basin of Morocco, CAMP magmas just above the extinction level have been dated to about 201.56 Ma. So assuming that the Marshi CIE marks the onset of this extinction and the CIE just below those CAMP lavas, the gap between the Marshi and Spelae CIEs should be no more than 250 kyr (and some estimates are shorter). So I’m not sure how the authors have reached a figure as high as 450 kyr. I’d suggest either to explain this, or (if it isn’t important for the manuscript) to remove it.

L. 133-135: We have added this line to provide a sense of time progression of the extinction, although there are still a lot of uncertainties. See response to one of the major concerns of reviewer #2 to read how we changed and further clarified this in the text.

L. 142: Is this duration of 1.3 Myr consistent with previous estimates of the Hettangian? I’ve seen different durations estimated.

L. 146-147: This duration is indeed the Hettangian that is present in the Schandelah-1 core. Most inferred durations of the Hettangian of other sections suggest roughly 2 millions years (some suggest longer, although this is not fully accepted). We have clarified that an unconformity is present at the top of the Angulatenton Fm in the Schandelah-1 core by adjusting figures 3 and 4.

L. 152–153: Technically the lack of disturbances in Sinemurian palynoflora shows stabilization of the vegetation, from which stabilization of the climate has been inferred, not the other way around as implied here.

L. 156-158: We agree. We have reformulated this sentence to reflect this.

L. 156: Why only normalizing against TOC, rather than against other possible host species such as sulfur or clays?

L. 161: Normalization of Hg against TOC is the most studied and well-known method of examining sedimentary Hg-enrichments. Although sulphur-rich minerals and clays can potentially absorb Hg, this occurs in a much lesser degree than bounding with organic material and therefore is preferred to assess Hg-dynamics in marine deposition settings.

L. 174–175: This is true, although from both going over the data in the supplementary tables I found that the average Hg concentration and the variations in them are just as high for the Sinemurian as for the Hettangian. The thing that has changed is that TOC levels are much higher in the Sinemurian. So has mercury cycling really changed, or is the signal just being overprinted by the excess TOC burial?

L. 187–181: This is true and we infer as much in the discussion section. A crucial point that we infer in the later sections of the discussion, is that a disturbed system with minimal TOC burial is unable to sequester Hg and provide a pathway for environmental Hg to be buried on longer timescales. Combined with an increase in the global budget of Hg (due to CAMP) this was the cause for an exaggerated Hg cycle due to the remobilization of Hg. A deepening of the regional basin, as inferred by the shift in lithology and palynology in Schandelah-1, promoted carbon burial and the sequestering of Hg in the larger region. However, we have decided to put less emphasis on this due to the fact that the Sinemurian strata of Schandelah-1 is partly incomplete and deposited under a different regime (deeper basin). See comments and responses to reviewer #3

L. 184: What is a fern ally?

L. 376–379: A fern ally is a commonly used term in (paleo)botany for non-fern spore-producing plants such as mosses, clubmosses and horsetails. However, based on comments of reviewer #3, we have decided to omit this paragraph for the result section and relocate part of this section to the discussion. The term fern ally is removed to avoid confusion. Instead, more emphasis has been put on the relationship between the S/P ratio and microcharcoal abundance in the Schandelah-1 core.

L. 190–191: The trends are similar, but not perfect. Charcoal is low but S/P high just above the Triassic-Jurassic boundary (310 m), whilst charcoal levels rise but S/P stays low between 300–290 m. So there could be more nuance here.

L. 376–381: This is true and is partly due to the fact that this record is based on microcharcoal (<30µm) which often underestimates charcoal abundance compared to bigger fractions (Hollaar et al., 2021). The pattern is therefore interpreted in more general terms and seems to largely fit. The nuance is discussed in more detail in Bos et al., 2023, but this study concluded the presence of more frequent wildfires in the spore spiked intervals in the Hettangian, while the TJ-Boundary interval is more complicated.

L. 194–199: The obvious devil's advocate question here is why the base of the extinction interval is below the biggest spike in plant turnover and malformation and the biggest d13C excursion (if we assume that the biggest negative excursion in d13C = the largest carbon cycle disruption)? Taking this line of reasoning further, could it be argued that the very large d13C negative at 318 m is the Marshi CIE and then the Spelae CIE is immediately above it between 316–311 m? In the absence of ammonite biostratigraphy I assume that CIEs have been assigned based on correlations with other sites (as laid out in Bos et al., 2023, Global and Planetary Change). This is more of a curiosity question about the stratigraphy than a major issue with the manuscript, but I'd be keen to understand the rationale better.

See new supplementary figure 4: We have created a new figure (supplementary figure 4) to address the issue of correlation of between carbon isotope excursion and how this was established through the position of clear and widely traceable palynofloral turnovers. Assuming these palynofloral turnovers occurred synchronously across NW Europe, we are confident that this is the correct progression extinction phases.

L. 204: Why are these occurrences of Type III, Type IV, or Type V not shown in Figure 4, or in a supplementary Figure?

We have plotted mainly Type-I malformations in figure 4, because this type makes up the largest fraction of the total malformations by a great margin. The other malformation types occur more frequently in intervals of increase malformation abundance, but this is not discernable when plotted as a fraction in figure 4. We feel that the total abundance of malformations is more relevant due to the fact we do not know the individual causes for each malformation type. Therefore, we express the frequency of the other type malformations in text, rather than visually and provide supplementary data for examination.

L. 245–249: ‘two phases of extrusive volcanism that ended following the Spelae event’. What is meant by the Spelae event? Is this synonymous with the CIE at the TJ boundary? The youngest CAMP lavas are from about 201.0 Ma, which is well in to the Hettangian after the boundary at 201.3 Ma. Also, it would be better to cite the original U-Pb references on line 246.

L. 243-246: According to the CAMP stratigraphy overview presented by Lindström et al., 2021, the youngest lava flows associated with CAMP and the ETME are of the Preakness Basalt of which U-Pb dating indicate an age of 201.274 ± 0.032 Ma (Blackburn et al., 2013). Another prominent extrusive phase is from the Northern Mountain Basalt and is dated to 201.566 ± 0.031 Ma (Blackburn et al., 2013). Based on further work by Heimdal et al., 2018 in the Amazon Basin, which is supplemented by the palynological record correlation of Lindström et al., 2017, these ages roughly correlate to the Marshi and Spelae CIEs. Details on this correlation are discussed in the relevant literature. We have changed some citations in this sentence to better reflect the current literature.

Following on from this, the sentence ‘This post-ETME intrusive phase was likely restricted to passive and diffusive emissions unable to significantly affect the stratosphere, where the ozone layer resides.’ is highly speculative. We know that there were definitely Hettangian lavas, and given how poorly CAMP is preserved, if there are some surviving lavas then it is not unreasonable to assume that there could have been extensive extrusive volcanism. In any case, we know very little about the dynamics of LIP magmas and how they could have impacted the stratosphere. Indeed, it has been modelled that effusive eruptions could have ejected gas emissions high enough to potentially reach the lower stratosphere (see Schmidt et al., 2016, Nature Geosciences).

L. 246-249: Similar to the previous response, the youngest CAMP basalts have been U-Pb dated to 200.916 ± 0.064 Ma (Butner intrusive; Blackburn et al., 2013) and 201.111 ± 0.071 Ma (Foum-Zguid dyke; Davies et al., 2017) and are definitely intrusive. Ar/Ar dating suggest much younger ages into the Sinemurian (195-192 Ma) but are considered to be unreliable and not cross-checked with U-Pb ages (Lindström et al., 2021). It is however likely that lavas did persist into the Hettangian and were eroded and are non-detectable. It could therefore be argued that a combination of diffusive degassing and potential lava output was able to deplete the ozone layer during the Hettangian and contribute to the mutagenetic appearance of spores. However, a lack of evidence for long and persistent volcanic activity during the entirety of the Hettangian, combined with the punctuated and periodic nature of the Hg and spore malformations, strongly hints towards orbitally-driven climate changes to be main cause. In addition, ozone recovery can occur relatively quickly (tens to hundred years).

L. 267–269: So why is there no attempt to normalize against these other potential host phases? Especially given that on line 259, adsorption on to S-rich minerals is also mentioned. See earlier point.

L. 265-269: See response to earlier comment (line 156). While S-rich mineral adsorption might have occurred in the lowermost section of the Spelae CIE (spike in TOC), this was likely very restricted and did not overall affect the bulk Hg and Hg/TOC signature.

L. 271–275: See also the model of Sanei et al. (2012, *Geology*), which invokes a similar model regarding the balance of TOC and S drawdown of Hg being disrupted at a time of increased volcanic influx (in the context of the Siberian Traps and Permian extinction). Might be worth thinking about.

L. 268-274: Similar to previous response. This was considered and might have occurred in the lowermost interval of the Spelae CIE. However, this does not explain the overall signature of Bulk Hg and Hg/TOC. We have cited Sanei et al., 2012 as a potential explanation.

Also are the several sites referred to as having euxinic conditions around the Spelae CIE all in this area of central Europe? Or distributed globally? Clarify.

L. 270-274: Based on a new study in *Nature Geoscience* (Bond et al., 2023) that uses Mo-isotopes and which includes data from the Schandelah-1 site, we infer severe localized euxinic conditions at this interval. We have changed the text to reflect this new data.

L. 275–277: Again, be careful with phrasing here. Firstly, the Hg peaks at several sites (certainly from the Thibodeau et al., 2016; Percival et al., 2017; and Kovacs et al., 2020 papers) were correlated against the Initial and/or Main CIEs, and as noted above, conversion to the Marshi/Spelae CIEs is not always straightforward. But in any case, at both Kuhjoch and New York Canyon, the Hg peak begins at the Marshi CIE (see Lindström et al., 2021, *Earth Science Reviews*), not the Spelae. And there are some sites where there is only one clear $\delta^{13}\text{C}$ excursion, which makes it harder to work out which level (and any correlative Hg spike) is equivalent to (e.g., Kovacs et al., 2020, *Global and Planetary Change*; De Graaff et al., 2022, *PPP*).

L. 275-290: We agree. We have omitted these sentences from the manuscript and provided a more detailed explanation of Hg degassing related to CAMP intrusions in text. In addition, we refer to the newly constructed supplementary figure 4 for details on correlation of carbon isotope stratigraphy and Hg/TOC records.

L. 282–283: ‘assumed absence’.

L. 293: Done

L. 286–288: Looking at the supplementary data tables, many of the samples have fairly low HI values (often under 50), and relatively high OI (often over 100, sometimes well over 100). Whilst the fact that the organic matter is primarily terrestrial does make low HI and higher OI more likely, some of these OI values are very high and does suggest some oxidation of the organic matter throughout. Is there any petrographic evidence to support this? If so, even if there is no direct correlation between Hg and HI or OI, it could still hint that the record is potentially oxidized, which can complicate interpretation of Hg and Hg/TOC trends.

L. 298-307: We have no petrographic data, although we do know all the organic matter in Schandelah-1 is immature (Bos et al., 2023). Whilst we acknowledge the fact that oxidation might have influenced the expression of the Hg/TOC, a holistic assessment that includes reworked palynomorphs, microcharcoal and lithology strongly hints to increased terrestrial organic matter to be delivering terrestrial Hg to the marine realm which is clearly reflected in the Hg-isotope data. The overall picture seems to suggest that increased amounts of Hg was being transported to the basin during times of increased wildfires and weathering, which could

have been oxidized and cause sharp increases in the Hg/TOC expression. We have reformulated the text to address this.

L. 306–318: This section would really benefit from some quantitative modelling to estimate the flux of Hg available for runoff and/or stomatal uptake. Fendley et al. (2019, EPSL) reconstructed the Hg cycle and volcanic output during the KPg boundary by utilizing the Hg proxy record and box model of Amos et al. (2015, Environmental Science and Technology). Could something similar be attempted, even roughly, here to at least test whether the degree of Hg pollution required to cause spore mutation is consistent with the recorded Hg and Hg/TOC peaks? I acknowledge that this could be quite a big ask, but given the potential impact of the conclusion and journal, having some quantification would really help to strengthen the arguments presented.

L. 308-324: We have discussed this with Isabel Fendley (co-author on this manuscript) in order to quantify the Hg pollution and potential concentrations. However, this requires a good control on the sedimentation rates of the Schandelah-1 core which is variable in many intervals and makes this exercise extremely uncertain. In addition, if we want to infer terrestrial Hg concentrations from marine records, we need to do another large set of assumption that would ultimately defeat the purpose of trying to assess the potential concentrations that could induce fern mutagenesis. See response to major concerns of reviewer #2 for more details on why this is impractical and beyond the scope of this manuscript.

L. 321: ‘assumed to be near-zero MIF’. Yes, assuming that the Hg emissions were magmatically sourced rather than from heating of sediments by intrusive magmas (see Shen et al., 2022a, Nature Communications).

L. 332-336: Shen et al., 2022 indeed suggested that emissions from magmatic intrusions display negative MIF. Coals on average display slightly negative MIF (-0.2‰ to 0‰) and therefore are more positive compared to vegetation-derived Hg reservoirs. Subsurface coal-burning could produce slightly negative (total gaseous Hg) to slightly positive (rainfall) MIF signature as demonstrated by measurements near coal-fired powerplants (Sherman et al., 2012). It stands to reason, that volcanic activity and subsurface heating of coal/shale beds both produce a near-zero to slightly negative MIF signature which would still cause a positive shifts compared to the highly negative MIF values derived from vegetation-derived Hg reservoirs.

L. 324: I wouldn’t describe -0.5 to -0.3 as ‘minimal variability’. It may be less variability than in the rest of the Schandelah-1 record, but it’s still quite a lot and is equal to or in some cases more variability than in other TJ records (see compilation in Shen et al., 2022a), especially given the relative paucity of data points. It’s also worth noting that there is only one data point from below the proposed Marshi CIE and extinction interval, so not really a pre-CAMP background.

L. 325-326: We agree. We have reformulated this sentence to reflect this.

L. 333: The word ‘volcanic’ isn’t really needed before ‘Hg-input’ here. The low Hg concentrations presumably indicate a low input of mercury overall, regardless of actual source(s).

L. 330: We agree. We have removed the word “volcanic”.

L. 345: Give the low MIF values for these coal deposits.

L. 352: Done

L. 349–351: Re-emphasize any other evidence that independently supports this to strengthen the argument here.

L. 358-362: We have re-emphasized this statement by invoking the increased abundance of reworked palynomorphs to indicate a higher fraction of mineral Hg to be transported to the shallow marine basin. Increased abundances of reworked palynomorphs indicates that soil was partly stripped from the forest floors and resulted in increasing exposed bedrock area that were susceptible to weathering and erosion.

L. 429–430: It is worth reminding the reader here that a similar scenario has been suggested for the marine realm as well (see Thibodeau et al., 2016), with CAMP potentially delaying the recovery of calcareous fauna and allowing an initial dominance of siliceous organisms post-extinction (although not for as long).

L. 427-429: While we are aware of this, we feel that the message of this manuscript should focus on the progression of extinction and recovery in the terrestrial realm. Earlier versions of this manuscript incorporated the works on marine recovery after the extinction. However, we feel this distracts too much from the main points that we are trying to make.

Figures 3 and 4: I find the illustrated history of CAMP (showing an extrusive phase at the start with only intrusive pulses at the end) odd. The oldest known CAMP magmas are intrusive bodies from Guinea and Bolivia (Davies et al., 2017, Nature Communications), and as noted in earlier comments, there are later extrusives from well into the Hettangian.

See adjustments figures 3 and 4: As noted in earlier responses, the CAMP stratigraphy and correlation to the Schandelah-1 record is based on the position of palynological turnovers (Lindström et al., 2017; Lindström et al., 2021). The oldest known CAMP intrusions are indeed dated to ~201.6 Ma, while the youngest intrusions are dated to ~201.1 (Foum Zguid dyke, Morocco; Davies et al., 2017) and ~200.9 Ma (Butner intrusive, N. America; Blackburn et al., 2013). Dated CAMP lava flows are derived from the Preakness Basalt (~201.27 Ma; Blackburn et al., 2013) from N. America and Northern Mountain Basalt (201.56 to 201.49 Ma; Blackburn et al., 2013 and Davies et al., 2017). The youngest of these roughly correlates to the TJ-Boundary that is often associated with the Spelae CIE and the second phase of terrestrial extinction (Lindström et al., 2017; 2019 and Lindström, 2021). The correlation with the Stenlille record in Davies et al., 2017 is considered to be incorrect based on further palynological examination that were presented in Lindström et al., 2019. We have added CAMP stratigraphy in figure 3 with age estimation to clarify the correlation used in this manuscript.

General point on figures: Given that the authors have an age model for this site, it would be nice to have a figure plotting $\delta^{13}C$, Hg, palynology, CAMP magmatism vs time in Ma.

See adjustment figure 3: We have added CAMP stratigraphy in figure 3 with age estimation to clarify the correlation used in this manuscript. However, age estimations for the Hettangian/Sinemurian section of Schandelah-1 core is more difficult to assess since sedimentation rates are not constant. Therefore, we are more comfortable to present the core in the depth domain and report the ammonite and palynomorph zones for age correlations to other sites.

References:

Blackburn, T.J., Olsen, P.E., Bowring, S.A., McLean, N.M., Kent, D.V., Puffer, J., McHone, G., Rasbury, E.T. and Et-Touhami, M., 2013. Zircon U-Pb geochronology links the end-Triassic extinction with the Central Atlantic Magmatic Province. *Science*, 340(6135), pp.941-945. <https://doi.org/10.1126/science.1234204>.

Davies, J.H.F.L., Marzoli, A., Bertrand, H., Youbi, N., Ernesto, M. and Schaltegger, U., 2017. End-Triassic mass extinction started by intrusive CAMP activity. *Nature communications*, 8(1), 15596. <https://doi.org/10.1038/ncomms15596>.

De Graaff, S.J., Percival, L.M., Kaskes, P., Déhais, T., De Winter, N.J., Jansen, M.N., Smit, J., Sinnesael, M., Vellekoop, J., Sato, H. and Ishikawa, A., 2022. Geochemical records of the end-Triassic Crisis preserved in a deep marine section of the Budva Basin, Dinarides, Montenegro. *Palaeogeography, Palaeoclimatology, Palaeoecology*, 606, 111250, <https://doi.org/10.1016/j.palaeo.2022.111250>.

Fendley, I.M., Mittal, T., Sprain, C.J., Marvin-DiPasquale, M., Tobin, T.S. and Renne, P.R., 2019. Constraints on the volume and rate of Deccan Traps flood basalt eruptions using a combination of high-resolution terrestrial mercury records and geochemical box models. *Earth and Planetary Science Letters*, 524, 115721, <https://doi.org/10.1016/j.epsl.2019.115721>.

Hesselbo, S.P., Robinson, S.A., Surlyk, F. and Piasecki, S., 2002. Terrestrial and marine extinction at the Triassic-Jurassic boundary synchronized with major carbon-cycle perturbation: A link to initiation of massive volcanism?. *Geology*, 30(3), pp.251-254, [https://doi.org/10.1130/0091-7613\(2002\)030<0251:TAMEAT>2.0.CO;2](https://doi.org/10.1130/0091-7613(2002)030<0251:TAMEAT>2.0.CO;2).

Jost, A.B., Bachan, A., van De Schootbrugge, B., Lau, K.V., Weaver, K.L., Maher, K. and Payne, J.L., 2017. Uranium isotope evidence for an expansion of marine anoxia during the end-Triassic extinction. *Geochemistry, Geophysics, Geosystems*, 18(8), pp.3093-3108, <https://doi.org/10.1002/2017GC006941>.

Kovács, E.B., Ruhl, M., Demény, A., Fórizs, I., Hegyi, I., Horváth-Kostka, Z.R., Móricz, F., Vallner, Z. and Pálfi, J., 2020. Mercury anomalies and carbon isotope excursions in the western Tethyan Csóvár section support the link between CAMP volcanism and the end-Triassic extinction. *Global and Planetary Change*, 194, 103291, <https://doi.org/10.1016/j.gloplacha.2020.103291>.

Lindberg, S.E., Brooks, S., Lin, C.J., Scott, K.J., Landis, M.S., Stevens, R.K., Goodsite, M. and Richter, A., 2002. Dynamic oxidation of gaseous mercury in the Arctic troposphere at polar sunrise. *Environmental science & technology*, 36(6), pp.1245-1256, <https://doi.org/10.1021/es0111941>.

Lindström, S., van De Schootbrugge, B., Hansen, K.H., Pedersen, G.K., Alsen, P., Thibault, N., Dybkjær, K., Bjerrum, C.J. and Nielsen, L.H., 2017. A new correlation of Triassic–Jurassic boundary successions in NW Europe, Nevada and Peru, and the Central Atlantic Magmatic Province: a time-line for the end-Triassic mass extinction. *Palaeogeography, Palaeoclimatology, Palaeoecology*, 478, pp.80-102, <https://doi.org/10.1016/j.palaeo.2016.12.025>.

Lindström, S., Callegaro, S., Davies, J., Tegner, C., van De Schootbrugge, B., Pedersen, G.K., Youbi, N., Sanei, H. and Marzoli, A., 2021. Tracing volcanic emissions from the Central Atlantic Magmatic Province in the sedimentary record. *Earth-Science Reviews*, 212, 103444, <https://doi.org/10.1016/j.earscirev.2020.103444>.

Percival, L.M., Ruhl, M., Hesselbo, S.P., Jenkyns, H.C., Mather, T.A. and Whiteside, J.H., 2017. Mercury evidence for pulsed volcanism during the end-Triassic mass extinction. *Proceedings of the National Academy of Sciences*, 114(30), pp.7929-7934, <https://doi.org/10.1073/pnas.1705378114>.

Pironne, N. and Mason, R. (Eds.), *Mercury Fate and Transport in the Global Atmosphere: Emissions, Measurements and Models*. Springer, Dordrecht, The Netherlands, 2009.

Sanei, H., Grasby, S.E. and Beauchamp, B., 2012. Latest Permian mercury anomalies. *Geology*, 40(1), pp.63-66, <https://doi.org/10.1130/G32596.1>.

Shen, J., Yin, R., Algeo, T.J., Svensen, H.H. and Schoepfer, S.D., 2022. Mercury evidence for combustion of organic-rich sediments during the end-Triassic crisis. *Nature Communications*, 13(1), 1307, <https://doi.org/10.1038/s41467-022-28891-8>.

Shen, J., Yin, R., Zhang, S., Algeo, T.J., Bottjer, D.J., Yu, J., Xu, G., Penman, D., Wang, Y., Li, L. and Shi, X., 2022b. Intensified continental chemical weathering and carbon-cycle perturbations linked to volcanism during the Triassic–Jurassic transition. *Nature Communications*, 13(1), 299, <https://doi.org/10.1038/s41467-022-27965-x>.

Somlyay, A., Palcsu, L., Kiss, G.I., Clarkson, M.O., Kovács, E.B., Vallner, Z., Zajzon, N. and Pálffy, J., 2023. Uranium isotope evidence for extensive seafloor anoxia after the end-Triassic mass extinction. *Earth and Planetary Science Letters*, 614, 118190, <https://doi.org/10.1016/j.epsl.2023.118190>.

Thibodeau, A.M., Ritterbush, K., Yager, J.A., West, A.J., Ibarra, Y., Bottjer, D.J., Berelson, W.M., Bergquist, B.A. and Corsetti, F.A., 2016. Mercury anomalies and the timing of biotic recovery following the end-Triassic mass extinction. *Nature Communications*, 7(1), 11147, <https://doi.org/10.1038/ncomms11147>.

Whiteside, J.H., Olsen, P.E., Eglinton, T., Brookfield, M.E. and Sambrotto, R.N., 2010. Compound-specific carbon isotopes from Earth's largest flood basalt eruptions directly linked to the end-Triassic mass extinction. *Proceedings of the National Academy of Sciences*, 107(15), pp.6721-6725, <https://doi.org/10.1073/pnas.1001706107>.

Zhang, P., Lu, J., Yang, M., Bond, D.P., Greene, S.E., Liu, L., Zhang, Y., Wang, Y., Wang, Z., Li, S. and Shao, L., 2022. Volcanically-induced environmental and floral changes across the Triassic-Jurassic (TJ) transition. *Frontiers in Ecology and Evolution*, 10, <https://doi.org/10.3389/fevo.2022.853404>.

Reviewer #3:

Climate-forced Hg-remobilization driving mutagenesis in ferns in the aftermath of the end-Triassic extinction

Remco Bos, Wang Zheng, Sofie Lindström, Hamed Sanei, Irene Waajen, Isabel Fendley, Tamsin A. Mather, Yang Wang, Jan Rohovec, Tomáš Navrátil, Appy Sluijs, Bas van de Schootbrugge

Dear authors,

I read the manuscript with great interest. The work presents new insights into the environmental dynamics in the aftermath of large-scale volcanism. The study is the first to trace mercury (Hg) concentrations and isotopes through the Lower Jurassic Hettangian stage in a stratigraphic framework and resolution high enough to interpret shifts in relation to orbital cycles, and in combination with data on the mutagenesis in ferns. The Hg concentrations and Hg isotopes show that orbitally-paced modulation of the climate leads to the remobilization and emission of initially volcanically sourced Hg over a time period of (at least) 2 million years following volcanic emission. The study further shows that the recycling of Hg into the environment causes the repeated increase in the abundance of mutagenic fern spores. The findings are of interest to a wide range of readers from the fields of earth science, paleoclimate and paleoenvironments.

Overall, the study presents a robust data set, which is reported, presented and discussed in detail. In parts, the text is hard to follow for readers that are not familiar with Hg isotope data, and the discussion of the data should be extended to certain aspects, also in regards to published literature (for details see below).

I recommend the study for publication after minor revision.

For the revision, I suggest to address the following concerns/suggestions:

- Hg concentrations and isotope data obtained from the Triassic–Jurassic (T–J) boundary interval should be discussed in the context of recent studies providing similar data sets from the same time interval from various settings (e.g., [e.g., 1, 2, 3]). Sections on which Hg data is available for the T–J boundary are shown in Figure 1, but there is little discussion in how your data compare to the published data and data interpretation.

We agree that discussion on contemporaneous sites regarding similar data is under developed. Therefore, we have constructed a new correlation figure (supplementary figure 4) to clearly depict the correlation between the Hg/TOC records across Europe. In addition, we added a new section in the discussion (L. 275-282) briefly comparing the Hg records from the different sites with reference to the newly constructed figure.

- The Hg enrichments and Hg isotope data of the T–J boundary interval should also be discussed in respect to thermogenic Hg generated through volatilization of sedimentary organic matter through CAMP-related igneous sills

We agree this is of importance to the overall assessment of Hg-sources and Hg-isotope signature. Subsurface coal-burning due to magmatic sill intrusions are a likely event to have occurred during CAMP-volcanism and potentially provided an additional Hg-source during Hg-enrichments. Hg-isotope studies of coal-fired plant emissions have demonstrated slight positive MIF and highly negative MDF. Therefore, the sudden and rapid negative shift at the Spelae CIE could be an indication of surface coal-burning. Have added a few sentences (L. 334-340) discussing this in the discussion section, although this is difficult to state as a certainty.

- I recommend to discuss the impact of wildfires on the Hg isotope values. Since the frequency of wildfires is modulated by orbital parameters (e.g., [4]), this should be especially important for the Hettangian data.

We agree with the importance to incorporate the effects of wildfire activity on the Hg-isotope signature and the potential release of stored Hg to the atmosphere. A recent study (Richter et al., 2023) demonstrated no significant mass-independent fractionation (MIF) in wildfire Hg-emissions, while mass-dependent fractionation (MDF) displays more significant variation. Since our assessment mostly focusses on MIF to trace Hg-sources and pathways, we are confident that wildfires had no significant impact on the MIF signature. MDF is influenced by many more processes and is therefore less reliable as stated in the main text. We have added a few sentences (L. 379-385) in the main text to address these issues and contextualize them in the appropriate section.

- With the orbital pacing of the climate and subsequent environmental shifts being the main mechanism driving the Hg remobilization in the Hettangian, I recommend to put more emphasis on this aspect, also in the visual presentation of your data. You refer to the eccentricity pacing of other data as shown by [5], and present a correlation of the bandpass filter of the dominant 26.5 m dominant spectral peak (which are interpreted to represent long eccentricity cycles) to the Hg concentration data in Figure 3. You also present a model illustrating the mechanisms of Hg mobilization and isotope fractionation in response to eccentricity forcing (Figure 6). Correlation between eccentricity pacing of the Hg isotopes shifts and abundance of malformed spores is not presented but referred to in the text. I recommend to visualize this aspect more clearly in figure 4 and add some information in the text additional to referring to [5].

See new figures 4 and 6: We agree with this assessment and in order to put more emphasis on hydrological regime shifts due to orbital variation, we have made major revisions to several figures. Firstly, we have added a secular variation curve in figure 4 that visualizes the periods of enhanced hydrological cycles based on previous studies (Bos et al., 2023) that correlates with the Hg-concentration and Hg-isotope records. In addition, we have completely revised figure 6 that visualizes an environmental model for Hg-mobilization. Here we have similarly put more emphasis on the contrast between eccentricity minima and maxima and the effect on the hydrological cycle and the resulting pathways of Hg-mobilization.

- The naming of Marshi CIE (referring to the precursor CIE) and Spelae CIE (referring to the initial CIE) is somewhat confusing when looking at the stratigraphic position of these shifts in other section with respect to the last occurrence of *C. marshi* and first occurrence *P. Spelae* (e.g. [6, 7]). In the Kuhjoch section, for example, the initial CIE corresponds to the LwO of *C. marshi*, while the first occurrence of *P. spelae* marks the main CIE [6]. I recommend to show a correlation in the supplements that correlates the isotope shifts recorded in Schandelah-1 to a section with better (bio) stratigraphic control in this stratigraphic interval.

See new supplementary figure 4: We agree that the naming of the carbon isotope stages can be construed as confusing. In order to clarify this, we have constructed a new supplementary figure (4) that correlates the carbon isotope stages to the sites mentioned in figure 1. This clearly shows the position of the isotope excursion in relation to other sites and extinction events in both the terrestrial and marine realms. In addition, we have made several in text references to this figure at the appropriate sections to further clarify this issue.

- HI and OI data are not mentioned in the results section but discussed in the discussion part. It is not clear if the data are from previously published work or new data of this study.

Hydrogen Index (HI) and Oxygen Index (OI) data set was originally presented in Bos et al. (2023) published in *Global and Planetary Change*. We have added a citation at the appropriated section in text (L. 297) to clarify the origin of this data set.

- The data resolution is not sufficient to make statements about whether the eccentricity-paced Hg mobilization had ceased in the Sinemurian, considering the time covered by the Arietenton and Obtususton Formations.

We agree that the data-resolution is not sufficient to state that eccentricity-paced Hg dynamics had ceased during the early Sinemurian. However, we don't make this direct statement in text. We have clarified in the text that the Sinemurian palynofloral conditions seem to have stabilized with no significant orbital pacing (based in Bos et al., 2023) and that this may have led to Hg-mediation through enhanced carbon burial and soil sequestering (L. 408-412).

Minor comments:

Abstract:

Line 27: change to lower Jurassic Hettangian stage (provides more context for the reader that is not familiar with the Jurassic stages since you refer to 'early Jurassic peaks' later in the text)
L. 27: Done

Line 28 – change 'spikes' to increase
L. 29: Done

Line 29 – volcanic source of the Hg enrichment
L. 29-30: Done

Line 33 – do you mean forcing?

L. 32: If you mean instead of "forced", then we prefer the term "eccentricity-forced phases" do denote the orbital influence on the hydrological cycle.

Introduction:

Line 40–42: mention that the negative carbon-isotope excursion indicate the release of isotopically-light carbon into the ocean-atmosphere system, otherwise the context is not clear
L.40-43: Done

Line 45–46: perhaps link the nutrient flux to the increased weathering rates for context
L. 45-47: Done

Line 55–56: mention that the experimental studies were performed on plants
L. 56-57: Done

Line 56: concentrations of $>10 \mu\text{M}$ (2 ppm) Hg
L. 57: Done

Line 65: add reference for the hypothesized release of Hg
L. 66: Done

Line 69–75: Hg isotopes are not my expertise and I find the introduction of the Hg isotope system hard to follow in parts. Does the deposition via rainfall and and/or particle fallout refer

to both Hg²⁺ and Hg_p? Otherwise, how does the Hg²⁺ enters the environment that contains sulfate- and/or iron-producing bacteria?

L. 70-71: Both Hg²⁺ and Hg_p are deposited via rainfall and particle fallout. We have changed the sentencing to reflect this notion more accurately.

Line 75: explain briefly what ligand-bound means

L. 77: Done

Line 79: gaseous uptake of Hg⁰ is

L. 78: Done

Line 82–83: Make clearer that Hg-speciation across the ETME has been presented in previous studies, and the new approach of this study is the Hg-speciation throughout the Hettangian and lower Sinemurian to assess the long-term effects.

L. 82-84: Done

Line 85: A shift Shifts in

L. 85: Done

Line 87: refer to methods section here for details on δ²⁰²Hg and Δ¹⁹⁹Hg

L. 87-88: Done

Line 88: low depleted

L. 89: Done

Line 92: often can

L. 92: Done

Line 92: what does ‘state of terrestrial biosphere’ refer to?

L. 92-93: This is indeed confusing. We have changed it to “Hg-absorption in terrestrial biosphere” to state that terrestrial vegetation and microbial activity mainly influences MDF in coastal and shallow marine sediments

Line 92–104: This paragraph is very difficult to follow. First, the MDF isotope effect of stomatal Hg uptake in foliage is discussed, then MIF is discussed, and at the end the photochemical reaction of MDF is mentioned. For MIF, the photochemical reactions are mentioned as a key role in MIF variability. That photochemical reduction causes positive shifts is only at the very end of the paragraph. I think this can be re-structured in a way that is easier to follow.

L. 85-108: We have restructured this paragraph and separated the discussion on MDF in the first half and then shift towards discussion on MIF. This separates the two main Hg-isotopic records based on the processes that influence them.

Also, be consistent with either, e.g., Hg-MDF of just MDF.

We agree. We have changed the text to consistently use MDF and MIF

Line 116–117: this could be more specific with what you mean with ‘a shift to gaseous forms of Hg’. Do you mean re-mobilization through continental erosion and subsequent shift to gaseous Hg through photochemical reaction?

L. 120-122: While that statement is correct for the early Jurassic (Hettangian) section of our record, the volcanic anomaly at the T-J boundary is also driven by atmospheric gaseous Hg and not by re-mobilization. We have changed the statement in this section to the hypothesis that increase amounts of gaseous Hg in the environment would results in phytotoxic consequences which covers both the T-J boundary event and Hettangian re-mobilization/photochemical reduction phases.

Results:

Line 125: cross-correlated cross-correlated

L. 128: Done

Line 126–128:check references. The Marshi (precursor) CIE is not discussed in [7] and not explicitly associated with carbon injection in [8]. Please cite a reference that clearly defines the stratigraphic position of these two CIEs, and defines the Marshi CIE as precursor, and the Spelae CIE as initial CIE. Ideally present a correlation in the supplements. As far as I am aware, the LwO of *P. spelae* marks the T-J boundary, but the initial CIE (which is here called Spelae CIE) correlates with the last occurrence of *C. marshi* [6]

L. 129-132: We have changed the reference to Lindström et al., 2017 which introduces a new correlation between European T-J boundary section based on carbon isotope stratigraphy (naming the Marshi and Spelae CIEs) and on palynological records. We have summarized and expanded upon this correlation in a new figure (supplementary figure 4) that depicts the position of the negative CIEs and their correlation to Hg and Hg/TOC anomalies.

Line 128–130: Also refer to [9], who are cited in [5] referring to the palynological assemblage

L. 133: Done

Line 132–133: Ref. [10] correlate the onset of the extinction interval globally. If the regression is limited to the NW European realm the sentence needs to be restructured.

L. 136-138: We have restructured the sentence to refer to the last occurrence of the ammonite species *Choristoceras marshi* to be the horizon of onset of extinction in NW Europe and that this is associated with a negative CIE and transgression. The purpose of this sentence is to provide background on stratigraphic correlation of the regional structure of the ETME.

Line 138: Give reference for the transgression and state how you identify the transgression in the Schandelah-1 core. Also refer to the [5, 9] regarding the $\delta^{13}\text{C}$ and TOC data in text, and give reference for TOC data in Fig. 3

L. 138: We have added a reference for the presence of a transgression at the Spelae CIE and specified that this is noted in the Schandelah-1 core by an increase in TOC and abundances in aquatic palynomorphs with reference to Bos et al., 2023 and van de Schootbrugge et al., 2019.

Line 140: often widely

L. 145: Done

Line 142: Make clear that the duration of ~1.3 Ma is referring to the Hettangian succession preserved in the Schandelah-1 core, and not to the Hettangian stage in general. Due to the unconformities at the base and top of the Angulatenton Fm, the cyclostratigraphy performed on the Schandelah-1 data did not capture the entire stage (as mentioned in [5])

L. 145-147: Done

Line 145: be more specific about what you refer to with the word ‘disturbance’

L. 150: This sentence suggests that the intervals with dominant number of spore-taxa in the overall palynofloral assemblage is indicative of terrestrial disturbance. We have changed terrestrial disturbance of palynofloral disturbance to better reflect this issue.

Line 148: How do you define ‘pre-extinction’ $\delta^{13}\text{C}$ values? In the Schandelah-1 record, there are only 5 $\delta^{13}\text{C}$ values from the strata preceding the Marshi CIE.

We agree that this is difficult to establish from the $\delta^{13}\text{C}$ and TOC data. We have omitted this sentence from the manuscript and focus more on the palynofloral changes in the Sinemurian of Schandelah-1

Line 152: What kind of disturbances are you referring to?

L. 156: This refers to changes in biodiversity of the total palynofloral assemblage as noted in Bos et al., 2023. We have changed the terminology to “No significant biodiversity disturbances have been recorded”.

Hg trends and anomalies

Line 162: this would be easier to follow if the stratigraphic position of the Spelae CIE/anomaly was more precisely defined in the above text

L. 142-144: We have clarified this in text and provide reference to figure 3 show the expression and position of the Spelae CIE and the Hg-anomaly associated with this.

Line 169: I am not sure if you can distinguish between a high and a low anomaly

L. 174: The purpose of this sentence is the note that the lower TOC in these intervals, results in more pronounced Hg/TOC anomalies. We have changed “relatively higher” to “more pronounced”

Line 178–179: comma missing

L. 184: Done

Palynofloral disturbance and spore malformations:

Line 182–191: this paragraph is mainly summarizing the findings of [5], which should not be part of the results section

We agree that this summarized section from data of Bos et al., 2023 does not belong in the Result section. Therefore, we have compressed this paragraph to a few summarizing sentences and added it to the Discussion section.

Line 201–204: For consistency, if you mention that malformation Type-1 is linked to abortion and premature shedding of non-viable spores, you should mention what the other malformation types are linked to (if known). This could be placed in the supplements

There is still a lot not known about the exact causes for these spore malformations. Although some research, mainly derived from Lindström et al., 2019 (and reference therein) elaborate somewhat on this topic. We don’t want to put too much emphasis on the potential impairments of spore development because this remains largely speculative. We have summarized the speculative ideas on the developmental impairments of each malformation type in the Materials and Methods section.

Line 204: Unless you clearly link the four distinct peaks in spore malformation abundance to the long-eccentricity cycles identified in TOC and the CIEs you should refer to them as reoccurring rather than periodic

L. 202: We do link the spore malformation increases to the eccentricity-paced cycles in Schandelah-1, although we confidently make arguments for this in the discussion. Therefore we have taken your suggestion and changed “periodic” for “reoccurring” in this sentence.

Line 208–211: refer to data table in the supplements

L. 207: Done

Line 211: lower lowermost Sinemurian

L. 207-208: Done

Hg-isotopes:

Line 219: following the sharp negative shift (...), average \square 199Hg values remain a -0.18% within the Hettangian

L. 217-218: Done

Line 221: periodic reoccurring/repetitive

L. 219: Done

Line 222: anomalies in Hg concentrations

L. 220: Done

Line 222–223: the main feature in the Sinemurian seems to be a broad negative shift in \square 199Hg

L. 220-221: Done

Line 223: place discussion on \square 202Hg in new paragraph

L. 222: Done

Line 223–225: there is a word missing in this sentence.

L. 222: We have noticed. We changed the sentence to “Rhaetian $\delta^{202}\text{Hg}$ values average at -1.55% ”

Line 223–226: You call the Triletes Beds the main extinction interval, but Fig. 3 and 4 mark the ETME for the interval of the Contorta beds to Pilonoten Sst, which makes this section hard to follow. Also, check the ranges given here. The higher values preceding the negative Spelae shift seem to range from -1.60% to -1.01% .

L. 222-227: We have omitted the term “main extinction interval” from this section. Terminology regarding the ETME assigns the terms “main extinction” often to the interval equivalent to the Triletes Beds. However, this has been noted in recent times as a misnomer. Instead, “main disturbance” is suggested by Wignall & Atkinson, 2020. This is also dependent on the marine versus terrestrial extinction records. In addition, we have changed the ranges in MDF for this section to accurately reflect the measurements.

Line 227: anomalies in Hg concentrations

L. 226: Done

Discussion:

Line 232–233: label the Marshi and Spelae CIEs as nCIEs in Figure 3 and 4, or refer to the Marshi and Spelae negative CIEs in text

L. 230-231: We agree. We have changed the negative CIE in text to Spelae CIE for clarity.

Line 234–235: Be more specific about the similarities. Is it the timing, or the severity of malformation?

L. 232-233: We agree. We have restructured the sentence to refer to the similarities in both the malformation morphologies and abundances.

Line 240–241: better refer to greenhouse gas emissions as it includes other carbon sources such as thermogenic methane

L. 239: Done

Line 242–245: I suggest to rephrase this sentence, suggesting that the prolonged and repeated mutagenesis in fern communities suggests additional mechanisms acting on longer time scales

L. 240-243: While this is true, the effect of additional mechanisms is more comprehensively explained in the following sentences, linking this to the potential accumulation of remobilized Hg in the terrestrial environment (particularly low-lying coastal margins). Therefore, we feel this is expressed sufficiently in this section.

Line 265: carbon burial did not play a significant role in Hg enrichment

L. 263: Done

Line 271: a short-lived peak in TOC is not sufficient evidence for a transgression event

L. 269: This is true. We have added the presence of an influx of aquatic palynomorphs to this sentence as evidence for a transgression at this horizon. The data is derived from Bos et al., 2023 and is cited in the correct place.

Line 273–275: please be more specific what you refer to as initial rise in bulk Hg concentrations. Do you refer to the interval that correlates with increased TOC? It could also be useful to separate the two intervals in the TOC/Hg cross-plot (Figure S2B). How do you exclude that the decoupling between Hg and TOC (meaning more Hg than it would be expected if it was linked increased organic matter drawdown) is not caused by volcanic input? This would be consistent with the MIF data.

The increase of Hg-burial through increased S-binding is a local acting mechanisms and inter-basin correlation is not sufficient proof. Can you provide any sedimentological or geochemical evidence that suggest photic zone euxinia in the Schandelah-1 core?

L. 270-274: We agree with this assessment. Luckily, since submitting this paper, a new study was published in Nature Geoscience that presented Molybdenum isotope data from several sites including Schandelah-1 (Bond et al., 2023). This study revealed the presence of euxinic conditions at the horizon in question. We have added a sentence and reference to explain this. In addition, we have more clearly formulated the presence of euxinia at this horizon by stating the presence of a short-lived peak in isorenieratane in the Mariental core which is closely located to the Schandelah-1 core. This is also consistent with the MIF data which shows one measurement which displays a highly negative value. We have also changed supplementary figure 2 to depict the drawdown and burial processes more accurately.

Line 280: add that these extrusive phases might be reflected in the Hg-enrichment intervals to bring this into context

L. 275-290: We added a new paragraph that adds more detail on the CAMP phases and how this might have generated Hg at different quantities and how this might be reflected in the marine sedimentary record.

Line 281: Spelae Hg anomaly (to indicate that the Hg enrichment was not driven by increased OM deposition and/or preservation)

L. 291: Done

Line 282: cyclic repetitive/reoccurring nature

L. 292: Done

Line 287–288: It is not clear whether the HI and OI data are part of this study (mentioned in methods section, but not discussed in the results section), or if this is published data from [5], in which case you need to add the reference and remove the respective part in the methods section

L. 297: We have added the reference that originally published this data set (Bos et al., 2023) and removed the detail in the Methods in favour of a reference

Line 287: how do you distinguish post-depositional from primary (e.g. water column) oxidation? Do you have Tmax data as maturity indicator? What would be the effects of thermal alteration on Hg concentrations and Hg isotopes?

L. 303-307: Based on the data from Bos et al., 2023, all samples were thermally immature and didn't experience any thermal alteration. We have added this fact to the sentence. In addition, we have changed the phrasing to suggest that oxidation in general had a limited effect on the expression of the Hg/TOC record.

Line 290–291: 'halted paleo-redox front' needs a bit more explanation

L. 301: We have changed this to "paleo-redox fronts in the sediment" to state that oxidation could have potentially focused Hg in the sediment, although wholistic assessment of the organic matter suggests that this is limited and would not have significantly altered the expression of the Hg-isotope record.

Line 292–296: specify if this is true for all shifts in Hg/TOC or if you refer to the Hg-enrichments in the Hettangian interval

L. 305-307: We specifically refer to the Hg anomalies in the Hettangian section. We have clarified this in text

Line 311–318: shifted to its gaseous form would magnify the mutagenic potency of Hg on vegetation, but it would also decrease the amount of Hg that is transported and accumulated in shallow marine environments. Regarding gaseous Hg, I am wondering about the effect of fires on the release of Hg, also in relation to eccentricity pacing (see [4] for eccentricity forcing of wild fires)

L. 324: We have removed this section from the manuscript, since this is also discussed later on the manuscript referring to the new environmental model. We have put more emphasis on the potential effects of wildfires on Hg distribution, isotopes and potential mutagenic effects on fern communities.

Line 319: Spelae Hg anomaly

L. 330: Done

Line 324: $\delta^{199}\text{H}$ values range from -0.50‰ to -0.31‰ according to results section, and were described as highly negative

L. 325-326: We changed this sentence to $\Delta^{199}\text{Hg}$ values show highly negative variability (-0.50‰ to -0.30‰) consistent with a plant material source"

Line 325–326: it would be useful to mention in the introduction of the Hg isotope system that negative MIF in vegetation is attributed to photochemical reduction (loss of Hg) in foliage
L. 102-104: We agree with this assessment. We have moved this sentence to the introduction.

Line 328–329: Positive shift in MDF in the Hettangian Triplets beds preceding the Spelae CIE (makes it easier to follow)
L. 328: We have added “preceding the Spelae CIE” to this sentence, but Hettangian is not correct for this interval.

Also, does the reference indicate that this is the interpretation of another study, or are you referring to the processes? Can you add more context on the microbial and abiotic processes that you are referring to?

This study refers to a large review article that summarizes all processes and pathways of Hg speciation alteration and isotope signatures. The overall idea is that degradation of bound-Hg is causing the positive MDF in the record.

Line 329–332: This is really hard to follow. Do you mean that due to low abundance of vegetation the uptake by soil increases, where it is degraded and produced positive MDF? Also, wouldn't a low abundance of vegetation cover lead to higher photochemical reduction in soils? We agree. In re-assessment of the data, we have decided to omit this line of reasoning from the manuscript due to the fact that many other processes can influence MDF which makes it hard to interpret in this interval. We choose to follow the line of reasoning in the previous comment that over degradation of bound-Hg is likely responsible for the overall MDF signature. The lack of evidence for photochemical reduction (positive MIF) might be related to the deposition regime for the Triletes Beds which is governed by a sandy deltaic/fluvial deposition, although this is unclear.

Line 319–335: the paragraph on MDF and MIF signals associated with the T–J transition and volcanism is missing discussion of published data from the same interval. There are a number of Hg-concentration records and Hg-isotope records from different sites [e.g., 1, 2, 3]. Do the data show the same trends and lead to similar conclusions? The paragraph is also missing discussion regarding Hg release from combustion of organic-rich sediments
L. 338-340: This is accurate. We have added a few sentences discussing the Hg-isotope signature of St Audries Bay which displays a similar pattern compared to the Schandelah-1 record.

Line 341: specify mineral sources

L. 349: Mineral sources overall refer to Hg stored in bedrock. We have clarified this in text

Line 345: this might not be the right reference there. But since coal is plant material, a negative MIF would be expected

L. 351-352: This is the correct reference for the Hg-isotope signature in coals deposits which was discussed in Biswas et al., 2008 for account for coal-fired plant emission in forest systems. This statement was originally derived from Blum et al., 2014. Coal shows in average more positive MIF values (~-0.2‰) compared to plant material which is likely due to a mixed plant/mineral source.

It could also be mentioned that ref. [5] show that the positive CIEs in the Hettangian (that correspond to increased mutagenesis) also correspond to the occurrence of reworked palynomorphs, implying increased reworking in this interval

L. 358-387: We make this argument in the following paragraphs. This particular paragraph summarizes potential influences on MIF which are later substantiated by the presence of reworked palynomorphs.

Line 377–378: the periodic nature of what?

L. 373-387: We have changed this section to better reflect the observed changes in the Hettangian record of Schandelah-1 (i.e. charcoal and magnetic susceptibility) and how this is linked to changes in the hydrological regime modulated by the long-eccentricity cycle.

Line 380–381: This needs a bit more detail. The fire activity is mentioned here for the first time.

Additional to the photochemical reduction, I am wondering if fires would not lead to a volatilization of Hg, and what impact the fire would have on the Hg-isotope signature (see, e.g., [11]).

The orbital pacing of wild fire events (e.g., [4]) seems to be an important aspect here that should be mentioned here, also in correlation with the charcoal abundance presented in [5].

See new figure 6: As stated in the previous comment, we have adapted this line of reasoning and provide more detail on the effects of wildfire activity and the potential consequences on Hg dynamics and Hg-isotope signature. This is also adopted in the major revised environmental model of figure 6.

Line 384: cyclic reoccurring

L. 383: Done

Line 389: specify what extinction you refer to (marine?)

L. 388: We have omitted this phrase in order to focus more on the palynofloral disturbance and spore malformations at the Spelae CIE.

Line 308–499: This plot shown in Fig. 5D appears as important to the overall story and may need a bit more detail. It would be useful to indicate which data corresponds to the Hettangian Hg-isotope anomalies

See new figure 5; L. 325-387: We have added an extra layer of detail to this figure by clearly marking the difference between Hettangian Hg background conditions and Hg-anomalies. Several references to this figure have been added in text as well.

Line 399–404: Here you mention orbital pacing and refer to Fig. 6B, which shows the scenarios of eccentricity minima and maxima, and in the following text you briefly discuss the impact of orbital eccentricity forcing on the vegetation and Hg mobilization and degassing. The climate forcing in the fern mutagenesis appears as one of the main findings of the study (according to the title of the manuscript, and the abstract). For the importance this aspect of orbital forcing is giving in the beginning of the manuscript, the discussion as relatively short. In Figure 3 you show the correlation of Hg-concentration anomalies and eccentricity maxima. It would be useful to also show the correlation of the spore malformation and Hg-isotope data to the eccentricity. The climatic effects of eccentricity minima and maxima need a reference, e.g., [12].

L. 388-412; See new figure 6: Most of the consequences of orbital forcing as record in the Schandelah-1 records have been detailed in Bos et al., 2023 and is referenced throughout the

manuscript. We have emphasized the importance of the consequences of eccentricity minima and maxima by adding new visualizations in figure 4 that shows the position of enhanced hydrological cycle (orbitally forced). This is further emphasized in the new and revised figure 6 that shows the contrasting conditions between eccentricity minima and maxima which is again based on the works of Bos et al., 2023.

Line 411–413: specify ‘disturbed conditions’

I think the resolution of the presented data is not sufficient for stating that the orbitally-paced Hg release has ceased in the Sinemurian. The time interval that is covered by the Sinemurian strata (bucklandi to mid-obtusum Zone) may cover some 4 Myr (see [13]). The amount of data points for both Hg concentrations and isotopes may not be sufficient to identify eccentricity cycles. Furthermore, considerable amounts of strata are missing in the Schandelah-1 core due to a hiatus (indicated by missing turneri Zone)[9].

L. 408-412; See figure 3: We have specified disturbed conditions as “Palynofloral diversity disturbances”. We do agree that the record of the Sinemurian is not suited for this type of assessment due the presence of unconformities. We have added the position of potential unconformities in the figures and have put less emphasis on the Sinemurian in the text. We specify that the Sinemurian does not display any signs of palynofloral diversity disturbance or Hg-anomalies and that this impact might have ceased during these times. Uncertainty remains, however.

Line 412: The time scale presented in this study refers to 1.3 Million years based on [5]. The duration of 2 million years for the Hettangian is mentioned here for the first time and needs a reference

L. 416-417: This is true. Therefore we have reformulated to “Hg-mobilization continued during the Hettangian under extreme greenhouse climate conditions” with uncertainty for when these conditions ceased completely.

Line 418: this is the first time that extreme greenhouse conditions are mentioned. Needs reference and some context.

L. 379-381: We agree. We have given more explanation in the discussion that suggest that the major swings in the hydrological cycle were exacerbated by extreme greenhouse conditions following the ETME and CAMP emissions. This interpretation was initially reported by Bos et al., 2023 based on data from the Schandelah-1 core and has been cited in the proper position.

Line 420–421: It needs to be pointed out that the volcanic sourcing of Hg during the T–J transition is not a new finding but was already observed in previous studies

L. 418-419: We agree. We added that this finding is consistent with other studies.

Line 414–430: the concluding paragraph should mention the eccentricity pacing to draw a full circle back to the abstract and bring the climatic shifts in context

L. 419-422: We agree. We have added that the increases in wildfire activity and weathering are due to eccentricity-paced changes in the hydrological cycle.

Materials and methods:

Line 475–488: is all organic carbon data is not discussed in the results section. Please specify if the data (or some of the data) are part of this study, or refer to reference

L. 476-483: We have reduced this section due to the fact that most organic carbon data was reported in Bos et al., 2023. We have made the proper adjustments to this section and cited the correct studies that reported important data that is used in this manuscript.

Figures:

Fig. 1: The sections containing Hg concentration data and Hg isotope data shown in the figure are mentioned in the introduction, but are not discussed in relation to your data

See new supplementary figure 4: We have added discussion points comparing the Hg records between these sites and constructed a new supplementary figure (4) that compare the Hg/TOC records.

Fig. 4: Ammonite Zonation is missing in the stratigraphic column. The ammonite zonation is crucial in order to correlate the data to other sections.

See new figure 4: We have added the ammonite zones in this figure.

I hope this review is fair and the comments constructive.

Kind regards,

Marisa Storm

1. Thibodeau, et al. (2016) - Mercury anomalies and the timing of biotic recovery following the end-Triassic mass extinction. *Nature Communications*
2. Yager, et al. (2021) - Mercury contents and isotope ratios from diverse depositional environments across the Triassic–Jurassic Boundary: Towards a more robust mercury proxy for large igneous province magmatism. *Earth-Science Reviews*
3. Shen, et al. (2022) - Mercury evidence for combustion of organic-rich sediments during the end-Triassic crisis. *Nature Communications*
4. Hollaar, et al. (2021) - Wildfire activity enhanced during phases of maximum orbital eccentricity and precessional forcing in the Early Jurassic. *Communications Earth & Environment*
5. Bos, et al. (2023) - Triassic-Jurassic vegetation response to carbon cycle perturbations and climate change. *Global and Planetary Change*
6. Ruhl, et al. (2009) - Triassic–Jurassic organic carbon isotope stratigraphy of key sections in the western Tethys realm (Austria). *Earth and Planetary Science Letters*
7. Hesselbo, et al. (2002) - Terrestrial and marine extinction at the Triassic-Jurassic boundary synchronized with major carbon-cycle perturbation: A link to initiation of massive volcanism? *Geology*
8. Lindström, et al. (2012) - No causal link between terrestrial ecosystem change and methane release during the end-Triassic mass extinction. *Geology*
9. van de Schootbrugge, et al. (2018) - The Schandelah Scientific Drilling Project: A 25-million year record of Early Jurassic palaeo- environmental change from northern Germany. *Newsletters on Stratigraphy*
10. Lindström, et al. (2017) - A new correlation of Triassic–Jurassic boundary successions in NW Europe, Nevada and Peru, and the Central Atlantic Magmatic Province: A time-line for the end-Triassic mass extinction. *Palaeogeography, Palaeoclimatology, Palaeoecology*
11. Richter, et al. (2023) - Impact of forest fire on the mercury stable isotope composition in litter and soil in the Amazon. *Chemosphere*
12. Martinez, et al. (2015) - Orbital pacing of carbon fluxes by a ~9-My eccentricity cycle during the Mesozoic. *Proceedings of the National Academy of Sciences*
13. Storm, et al. (2020) - Orbital pacing and secular evolution of the Early Jurassic carbon cycle. *Proceedings of the National Academy of Sciences*

References used in responses:

1. Benca, J.P., Duijnste, I.A. and Looy, C.V., 2018. UV-B–induced forest sterility: Implications of ozone shield failure in Earth’s largest extinction. *Science Advances*, 4(2): p.e1700618.
2. Blackburn, T.J., Olsen, P.E., Bowring, S.A., McLean, N.M., Kent, D.V., Puffer, J., McHone, G., Rasbury, E.T. and Et-Touhami, M., 2013. Zircon U-Pb geochronology links the end-Triassic extinction with the Central Atlantic Magmatic Province. *Science*, 340(6135): 941-945.
3. Bond, A.D., Dickson, A.J., Ruhl, M., Bos, R. and van de Schootbrugge, B., 2023. Globally limited but severe shallow-shelf euxinia during the end-Triassic extinction. *Nature Geoscience*: 1-7.
4. Bos, R., Lindström, S., van Konijnenburg-van Cittert, H., Hilgen, F., Hollaar, T.P., Aalpoel, H., van der Weijst, C., Sanei, H., Rudra, A., Sluijs, A. and van de Schootbrugge, B., 2023. Triassic-Jurassic vegetation response to carbon cycle perturbations and climate change. *Global and Planetary Change*, 228: 104211.
5. Davies, J., Marzoli, A., Bertrand, H., Youbi, N., Ernesto, M. and Schaltegger, U., 2017. End-Triassic mass extinction started by intrusive CAMP activity. *Nat Commun*, 8: 15596.
6. Heimdal, T.H., Svensen, H.H., Ramezani, J., Iyer, K., Pereira, E., Rodrigues, R., Jones, M.T. and Callegaro, S., 2018. Large-scale sill emplacement in Brazil as a trigger for the end-Triassic crisis. *Scientific Reports*, 8(1): 1-12.
7. Hollaar, T.P., Baker, S.J., Hesselbo, S.P., Deconinck, J.-F., Mander, L., Ruhl, M. and Belcher, C.M., 2021. Wildfire activity enhanced during phases of maximum orbital eccentricity and precessional forcing in the Early Jurassic. *Communications Earth & Environment*, 2(1): 1-12.
8. Lindström, S., van de Schootbrugge, B., Hansen, K.H., Pedersen, G.K., Alsen, P., Thibault, N., Dybkjær, K., Bjerrum, C.J. and Nielsen, L.H., 2017. A new correlation of Triassic–Jurassic boundary successions in NW Europe, Nevada and Peru, and the Central Atlantic Magmatic Province: A time-line for the end-Triassic mass extinction. *Palaeogeography, Palaeoclimatology, Palaeoecology*, 478: 80-102.
9. Lindström, S., Sanei, H., Van De Schootbrugge, B., Pedersen, G.K., Leshner, C.E., Tegner, C., Heunisch, C., Dybkjær, K. and Outridge, P.M., 2019. Volcanic mercury and mutagenesis in land plants during the end-Triassic mass extinction. *Science Advances*, 5(10): eaaw4018.
10. Lindström, S., 2021. Two-phased Mass Rarity and Extinction in Land Plants During the End-Triassic Climate Crisis. *Frontiers in Earth Science*, 9(9): 1079.
11. Lindström, S., Callegaro, S., Davies, J., Tegner, C., van de Schootbrugge, B., Pedersen, G.K., Youbi, N., Sanei, H. and Marzoli, A., 2021. Tracing volcanic emissions from the

Central Atlantic Magmatic Province in the sedimentary record. *Earth-Science Reviews*, 212.

12. Richter, L., Amouroux, D., Tessier, E. and Fostier, A.H., 2023. Impact of forest fire on the mercury stable isotope composition in litter and soil in the Amazon. *Chemosphere*, 339: 139779.
13. Sherman, L.S., Blum, J.D., Keeler, G.J., Demers, J.D. and Dvonch, J.T., 2012. Investigation of local mercury deposition from a coal-fired power plant using mercury isotopes. *Environmental Science & Technology*, 46(1): 382-390.
14. van de Schootbrugge, B., Richoz, S., Pross, J., Luppold, F.W., Hunze, S., Wonik, T., Blau, J., Meister, C., van der Weijst, C.M.H., Suan, G., Fraguas, A., Fiebig, J., Herrle, J.O., Guex, J., Little, C.T.S., Wignall, P.B., Püttmann, W. and Oschmann, W., 2019. The Schandelah Scientific Drilling Project: A 25-million year record of Early Jurassic palaeo-environmental change from northern Germany. *Newsletters on Stratigraphy*, 52(3): 249-296.
15. Wignall, P.B. and Atkinson, J.W., 2020. A two-phase end-Triassic mass extinction. *Earth-Science Reviews*, 208: 103282.

REVIEWER COMMENTS

Reviewer #2 (Remarks to the Author):

This is the second time that I have reviewed this work by Bos et al. A number of my comments/queries on the first version have been addressed, or the reason why they don't need to be explained.

Reading through the revised manuscript, the one thing that gives me pause is that the authors outline a variety of possibilities outlined to explain the changes in recorded Hg MIF in the Hettangian, notably including increased photochemical reactions and changes in the type of runoff spanning mineral vs soil vs different kinds of organic matter. I agree with their assessment that the controls on Hg input to the northern German Basin are likely to have been complicated, and I think that the final sentence of the manuscript is actually a very nice summary and reflection of this (L. 427–429: 'Although environmental Hg dynamics over hundreds of thousand years are still unclear, our results point to the long-term implications of large-scale volcanism on terrestrial vegetation following major extinction.').

However, at times prior to that, it feels like the authors are pushing for the increased photochemical reduction following vegetation change as the main driver of Hettangian mutagenesis events and changes in MIF. Maybe this is possible, but personally, I found the idea of changes in runoff source (e.g., increased mineral vs leaf-litter soil) and possibly wildfires to be far more compelling. Wildfire activity definitely seems to have increased around the times of increased mutagenesis and Hg peaks in the Hettangian, given the increased abundance of charcoal, so could it have provided a major source of gaseous Hg?

Although on that note, the argument that gaseous Hg is the most important form for mutagenesis currently seems to be based on the mutagenesis spike and volcanic Hg influx during the Spelae CIE (L. 389–392), but since that time was unquestionably coeval with CAMP volcanism, this is the mutagenesis pulse where a non-Hg driver (particularly UV-exposure following ozone destruction) cannot be ruled out. So it may not be the best example to base a model of gaseous Hg being critical for mutagenesis on.

I accept the authors' point that wildfires may not have greatly affected the MIF signal, but changes in runoff source could have done (as is already stated). And both wildfires and runoff would surely have been impacted by orbital forcing. I'm not ruling out a potential role for increased photochemical reduction on land following the vegetation destruction, but for me, the other mechanisms feel just as likely. I think that the best route is to say that all of these processes were probably happening, and all contributing to the mutagenesis, Hg enrichment, and MIF variation to a greater or lesser extent, rather than trying to speculate on which process was the major driver of which observation.

Other comments:

L. 39–40: McElwain et al. (1999, Science) could also be cited here.

L. 43: 'and terrestrial realms'.

L. 46–47: I know it's cited later, but maybe Bond et al. (2023, Nature Geoscience) could already be referenced here.

L. 56–57: In my first review of the manuscript, I raised the point about whether it was possible to model whether the exposure to mercury could have reached the 2ppm concentration mentioned here, or if this concentration was realistic in nature. While the authors gave a very detailed response to my previous comment, I can't find any of the arguments utilised in that rebuttal stated here or in the following sentences. I think that many of the points made by the authors in their response are valid

(i.e., that the concentrations required to induce mutagenesis would vary depending on the plant, setting, and Hg species). But in this case, I would make this point that mutagenesis could be induced at lower concentrations under certain conditions at this point in the text. Or, alternatively, remove the 2ppm figure to avoid confusion. These nuances might be discussed later in the manuscript, but for me it would be better to either add a sentence here or remove the 2ppm value so as to avoid causing any confusion early on.

L. 61–63: Percival et al. (2017, PNAS) did indeed study Greenland and Morocco, but Argentina, not Peru. A Peruvian site was studied by Yager et al. (2021, Earth Science Reviews). And the Moroccan record studied by Percival featured only very small Hg peaks, stratigraphically below the oldest known CAMP lava.

L. 80–84: Be careful here. My first thought upon seeing the word 'speciation' was that the impact and concentrations of methylmercury vs inorganic mercury had been studied. Whereas this study (and the published ones cited here) are more focussed around conversion between Hg⁰ and other species (mainly Hg²⁺). I guess that is technically a discussion of Hg speciation, but it isn't automatically what came to mind. At the very least, whilst the isotopic data are used to infer a change in photochemical reduction of Hg (and therefore change in Hg speciation), the Hg species aren't being measured directly, which is more what comes to mind. I think 'Hg-cycling' would be a safer term.

On line 81, change 'particular' to 'particularly'.

Also, the last part of these sentences (on lines 83–84) read to me like the degassing is happening post eruption, which I don't think is what the authors mean. Maybe the following would be clearer 'however, the long-term consequences of volcanically-degassed Hg in the environment following the cessation of LIP eruptions remain elusive.'

L. 100–102: I fully agree with the authors on this point.

L. 111: 'unique' in what way? Clarify.

L. 119: Again, this read to me like the authors were directly studying concentrations of Hg⁰, Hg²⁺, or methylmercury, whereas what is actually done is that changes in these species are inferred from the isotope data. I suggest rewording to 'we reconstruct Hg cycling' to avoid this confusion.

L. 128: I'm not sure that 'calibrated' is the right word here. 'using' might be better.

L. 130–131: I accept the authors' logic and reasoning (as explained in the responses to my previous review) for correlating the CIEs with St Audries Bay rather than Kuhjoch, and having a new supplementary figure to show this is a good idea. But I think it would also be good to clarify this in the main text as well. It would only need a few words.

For example, lines 129–131 could be changed to:

'Two distinctive negative organic C-isotope excursions (CIE; Fig. 3A and supplementary Fig. 4) represent the Marshi and widely recorded Spelae CIEs (correlated with the Precursor and Initial excursions at St Audries Bay, UK), which reflect the volcanic injection of ¹³C-depleted carbon in relation to the exogenic pool.'

L. 137: Change 'that' to 'which'.

L. 144: Change to 'The remainder of the Spelae CIE...'

L. 161: In their responses to my previous comments, the authors explained why they chose to only

normalise Hg against TOC, and I have no issue with their response, which is entirely valid. But it would be nice to see this explanation stated in the manuscript, again just adding a few words to clarify that bonding of Hg with organic material is the most common relationship, and the best studied one, and that this is why it has been focussed on. This is to some extent referred to later on in the discussion, where the capacity of Hg to also bind with other species is stated, but I think it would be better to state that here instead, and deal with this issue up front.

L. 163–164: Clarify what the value of 1 standard deviation is (I assume that 48.6 ppb/wt% is the median value. This is key to indicate how far above the medium a value needs to get before it is considered anomalous.

L. 208–209: Is there any other supporting evidence for this? In the sedimentological properties, or other plant fossils etc?

L. 243–246: Care is needed here. Yes, U-Pb ages suggest this, but not all CAMP lava flows have been dated using U-Pb geochronology. So whilst it is true to say that the youngest CAMP pulse dated by U-Pb geochronology is around 201.2 Ma, there are CAMP lavas higher in the stratigraphy that are thought to be approximately 300–400 kyr younger than the Preakness basalt, based on cyclostratigraphy (e.g., Whiteside et al., 2010, PNAS; Lindström et al., 2021), as well as the intrusive bodies already referred to by the authors.

Of course, these lava flows may not have constituted a major magmatic pulse, but they are documented in both North America and Morocco, so there was some extrusive CAMP volcanism happening in the Planorbis, even if we don't know its precise age and extent. I think it would be fairer to say here that the evidence for most intense/significant CAMP volcanism is around the TJ boundary and just beforehand, with some indication of intrusive and extrusive magmatism in the early Hettangian, which was apparently less pronounced than the earlier activity, although the heavy erosion of the LIP makes this uncertain. But yes I fully agree with the authors that there is no clear evidence for CAMP activity in the middle–late Hettangian that can explain the repeated Hg cycle disturbances recorded throughout that stage at this site.

L. 264: I think that 'largely facilitated' would be better here.

L. 284: No comma is needed after CIE.

L. 289–290: A reference from modern-day environmental studies is needed to support this statement.

L. 293: Change 'unlikely driven' to 'unlikely to have been caused'.

L. 324: 'in those areas.'

L. 335–336: But would these values of MDF (or even MIF) vary from coal to coal? Worth adding this caveat.

L. 391–392: Possibly, but I think this is a little bit speculative, unless it has been documented by studies of modern environments. If this has been previously reported, then a reference is needed. If not, then yes perhaps the form of Hg (e.g., gaseous) is important, but is it necessarily about that or about the concentration of the Hg? Also since this is the Spelae CIE being discussed, when CAMP was definitely still active, this is the mutagenesis pulse where UV-forced mutagenesis can't be ruled out. So there are a few potentially complicating factors here.

L. 395–396: Is volatilisation from photo-chemical reduction necessarily needed to provide a source of Hg for mutagenesis (assuming that gaseous Hg is the crucial form for that to happen)? Could the increased wildfires have provided an adequate source alone? I get that they impact the MIF signature

a lot, but if a change in runoff is also happening (as stated earlier in the discussion) then that could also be playing a major role in MIF variability. To me, it feels like there are several drivers operating. Have there been any modern studies on the potential toxicity of photochemically reduced Hg produced in certain ecosystems (such as the salt marshes mentioned previously)?

L. 404: 'higher' would be better than 'high'.

L. 427–429: I really like this sentence as a nice conclusion.

Reviewer #3 (Remarks to the Author):

Review of revised manuscript 'Climate-forced Hg-remobilization driving mutagenesis in ferns in the aftermath of the end-Triassic extinction' by Bos et al.

The revised manuscript is in line with the reviewers suggestions, which have been addressed and implemented clearly.

- The presented data has been brought into context with previous studies on elemental Hg and Hg-isotopes. A new section was added to the discussion which compares Hg records from different sites. Additionally, a supplementary figure has been added presenting a stratigraphic correlation of published $\delta^{13}\text{C}$ and Hg data.
- A discussion of the Hg-isotope trends in induced by thermogenic volatilization of Hg from sedimentary organic matter has been added.
- The effects of wildfires on the Hg isotope signature and the release of Hg has been added to the discussion.
- The eccentricity pacing of the occurrence of malformed spores and Hg-isotope shifts is now clearly presented and explained in the text figures.
- The naming of the Marshi and Spelae CIE have been brought in stratigraphic context in relation to the alternative nomenclature
- Previously published OI and HI data have been removed from the results section and linked to the original study in the discussion
- The low resolution of the Sinemurian data is clearly pointed out and it is referred to the hiatus

The text reads well and the supporting figures are clear. However, reading the revised version of I came across the following issues:

The main concern regards the palynology and spore teratology methodology. You state that folded, broken, or obscured specimen were omitted from the malformed categories. Furthermore, spores with reworked features such as darkened and broken wall material were excluded from the total count of malformations (Line 442 – 446). At the same time you state that spore malformations typically show no features indicating reworking (i.e. darkened/broken wall material) and conclude that the record represents an in situ signal (Line 209 – 211). This reads like a systematically introduced bias. I suggest to revise the methodology section and add more detail on how reworking in malformed spores was detected, documented and assessed in relation to reworked non-malformed spores.

Minor comments:

Either the results or the methods section it needs a clear descriptions of where the studied sample some from. You mention the Schandelah-1 core in the introduction, and you mention the Lower Saxony Basin (line 138), but in the following text you refer to the northern German Basin, which represents the greater structural feature that, at least to my understanding, encompasses the Lower

Saxony Basin and the Danish Basin amongst other structures. I recommend to briefly mention that the studied samples are from the Schandelah-1 core drilled in the Lower Saxony basin, Germany, and in the following mention the structural relation to the German Basin, which then makes transparent how you extrapolate from the Schandelah-1 core to the wider area by correlation and comparison to the Danish Basin.

Line 246–247: the study by Benca et al. (2018) concerns U-VB radiation during the end-Permian. Add reference for the passive and diffusive emission postdating the end-Triassic mass extinction.

Line 282–283: add reference concerning the generation of Hg through volcanic intrusions and its dependency on the composition of intruded sedimentary rock. To my knowledge Wignall 2001 is not addressing this.

Line 377: define the term S/P ratio

Line 304–307: This section could need another backup that the malformed spores do not exhibit signs of reworking

I recommend the publication of the manuscript after minor revisions regarding the methods section.

Kind regards,

References:

Benca, J.P., Duijnste, I.A.P. and Looy, C.V., 2018. UV-B-induced forest sterility: Implications of ozone shield failure in Earth's largest extinction. *Science Advances* 4, e1700618, doi:10.1126/sciadv.1700618.

Wignall, Paul B. "Large igneous provinces and mass extinctions." *Earth-science reviews* 53.1-2 (2001): 1-33.

Reviewer #2 (Remarks to the Author):

This is the second time that I have reviewed this work by Bos et al. A number of my comments/queries on the first version have been addressed, or the reason why they don't need to be explained.

Reading through the revised manuscript, the one thing that gives me pause is that the authors outline a variety of possibilities outlined to explain the changes in recorded Hg MIF in the Hettangian, notably including increased photochemical reactions and changes in the type of runoff spanning mineral vs soil vs different kinds of organic matter. I agree with their assessment that the controls on Hg input to the northern German Basin are likely to have been complicated, and I think that the final sentence of the manuscript is actually a very nice summary and reflection of this (L. 427–429: 'Although environmental Hg dynamics over hundreds of thousand years are still unclear, our results point to the long-term implications of large-scale volcanism on terrestrial vegetation following major extinction.').

However, at times prior to that, it feels like the authors are pushing for the increased photochemical reduction following vegetation change as the main driver of Hettangian mutagenesis events and changes in MIF. Maybe this is possible, but personally, I found the idea of changes in runoff source (e.g., increased mineral vs leaf-litter soil) and possibly wildfires to be far more compelling. Wildfire activity definitely seems to have increased around the times of increased mutagenesis and Hg peaks in the Hettangian, given the increased abundance of charcoal, so could it have provided a major source of gaseous Hg? Although on that note, the argument that gaseous Hg is the most important form for mutagenesis currently seems to be based on the mutagenesis spike and volcanic Hg influx during the Spelae CIE (L. 389–392), but since that time was unquestionably coeval with CAMP volcanism, this is the mutagenesis pulse where a non-Hg driver (particularly UV-exposure following ozone destruction) cannot be ruled out. So it may not be the best example to base a model of gaseous Hg being critical for mutagenesis on.

Although it is true that increased UV-B radiation likely played a contributing role in the expression of malformed spores at the TJB (Spelae event), the common factor between the malformed spores of the Spelae event and Hettangian is elevated environmental Hg. We further use the argumentation that Hg is most easily incorporated in plant tissues via stomatal uptake and requires a gaseous Hg species. As explained in text, this circumvents root-protective systems and would require lower concentrations of Hg to be a mutagenic stressor. This argumentation is partly based on the assumption that there is a single underlying cause to the development of spore malformations that exhibit uniform traits in the Spelae event and the Hettangian. Since Hg contamination and toxicity is more likely to have been a disrupter in both the Spelae event and Hettangian stage, we argue that Hg-pollution is the obvious candidate for spore malformation development in ferns. We have added a few sentences in the penultimate paragraph to reiterate this line of argumentation. In addition, we have also slightly altered the title of the manuscript to reflect that Hg-pollution is strongly associated with fern mutagenesis. This acknowledges the fact that multiple factors potentially contributed to plant mutagenesis as well.

I accept the authors' point that wildfires may not have greatly affected the MIF signal, but changes in runoff source could have done (as is already stated). And both wildfires and runoff would surely have been impacted by orbital forcing. I'm not ruling out a potential role for increased photochemical reduction on land following the vegetation destruction, but for me, the other mechanisms feel just as likely. I think that the best route is to say that all of these

processes were probably happening, and all contributing to the mutagenesis, Hg enrichment, and MIF variation to a greater or lesser extent, rather than trying to speculate on which process was the major driver of which observation.

We fully agree with this statement. It is very difficult to disentangle all these processes since they all can produce positive shifts in Hg-MIF. The important point that we are trying to make is the concentration of large amounts of Hg that is mobilized through erosion, facilitated through increased wildfire activity and shifted towards a gaseous species that has a higher toxic potency to induce fern mutagenesis. To this end, we have changed the argumentation to include the potential of Hg-emissions of wildfires on fern mutagenesis and put less emphasis on the effects of photochemical reduction. We argue that a combination of these processes likely caused abnormal spore development and mutagenesis in fern communities.

Other comments:

L. 39–40: McElwain et al. (1999, Science) could also be cited here.

Done

L. 43: ‘and terrestrial realms’.

Done

L. 46–47: I know it’s cited later, but maybe Bond et al. (2023, Nature Geoscience) could already be referenced here.

Done

L. 56–57: In my first review of the manuscript, I raised the point about whether it was possible to model whether the exposure to mercury could have reached the 2ppm concentration mentioned here, or if this concentration was realistic in nature. While the authors gave a very detailed response to my previous comment, I can’t find any of the arguments utilised in that rebuttal stated here or in the following sentences. I think that many of the points made by the authors in their response are valid (i.e., that the concentrations required to induce mutagenesis would vary depending on the plant, setting, and Hg species). But in this case, I would make this point that mutagenesis could be induced at lower concentrations under certain conditions at this point in the text. Or, alternatively, remove the 2ppm figure to avoid confusion. These nuances might be discussed later in the manuscript, but for me it would be better to either add a sentence here or remove the 2ppm value so as to avoid causing any confusion early on.

We agree that this is confusing. Therefore, we have followed your advice and omitted this sentence from the manuscript. We believe that the nuance and the argumentation for the factors that determine Hg-toxicity is well covered in the discussion.

L. 61–63: Percival et al. (2017, PNAS) did indeed study Greenland and Morocco, but Argentina, not Peru. A Peruvian site was studied by Yager et al. (2021, Earth Science Reviews). And the Moroccan record studied by Percival featured only very small Hg peaks, stratigraphically below the oldest known CAMP lava.

Thank you for the clarification! We have changed this sentence to accurately reflect the sites that do record a prominent Hg-anomaly associated with the ETME and provided correct citations.

L. 80–84: Be careful here. My first thought upon seeing the word ‘speciation’ was that the

impact and concentrations of methylmercury vs inorganic mercury had been studied. Whereas this study (and the published ones cited here) are more focussed around conversion between Hg⁰ and other species (mainly Hg²⁺). I guess that is technically a discussion of Hg speciation, but it isn't automatically what came to mind. At the very least, whilst the isotopic data are used to infer a change in photochemical reduction of Hg (and therefore change in Hg speciation), the Hg species aren't being measured directly, which is more what comes to mind. I think 'Hg-cycling' would be a safer term.

We agree. We have changed Hg-speciation to Hg-cycling

On line 81, change 'particular' to 'particularly'.

Done

Also, the last part of these sentences (on lines 83–84) read to me like the degassing is happening post eruption, which I don't think is what the authors mean. Maybe the following would be clearer 'however, the long-term consequences of volcanically-degassed Hg in the environment following the cessation of LIP eruptions remain elusive.'

We agree. We have adopted the suggestion in text

L. 100–102: I fully agree with the authors on this point.

-

L. 111: 'unique' in what way? Clarify.

The depositional site of Schandelah-1 is unique in the sense that it contains a long record of coastal margin deposits, which allows for an assessment of changes in very shallow marine conditions. We have changed the sentence to "a unique coastal margin archive", to clarify our statement.

L. 119: Again, this read to me like the authors were directly studying concentrations of Hg⁰, Hg²⁺, or methylmercury, whereas what is actually done is that changes in these species are inferred from the isotope data. I suggest rewording to 'we reconstruct Hg cycling' to avoid this confusion.

We agree and have changed the sentence to reflect the comments above.

L. 128: I'm not sure that 'calibrated' is the right word here. 'using' might be better.

We agree. We have changed calibrated to using.

L. 130–131: I accept the authors' logic and reasoning (as explained in the responses to my previous review) for correlating the CIEs with St Audries Bay rather than Kuhjoch, and having a new supplementary figure to show this is a good idea. But I think it would also be good to clarify this in the main text as well. It would only need a few words.

For example, lines 129–131 could be changed to:

'Two distinctive negative organic C-isotope excursions (CIE; Fig. 3A and supplementary Fig. 4) represent the Marshi and widely recorded Spelae CIEs (correlated with the Precursor and Initial excursions at St Audries Bay, UK), which reflect the volcanic injection of ¹³C-depleted carbon in relation to the exogenic pool.'

We agree that this phrasing is more representative of the correlation that we present here. Therefore, we have adopted this sentence in text.

L. 137: Change 'that' to 'which'.

Done

L. 144: Change to ‘The remainder of the Spelae CIE...’.

Done

L. 161: In their responses to my previous comments, the authors explained why they chose to only normalise Hg against TOC, and I have no issue with their response, which is entirely valid. But it would be nice to see this explanation stated in the manuscript, again just adding a few words to clarify that bonding of Hg with organic material is the most common relationship, and the best studied one, and that this is why it has been focussed on. This is to some extent referred to later on in the discussion, where the capacity of Hg to also bind with other species is stated, but I think it would be better to state that here instead, and deal with this issue up front.

We agree. We have used our response to the first round of reviews to add a few sentences to explain why we perform TOC corrections for marine sedimentary Hg-concentrations.

L. 163–164: Clarify what the value of 1 standard deviation is (I assume that 48.6 ppb/wt% is the median value. This is key to indicate how far above the medium a value needs to get before it is considered anomalous.

We agree. We have added the value of 1 standard deviation ($1\sigma = 64.3$ ppb/wt%) in text.

L. 208–209: Is there any other supporting evidence for this? In the sedimentological properties, or other plant fossils etc?

We do observe signs of reworking (reworked palynomorphs) in intervals that have higher abundance of malformed spores. However, this refers to specific types of reworked material that clearly displays signs of being transported (i.e. darkened/broken wall material). This is not found for the spore malformations and most other fern spores and therefore we infer an in-situ signal.

L. 243–246: Care is needed here. Yes, U-Pb ages suggest this, but not all CAMP lava flows have been dated using U-Pb geochronology. So whilst it is true to say that the youngest CAMP pulse dated by U-Pb geochronology is around 201.2 Ma, there are CAMP lavas higher in the stratigraphy that are thought to be approximately 300–400 kyr younger than the Preakness basalt, based on cyclostratigraphy (e.g., Whiteside et al., 2010, PNAS; Lindström et al., 2021), as well as the intrusive bodies already referred to by the authors.

It is true that the Hook Mountain Group basalts based on cyclostratigraphy (Olsen et al., 2003) can be placed 400 kyr after the TJB, but no robust radiometric dates are available. Also, the size of the flows in New Jersey and Morocco are of limited size, although erosion may have reduced the volume, and we therefore think that these might have had a small influence in the earliest Hettangian.

Of course, these lava flows may not have constituted a major magmatic pulse, but they are documented in both North America and Morocco, so there was some extrusive CAMP volcanism happening in the Planorbis, even if we don't know its precise age and extent. I think it would be fairer to say here that the evidence for most intense/significant CAMP volcanism is around the TJ boundary and just beforehand, with some indication of intrusive and extrusive magmatism in the early Hettangian, which was apparently less pronounced than the earlier activity, although the heavy erosion of the LIP makes this uncertain. But yes I fully agree with the authors that there is no clear evidence for CAMP activity in the middle–late Hettangian that can explain the repeated Hg cycle disturbances recorded throughout that stage at this site.

We agree with the reviewer and have accordingly changed the text to reflect the comments above.

L. 264: I think that ‘largely facilitated’ would be better here.
We agree and have made the changes to reflect the comments.

L. 284: No comma is needed after CIE.
Done

L. 289–290: A reference from modern-day environmental studies is needed to support this statement.
There is no reference that we know of that suggest this statement. The statement was a deduction of the other information that exists on environmental Hg mediation. However, this statement is not necessary for the larger argumentation in the discussion and therefore we have omitted it from the manuscript.

L. 293: Change ‘unlikely driven’ to ‘unlikely to have been caused’.
Done

L. 324: ‘in those areas.’
Done

L. 335–336: But would these values of MDF (or even MIF) vary from coal to coal? Worth adding this caveat.
This is true. There is some variability in the MDF and MIF values from coal to coal. Not to the extent that is unrecognizable, but it could vary. We have added some words to this sentence to address this.

L. 391–392: Possibly, but I think this is a little bit speculative, unless it has been documented by studies of modern environments. If this has been previously reported, then a reference is needed. If not, then yes perhaps the form of Hg (e.g., gaseous) is important, but is it necessarily about that or about the concentration of the Hg? Also since this is the Spelae CIE being discussed, when CAMP was definitely still active, this is the mutagenesis pulse where UV-forced mutagenesis can’t be ruled out. So there are a few potentially complicating factors here.
Similar to the comments of the reviewers more major concerns, we have added a sentence that further argues that the similarities between the morphological traits of spore malformation throughout the Schandelah-1 section, suggest a single underlying cause that is most obviously attributed to the distribution of gaseous Hg. (See response to earlier comments).

L. 395–396: Is volatilisation from photo-chemical reduction necessarily needed to provide a source of Hg for mutagenesis (assuming that gaseous Hg is the crucial form for that to happen)? Could the increased wildfires have provided an adequate source alone? I get that they impact the MIF signature a lot, but if a change in runoff is also happening (as stated earlier in the discussion) then that could also be playing a major role in MIF variability. To me, it feels like there are several drivers operating. Have there been any modern studies on the potential toxicity of photochemically reduced Hg produced in certain ecosystems (such as the salt marshes mentioned previously)?
Similar to the comments of the reviewers more major concerns, we have put less focus on

photochemical reduction and also addressed the possibility of wildfire Hg-emission contributing the gaseous Hg distribution and potentially inducing fern mutagenesis. Modern studies have recorded a higher degree of photochemical reduction (Hg-degassing) in salt marshes that is attributed to sunlight exposure (causing positive MIF excursions). However, no reports are made on the resulting toxicity of the degassed Hg or transferred MeHg.

L. 404: 'higher' would be better than 'high'.

Done

L. 427–429: I really like this sentence as a nice conclusion.

Thank you for the feedback!

Reviewer #3 (Remarks to the Author):

Review of revised manuscript 'Climate-forced Hg-remobilization driving mutagenesis in ferns in the aftermath of the end-Triassic extinction' by Bos et al.

The revised manuscript is in line with the reviewers suggestions, which have been addressed and implemented clearly.

- The presented data has been brought into context with previous studies on elemental Hg and Hg-isotopes. A new section was added to the discussion which compares Hg records from different sites. Additionally, a supplementary figure has been added presenting a stratigraphic correlation of published $\delta^{13}\text{C}$ and Hg data.
- A discussion of the Hg-isotope trends induced by thermogenic volatilization of Hg from sedimentary organic matter has been added.
- The effects of wildfires on the Hg isotope signature and the release of Hg has been added to the discussion.
- The eccentricity pacing of the occurrence of malformed spores and Hg-isotope shifts is now clearly presented and explained in the text figures.
- The naming of the Marshi and Spelae CIE have been brought in stratigraphic context in relation to the alternative nomenclature
- Previously published OI and HI data have been removed from the results section and linked to the original study in the discussion
- The low resolution of the Sinemurian data is clearly pointed out and it is referred to the hiatus

The text reads well and the supporting figures are clear. However, reading the revised version of I came across the following issues:

The main concern regards the palynology and spore teratology methodology. You state that folded, broken, or obscured specimen were omitted from the malformed categories. Furthermore, spores with reworked features such as darkened and broken wall material were excluded from the total count of malformations (Line 442 – 446). At the same time you state that spore malformations typically show no features indicating reworking (i.e. darkened/broken wall material) and conclude that the record represents an in situ signal (Line 209 – 211). This reads like a systematically introduced bias. I suggest to revise the methodology section and add more detail on how reworking in malformed spores was detected, documented and assessed in relation to reworked non-malformed spores. We understand the confusion. In fact, little to no malformed spores with reworked features were observed in the studied section. Therefore, nearly all the spore malformations that we counted are associated with an in-situ signal. We have changed the sentence in the Methodology section that states that only a few malformed spore specimens were observed in the studied section and that these were omitted from the malformation counts. Other types of reworked palynomorphs are recognized by the same standards and were counted separately. We also added this rationale to the methodology.

Minor comments:

Either the results or the methods section it needs a clear descriptions of where the studied sample come from. You mention the Schandelah-1 core in the introduction, and you mention

the Lower Saxony Basin (line 138), but in the following text you refer to the northern German Basin, which represents the greater structural feature that, at least to my understanding, encompasses the Lower Saxony Basin and the Danish Basin amongst other structures. I recommend to briefly mention that the studied samples are from the Schandelah-1 core drilled in the Lower Saxony basin, Germany, and in the following mention the structural relation to the German Basin, which then makes transparent how you extrapolate from the Schandelah-1 core to the wider area by correlation and comparison to the Danish Basin.

We have made changes in text that refer to the Lower Saxony Basin when we are specifically discussing the Schandelah-1 core and refer to the Central European Basin when discussing supra-regional trends. We do feel that we have made sufficient references to the Schandelah-1 core as the main subject of this study which is also stated in the headline of many figures.

Line 246–147: the study by Benca et al. (2018) concerns U-VB radiation during the end-Permian. Add reference for the passive and diffusive emission postdating the end-Triassic mass extinction.

This is true. We have made changes to this section based on the comments of reviewer #2 and have omitted this sentence due to the fact that Benca et al., 2018 refers to the end-Permian crisis.

Line 282–283: add reference concerning the generation of Hg through volcanic intrusions and its dependency on the composition of intruded sedimentary rock. To my knowledge Wignall 2001 is not addressing this.

This is true. Wignall, 2001 does not mention Hg in this context. We have changed this sentence to suggest that the generation of “volatiles” is dependent on the source rock intrusions which is mentioned in Wignall, 2001. We make the point a few sentences later that this can be applied to Hg generation as well, based on our observations.

Line 377: define the term S/P ratio

Done

Line 304–307: This section could need another backup that the malformed spores do not exhibit signs of reworking

Done

I recommend the publication of the manuscript after minor revisions regarding the methods section.

Kind regards,

References:

Benca, J.P., Duijnste, I.A.P. and Looy, C.V., 2018. UV-B–induced forest sterility: Implications of ozone shield failure in Earth’s largest extinction. *Science Advances* 4, e1700618, doi: doi:10.1126/sciadv.1700618.

Wignall, Paul B. "Large igneous provinces and mass extinctions." *Earth-science reviews* 53.1-2 (2001): 1-33.

REVIEWERS' COMMENTS

Reviewer #2 (Remarks to the Author):

The authors have done a good job of incorporating my comments and edits on the previous version of their manuscript. I am happy for the paper to be accepted, and I would like to congratulate the authors on a job well done.

Below are a few tiny corrections that the authors could implement when they are sent the proofs. But these really are tiny tweaks. For me, the manuscript can be published at this point.

L. 83: Change 'Hg-speciation' to 'Hg-cycling'. I missed this one previously.

L. 96: Change 'retaining' to 'retained'

L. 123–124: Change 'Hg-isotopes' to 'changes in Hg isotope compositions'.

L. 167: I think that 'most frequently employed' would be better than 'generally preferred'

L. 327: Change 'remain difficult to deduce' to 'remains challenging'

Reviewer #3 (Remarks to the Author):

All comments and suggestions have been addressed sufficiently. I have no further comments on the content. Listed below are some minor formal edit suggestions

- Use capital letters for Early/Lower and Late/Upper Jurassic, Rhaethian, Hettangian and Sinemurian
- Use n-dash for ranges, e.g. Hettangian–Sinemurian, atmospheric–ocean–terrestrial
- Consistently use a minus sign for negative numbers (some numbers have a hyphen instead, e.g. Line 227)
- Since the Marshi and Spelae CIEs refer to the respective ammonite zone I would suggest to use italics
- In the main text you refer to sub figures with capital letters (e.g., Fig. 4A) while you use small letters in the figure and figure captions (Fig. 4a...)
- Fig. 4 – The solid line separating the Hettangian and Sinemurian stages should not be drawn through the Lower Jurassic column

Reviewer #2 (Remarks to the Author):

The authors have done a good job of incorporating my comments and edits on the previous version of their manuscript. I am happy for the paper to be accepted, and I would like to congratulate the authors on a job well done.

Below are a few tiny corrections that the authors could implement when they are sent the proofs. But these really are tiny tweaks. For me, the manuscript can be published at this point.

L. 83: Change 'Hg-speciation' to 'Hg-cycling'. I missed this one previously.

Done

L. 96: Change 'retaining' to 'retained'

Done

L. 123–124: Change 'Hg-isotopes' to 'changes in Hg isotope compositions'.

Done

L. 167: I think that 'most frequently employed' would be better than 'generally preferred'

Done

L. 327: Change 'remain difficult to deduce' to 'remains challenging'

Done

Reviewer #3 (Remarks to the Author):

All comments and suggestions have been addressed sufficiently. I have no further comments on the content. Listed below are some minor formal edit suggestions

- Use capital letters for Early/Lower and Late/Upper Jurassic, Rhaethian, Hettangian and Sinemurian

The Early/Lower and Late/Upper notations of the Jurassic and Triassic are (chrono)stratigraphic defined sequences and are therefore capitilised. This is not the case for the stages Rhaetian, Hettangian and Sinemurian. When we refer to early/lower and late/upper in these stages, we refer to a stratigraphic section that is defined by depths in the core and often linked to ammonite zones, but not a formely recognized stratigraphic domain. Therefore, we have decided to not capitilise the stage intervals.

- Use n-dash for ranges, e.g. Hettangian–Sinemurian, atmospheric–ocean–terrestrial
Done

- Consistently use a minus sign for negative numbers (some numbers have a hyphen instead, e.g. Line 227)
Done

- Since the Marshi and Spelae CIEs refer to the respective ammonite zone I would suggest to use italics

We understand this suggestion. However, the previous studies that have adopted this terminology use these terms without italics. These names are derived from species names which are also written without captical letters. To avoid confusion, we have decided to don't use italics.

- In the main text you refer to sub figures with capital letters (e.g., Fig. 4A) while you use small letters in the figure and figure captions (Fig. 4a...)
Done

- Fig. 4 – The solid line separating the Hettangian and Sinemurian stages should not be drawn through the Lower Jurassic column
Done